# Multi-Agent Learning under Uncertainty: Recurrence vs. Concentration

**Kyriakos Lotidis**
Stanford University
klotidis@stanford.edu

**Panayotis Mertikopoulos**
Univ. Grenoble Alpes, CNRS, Inria, Grenoble INP
LIG 38000 Grenoble, France
panayotis.mertikopoulos@imag.fr

**Nicholas Bambos**
Stanford University
bambos@stanford.edu

**Jose Blanchet**
Stanford University
jose.blanchet@stanford.edu

## Abstract

In this paper, we examine the convergence landscape of multi-agent learning under uncertainty. Specifically, we analyze two stochastic models of regularized learning in continuous games—one in continuous and one in discrete time—with the aim of characterizing the long-run behavior of the induced sequence of play. In stark contrast to deterministic, full-information models of learning (or models with a vanishing learning rate), we show that the resulting dynamics *do not converge* in general. In lieu of this, we ask instead which actions are played more often in the long run, and by how much. We show that, in strongly monotone games, the dynamics of regularized learning may wander away from equilibrium infinitely often, but they always return to its vicinity in *finite* time (which we estimate), and their long-run distribution is sharply concentrated around a neighborhood thereof. We quantify the degree of this concentration, and we show that these favorable properties may all break down if the underlying game is not strongly monotone—underscoring in this way the limits of regularized learning in the presence of persistent randomness and uncertainty.

## 1 Introduction

In its most abstract form, the standard model for online learning in games unfolds as follows: (*i*) at each stage of the process, every participating agent selects an action; (*ii*) the agents receive a reward determined by their chosen actions and their individual payoff functions; (*iii*) the agents update their actions, and the process repeats. In this general context, the agents have to contend with various—and varying—degrees of uncertainty: (*a*) uncertainty about the game, the strategic interests of other players, and/or who else is involved in the game; (*b*) uncertainty about the outcomes of their actions, and which update directions may lead to better outcomes; and (*c*) uncertainty stemming from the environment, manifesting as random shocks to the players' payoffs and/or other disturbances. In this regard, uncertainty could be either endogenous or exogenous; but, in either case, it leads to players having to take decisions with very limited information at their disposal.

Our goal in this paper is to quantify the impact of uncertainty on multi-agent learning—and, more precisely, to understand the differences that arise in the players' long-run behavior when such uncertainty is present versus when it is not. A natural framework for exploring this question is within the greater setting of no-regret learning and, in particular, the family of *"follow-the-regularized-leader"* (FTRL) algorithms and dynamics [38, 64, 65]. This class contains several mainstay learning methods—like online gradient descent (or, in our case, *ascent*) [74], the exponential / multiplicative

weights (EW) algorithm and its variants (HEDGE, EXP3, etc.) [1, 2, 40, 69], and many others—so it has become practically synonymous with the notion of online learning in games. Accordingly, we seek to answer the following questions:

*What is the long-run distribution of regularized learning under uncertainty?*
*Which actions are played more often, and by how much?*
*Do the dynamics concentrate—and, if so, where?*

**Our contributions in the context of related work.** Needless to say, the interpretation of these questions is context-specific, and it depends on the particular learning setting at hand. In this paper, motivated by applications to machine learning, signal processing and data science (which typically involve continuous action spaces and rewards), we focus on *continuous games*, and we consider two models of regularized learning, one in continuous time, and one in discrete time.

In continuous time, we model the dynamics of FTRL in the presence of uncertainty as a stochastic differential equation (SDE) perturbed by a general Itô diffusion process, i.e., a continuous-time martingale with possibly colored and/or correlated components. In the context of *finite* games, models of this type have been studied by, among others, Bravo & Mertikopoulos [10], Foster & Young [20], Fudenberg & Harris [21] and Mertikopoulos & Moustakas [43, 44], the first two in an evolutionary setting, the latter as a continuous-time model of the EW algorithm in the presence of random disturbances. Follow-up works in this direction include [10–12, 18, 25, 28, 49] on *finite* games, while [33–35, 47] considered a regularized learning model in convex minimization problems. The model which is closest to our own is that of [46, 48], who study the regret properties and guarantees of a stochastic version of the dual averaging dynamics of Nesterov [57].

At a high level, our findings reveal a crisp dichotomy between games that are *null-monotone* (like bilinear min-max games or zero-sum bimatrix games), and *strongly monotone* games (like Kelly auctions, Cournot competitions, joint signal covariance optimization problems, etc.). Specifically:

1. In null-monotone games, uncertainty induces a persistent drift *away from equilibrium*: the dynamics reach greater distances from equilibrium in finite time (which we estimate) and they require, on average, infinite time to return. In particular, if the game admits an interior equilibrium, the dynamics diffuse away—escaping in the mean toward infinity or to the boundary of the game's action space—and they exhibit *no concentration* in any region of interior actions.

2. In strongly monotone games, uncertainty still induces a persistent outward drift, but this is now partially countered by the dynamics' deterministic component. Thus, in stark contrast to the null-monotone case, the players' learning trajectories end up in a near-equilibrium region whose size scales with the level of uncertainty, and we estimate both the size of this region and the time required to reach it. Somewhat paradoxically, the dynamics return with probability 1 arbitrarily close to where they started, infinitely often, in a way reminiscent of Poincaré recurrence in bimatrix min-max games [52, 59]; however, these returns can be exceedingly far apart, so there is no antinomy.

In discrete time, we consider a standard implementation of FTRL with a constant learning rate and stochastic first-order oracle feedback. Variants with a vanishing learning rate have been studied extensively in the stochastic approximation literature, and they are known to exhibit favorable convergence guarantees in, among others, strongly monotone games, cf. [50, 54] and references therein. At the same time however, these properties typically come at the expense of the algorithm slowing down to a crawl; for this reason, owing to their simplicity, robustness, and superior empirical performance, constant / non-vanishing learning rate schedules are much more common in practice.

On the downside, the long-run behavior of FTRL is much less understood in this case. To the best of our knowledge, the most relevant results come from recent works by Loizou et al. [41] and Huang & Zhang [27], who established upper bounds on the mean distance to equilibrium for stochastic gradient descent / ascent in strongly monotone games, and Vlatakis et al. [68], who studied the ergodic properties of constant step-size variants of the stochastic extragradient and stochastic gradient descent–ascent algorithms for weakly quasi-strongly monotone variational inequalities. Dually to this, in the null-monotone regime, Cauvin et al. [12] showed that FTRL exhibits a similar tendency to escape from interior equilibria in finite min-max and harmonic games; our continuous-time analysis is, in this view, an extension of the corresponding result of [12].

One reason that results about the statistics of the long-run behavior of FTRL are particularly scarce in the literature is that, in discrete time, even the most basic tools of stochastic analysis are often inapplicable; for an illustration of the difficulties involved, see e.g., Azizian et al. [3, 4] and references therein. Nevertheless, based in no small part on the insights gained by our continuous-time analysis, we manage to establish the following version of the strong-null dichotomy in discrete time:

1. In null-monotone games with an unbounded action space, the sequence of play under FTRL drifts away to infinity on average (though not necessarily with probability 1).
2. In strongly monotone games, we show that the mean time required to reach a given distance from the game's equilibrium is finite, and we provide an explicit estimate thereof. If the game's equilibrium is interior, we also show that FTRL converges strongly to a unique invariant measure, which is concentrated in a certain region around the game's equilibrium, which we also estimate.

We find these results particularly appealing as they provide the first glimpse into the distributional properties of multi-agent regularized learning under uncertainty.

## 2  Preliminaries

**2.1. Continuous games.** Throughout the sequel, we consider games with a finite number of players and a continuum of actions per player. Formally, players will be indexed by $i \in \mathcal{N} = \{1, \dots, N\}$ and, during play, each player will be selecting an action $x_i$ from a closed convex subset $\mathcal{X}_i$ of some $d_i$-dimensional normed space $\mathcal{V}_i$. Aggregating over all players, we will write $\mathcal{X} = \prod_i \mathcal{X}_i$ for the space of the players' joint action profiles $x = (x_1, \dots, x_N)$ and $d = \sum_i d_i$ for the dimension of the ambient space $\mathcal{V} = \prod_i \mathcal{V}_i$. Finally, we will use the shorthand $x = (x_i; x_{-i})$ when we want to highlight the action of player $i \in \mathcal{N}$ against the action profile $x_{-i} = (x_j)_{j \neq i}$ of all other players—and, in similar notation, $\mathcal{X}_{-i} = \prod_{j \neq i} \mathcal{X}_j$ for the space thereof.

The reward of each player $i \in \mathcal{N}$ in a given action profile will be determined by an associated payoff function $u_i \colon \mathcal{X} \to \mathbb{R}$, assumed here to be *individually concave* in the sense that $u_i(x_i; x_{-i})$ is concave in $x_i$ for all $x_{-i} \in \mathcal{X}_{-i}$. We will further assume that each $u_i$ is $\beta$-Lipschitz smooth, and we will write respectively

$$v_i(x) = \nabla_{x_i} u_i(x_i; x_{-i}) \quad \text{and} \quad v(x) = (v_1(x), \dots, v_N(x)) \tag{1}$$

for the individual gradient field of each player and the ensemble thereof.[1]

The tuple $\mathcal{G} \equiv \mathcal{G}(\mathcal{N}, \mathcal{X}, u)$ will be referred to as a *concave game* [62]. Mainstay examples of such games include (mixed extensions of) finite games, resource allocation problems, Kelly auctions, Cournot competitions, etc.; for completeness, we detail some of these applications in Appendix A.

**2.2. Nash equilibrium.** The leading solution concept in game theory is that of a *Nash equilibrium*, defined here as an action profile $x^* \in \mathcal{X}$ which discourages unilateral deviations, i.e.,

$$u_i(x^*) \geq u_i(x_i; x^*_{-i}) \quad \text{for all } x_i \in \mathcal{X}_i \text{ and all } i \in \mathcal{N}. \tag{NE}$$

A concave game always admits a Nash equilibrium if $\mathcal{X}$ is compact, and it admits a *unique* equilibrium if the game is strongly monotone in the sense of Definition 1 below:

**Definition 1.** A game $\mathcal{G} \equiv \mathcal{G}(\mathcal{N}, \mathcal{X}, u)$ is called $\alpha$-*monotone* if there exists some $\alpha \geq 0$ such that

$$\langle v(x') - v(x), x' - x \rangle \leq -\alpha \|x' - x\|^2 \quad \text{for all } x, x' \in \mathcal{X}. \tag{Mon}$$

If (Mon) holds for some $\alpha > 0$, the game will be called *strongly monotone*; otherwise, if (Mon) only holds for $\alpha = 0$, $\mathcal{G}$ will be called *merely monotone* (or simply monotone when the distinction is not important). Finally, if (Mon) binds for $\alpha = 0$ and all $x, x' \in \mathcal{X}$—that is, $\langle v(x') - v(x)), x' - x \rangle = 0$ for all $x, x' \in \mathcal{X}$—the game will be called *null-monotone*. ☙

*Remark* 1. Merely monotone games could be viewed as a "hybrid" between null and strictly monotone games: generically, at any given action profile of a merely monotone game, there would be directions of motion where the (symmetrized) Jacobian of the players' gradient field has a zero eigenvalue, and directions with positive eigenvalues; either set (but not both) could be empty, the former corresponding to the "null-monotone" directions, the latter corresponding to the "strongly monotone" ones. ☙

---

[1]We are tacitly assuming here that the players' payoff functions are defined in an open neighborhood of $\mathcal{X}$ in $\mathcal{V}$; this assumption is done only for convenience, and it does not affect any of our results.

*Remark* 2. A weighted variant of (Mon) is sometimes called *diagonal* (*strict* / *strong*) *concavity*, in reference to the work of Rosen [62]; for a pointed version of these conditions known as *variational stability* [26, 50, 54] or *coherence* [53, 73]. These variants will not be important for our purposes. ☙

**2.3. Regularized learning.** In the rest of our paper, we will consider a family of online learning schemes adhering to the following model of "regularized learning": players aggregate gradient feedback on their payoff functions over time and, at each instance of play, they choose the action which is most closely aligned to this aggregate. We provide a detailed description of this model in Sections 3 and 4—in continuous and discrete time respectively—and only describe here the core idea.

At a high level, the common denominator of these schemes is the way that players choose their actions based on the accumulation of payoff gradients over time. Formally, we will treat payoff gradients as dual vectors and we will write $\mathcal{Y}_i \coloneqq \mathcal{V}_i^*$ for the dual space of $\mathcal{V}_i$ and $\mathcal{Y} = \prod_i \mathcal{Y}_i = \mathcal{V}^*$ for the ensemble thereof. Then, given an aggregate of gradient steps $y_i \in \mathcal{Y}_i$, we will assume that the $i$-th player chooses an action via a "*generalized projection*"—or *mirror*—map $Q_i \colon \mathcal{Y}_i \to \mathcal{X}_i$ of the general form

$$Q_i(y_i) = \arg\max_{x_i} \{\langle y_i, x_i \rangle - h_i(x_i)\} \quad \text{for all } y_i \in \mathcal{Y}_i. \tag{2}$$

In the above $h_i \colon \mathcal{X}_i \to \mathbb{R}$ is a continuous $K_i$-strongly convex function, that is,

$$h_i(\lambda x_i + (1 - \lambda)x_i') \le \lambda h_i(x_i) + (1 - \lambda)h_i(x_i') - \tfrac{1}{2}K_i \lambda(1 - \lambda)\|x_i' - x_i\|^2 \tag{3}$$

for all $x_i, x_i' \in \mathcal{X}_i$ and all $\lambda \in [0, 1]$. This function is known as the *regularizer* of the method and it acts as a penalty term that smooths out the "hard" arg max correspondence $y_i \mapsto \arg\max_i \langle y_i, x_i \rangle$. This regularization scheme has a very long and rich history in game theory and optimization, where $Q$ is often referred to as a "*quantal*" or "*regularized*" best response operator, cf. [38, 42, 50, 64, 67] and references therein. For concreteness, we describe below the two leading examples of this regularization setup (suppressing in both cases the player index $i \in \mathcal{N}$ for notational clarity):

**Example 1** (Euclidean regularization). Let $h(x) = \tfrac{1}{2}\|x\|_2^2$. Then (B.12) boils down to the Euclidean projection map

$$Q(y) = \Pi_{\mathcal{X}}(y) \equiv \arg\max_{x \in \mathcal{X}} \|y - x\|_2. \tag{4}$$

Thus, in particular, if $\mathcal{X} = \mathcal{V}$, we readily recover the identity map $Q(y) = y$. ☙

**Example 2** (Entropic regularization). Let $\mathcal{X} = \{x \in \mathbb{R}_+^d : \sum_{k=1}^d x_k = 1\}$ be the unit simplex of $\mathbb{R}^d$, and let $h(x) = \sum_{k=1}^d x_k \log x_k$ denote the (negative) entropy on $\mathcal{X}$. Then (B.12) yields the *logit* map

$$Q(y) = \Lambda(y) \equiv \frac{(\exp(y_1), \ldots, \exp(y_d))}{\exp(y_1) + \cdots + \exp(y_d)}. \tag{5}$$

This map forms the basis of the seminal Hedge and Exp3 algorithms in online learning, cf. [1, 2, 13, 38, 40, 64] and references therein. ☙

To ease notation in the sequel, we will write $h(x) \coloneqq \sum_i h_i(x_i)$ for the players' aggregate regularizer, $K \coloneqq \min_i K_i$ for the strong convexity modulus of $h$, and $Q \coloneqq \prod_i Q_i \colon \mathcal{Y} \to \mathcal{X}$ for the resulting ensemble mirror map. In the next sections, we describe in detail how this regularization setup is used in a learning context.

# 3 Learning under uncertainty in continuous time

To set the stage for the sequel, we begin with two simple games that will serve as "minimal working examples" for the more general model and results presented in the sections to come. We focus for the moment on continuous-time interactions; the discrete-time setting is presented in Section 4.

**3.1. A gentle start.** Consider the following 2-player, convex-concave min-max games:

$$(a) \quad \text{Bilinear saddle:} \qquad u_1(x_1, x_2) = -u_2(x_1, x_2) = -x_1 x_2 \qquad \text{for } x_1, x_2 \in \mathbb{R}. \tag{6a}$$

$$(b) \quad \text{Quadratic saddle:} \qquad u_1(x_1, x_2) = -u_2(x_1, x_2) = x_2^2/2 - x_1^2/2 \qquad \text{for } x_1, x_2 \in \mathbb{R}. \tag{6b}$$

Both games are monotone and they admit a unique Nash equilibrium at the origin. Their gradient fields are $v(x_1, x_2) = (-x_2, x_1)$ and $v(x_1, x_2) = -(x_1, x_2)$ respectively, so the first game is *null-monotone* and the second one is 1-*strongly monotone*. Accordingly, if each player follows their individual payoff gradient to increase their rewards, we obtain the gradient descent / ascent dynamics

$$(a) \quad \dot{x}(t) = (-x_2(t), x_1(t)) \quad \text{and} \quad (b) \quad \dot{x}(t) = -(x_1(t), x_2(t)) \tag{GDA}$$

for the bilinear and quadratic games (6a) and (6b) respectively. It is then trivial to see that, in the bilinear case, (GDA) cycles periodically at a constant distance from the game's equilibrium, whereas, in the quadratic case, the dynamics converge to the game's equilibrium at a geometric rate.

To model uncertainty in this setting, we will consider the stochastic gradient dynamics

$$dX(t) = v(X(t)) \, dt + \sigma \, dW(t) \tag{S-GDA}$$

where $W(t) = (W_1(t), W_2(t))$ is a Brownian motion in $\mathbb{R}^2$ and $\sigma > 0$ is the magnitude of the noise entering the process. Intuitively, this SDE should be viewed as a rigorous formulation of the informal model $\dot{x}(t) = v(x(t)) +$ "noise", with the Brownian term $W(t)$ capturing all sources of randomness and uncertainty in the players' environment.[2] Consequently, to understand the impact of uncertainty in each case of (GDA), we will examine the following quantities:

1. The distance $\|X(t)\|_2^2$ of $X(t)$ from the game's equilibrium (that is, the origin of $\mathbb{R}^2$).
2. The time $\tau_r = \inf\{t > 0 : \|X(t)\|_2 \le r\}$ at which $X(t)$ gets within $r$ of the game's equilibrium.
3. The density $\mathcal{P}(x, t)$ of $X(t)$—and, if it exists, its long-run limit $\mathcal{P}_\infty(x) := \lim_{t \to \infty} \mathcal{P}(x, t)$.

When it exists, $\mathcal{P}_\infty$ is known as the *stationary*—or *invariant*—*distribution* of $X$, and it is closely related to the *occupation measure* $\mu_t$ of the process, defined here as

$$\mu_t(\mathcal{B}) = \frac{1}{t} \int_0^t \mathbb{1}\{X(s) \in \mathcal{B}\} \, ds \quad \text{for every Borel } \mathcal{B} \subseteq \mathcal{X}. \tag{7}$$

Under mild ergodicity conditions [30, Cor. 25.9], we have $\lim_{t \to \infty} \mu_t(\mathcal{B}) = \int_\mathcal{B} \mathcal{P}_\infty$ so, concretely, $\mathcal{P}_\infty$ measures the fraction of time that $X(t)$ spends in a given subset of $\mathcal{X}$ in the long run.

Taken together, these metrics provide a fairly complete picture of the statistics of $X(t)$ so, in the rest of this section, we analyze them in the context of (S-GDA) applied to the games (6a) and (6b).

**Case 1: Bilinear saddles.** In this case, by a direct application of Itô's formula—the chain rule of stochastic calculus [58, Chap. 4]—we readily obtain

$$d(\|X(t)\|_2^2) = 2X(t) \cdot dX(t) + dX(t) \cdot dX(t) = 2\sigma^2 \, dt + \sigma X(t) \cdot dW(t). \tag{8}$$

This suggests that, on average, $\|X(t)\|_2^2$ increases as $\Theta(\sigma^2 t)$. Building on this observation, we show in Appendix D that the dynamics (S-GDA) for the bilinear game (6a) enjoy the following properties:

**Proposition 1.** *Suppose that* (S-GDA) *is run on the game* (6a) *with initial condition* $x_0 \in \mathbb{R}^2$. *Then:*

1. $\lim_{t \to \infty} \mathbb{E}_{x_0}[\|X(t)\|_2^2] = \infty$, *i.e.,* $X(t)$ *escapes to infinity in mean square.*

2. $\mathbb{E}_{x_0}[\tau_r] = \infty$ *if* $r < \|x_0\|$, *i.e.,* $X(t)$ *takes infinite time on average to get closer to equilibrium.*

3. *The limit* $\mathcal{P}_\infty(x) = \lim_{t \to \infty} \mathcal{P}(x, t)$ *does not exist, i.e.,* $X$ *does not admit an invariant distribution.*

Proposition 1 shows that, in the presence of uncertainty, the periodicity of the deterministic dynamics (GDA) is completely destroyed. In fact, despite random fluctuations that occasionally bring $X(t)$ closer to equilibrium, (S-GDA) exhibits a consistent drift *away from equilibrium*, escaping any compact set in finite time and requiring infinite time to return on average. As a result, $X(t)$ becomes infinitely spread out in the long run, exhibiting no measurable concentration in *any* region of $\mathbb{R}^2$. For a partial illustration of this behavior—which we view as antithetical to convergence—cf. Fig. 1. ☙

**Case 2: Quadratic saddles.** We now proceed to examine the behavior of (S-GDA) in the quadratic min-max problem (6b), where (S-GDA) gives

$$dX(t) = -X(t) \, dt + \sigma \, dW(t). \tag{9}$$

As is well known [36, Chap. 7.4], this SDE describes the 2-dimensional *Ornstein–Uhlenbeck* (OU) process

$$X(t) = X(0)e^{-t} + \sigma \int_0^t e^{-(t-s)} \, dW(s). \tag{OU}$$

Hence, by unfolding the stochastic integral in (OU), we can draw the following conclusions:

**Proposition 2.** *Suppose that* (S-GDA) *is run on the game* (6b) *with initial condition* $x_0 \in \mathbb{R}^2$. *Then:*

---

[2]For a primer on SDEs, see [36, 58]; for completeness, we also present some basic definitions in Appendix C.

1. $\lim_{t\to\infty} \mathbb{E}_{x_0}\left[\|X(t)\|_2^2\right] = \sigma^2$, *i.e., the dynamics fluctuate at mean distance $\sigma$ from equilibrium.*

2. *The mean time required to get within distance $r$ of the game's equilibrium is bounded as*

$$\mathbb{E}_{x_0}[\tau_r] \le \frac{1}{2} \frac{\|x_0\|_2^2 - r^2}{r^2 - \sigma^2} \qquad \text{for all } \sigma < r < \|x_0\|_2. \tag{10}$$

3. *The density of $X(t)$ is $\mathcal{P}(x,t) = [\pi\sigma^2(1 - e^{-2t})]^{-1}\exp\left(-\frac{\|x - e^{-t}x_0\|_2^2}{(1 - e^{-2t})\sigma^2}\right)$. In particular, $X(t)$ converges in distribution to a Gaussian random variable centered at $0$, viz.*

$$\mathcal{P}_\infty(x) \equiv \lim_{t\to\infty} \mathcal{P}(x,t) = 1/(\pi\sigma^2) \cdot e^{-\|x\|_2^2/\sigma^2}. \tag{11}$$

Proposition 2 shows that the geometric convergence properties of the deterministic dynamics (GDA) are again destroyed in the presence of uncertainty. However, in stark contrast to Proposition 1 for the bilinear case, $X(t)$ now exhibits a consistent drift *toward equilibrium*, and it ends up being sharply concentrated at a distance of $\mathcal{O}(\sigma^2)$ from equilibrium. This interplay between recurrence and concentration will play a crucial role in the sequel, and our aim in the rest of this section will be to quantify the extent to which it holds in a more general setting.

**3.2. Learning in continuous time.** We now proceed to describe our general model for multi-agent learning under uncertainty, hinging on the stochastic *"follow-the-regularized-leader"* template

$$dY_i(t) = v_i(X(t))\,dt + dM_i(t) \qquad X_i(t) = Q_i(Y_i(t)). \tag{S-FTRL}$$

In the above, $(i)$ $Y_i(t) \in \mathcal{Y}_i$ is a "score" variable that tracks the aggregation of individual payoff gradients in $\mathcal{Y}_i$; $(ii)$ $M_i(t) \in \mathcal{Y}_i$ is a continuous square-integrable martingale acting as a catch-all, "colored noise" disturbance term; and $(iii)$ $Q_i : \mathcal{Y}_i \to \mathcal{X}_i$ is the regularized mirror map of player $i \in \mathcal{N}$, as per (B.12). In this regard, (S-FTRL) represents a noisy "stimulus-response" mechanism, where each player $i \in \mathcal{N}$ tracks the aggregation of payoff gradients under uncertainty—the "stimulus"—and "responds" to this aggregate via their individual regularized mirror map $Q_i$.

*Remark* 3. The terminology "follow-the-regularized-leader" is due to [64, 65], who first studied this scheme in the context of online convex optimization in discrete time. This family of algorithms and dynamics has been widely studied in the literature; we provide more details on this in Appendix B. ❧

For concreteness, we will assume that the noise term $M(t) = (M_i(t))_{i\in\mathcal{N}}$ in (S-FTRL) is of the form

$$dM(t) = \sigma(X(t)) \cdot dW(t) \quad \text{or, more explicitly} \quad dM_i(t) = \sigma_i(X(t)) \cdot dW(t) \tag{12}$$

where $W(t) = (W_1(t), \ldots, W_m(t))$ is a standard Brownian motion in $\mathbb{R}^m$, and $\sigma(x) = (\sigma_i(x))_{i\in\mathcal{N}}$ is an ensemble of state-dependent *diffusion matrices* $\sigma_i : \mathcal{X}_i \to \mathbb{R}^{d_i \times m}$, $i \in \mathcal{N}$.[3] Importantly, the model (12) allows for *correlated uncertainty* between different components of the process—e.g., accounting for random disturbances on shared road segments in a congestion game—so it will be the base model for our analysis. Our only standing assumption will be that $\sigma : \mathcal{X} \to \mathbb{R}^{d \times m}$ is bounded and Lipschitz continuous, which ensures that (S-FTRL) is *well-posed*, i.e., it admits a unique strong solution that exists for all time and for every initial condition $Y(0) \leftarrow y \in \mathcal{Y}$ (cf. Appendix C).

*Remark* 4. To connect the above with Section 3.1, note that (S-GDA) is recovered from (S-FTRL) by taking $\mathcal{X}_i = \mathbb{R}$, $M_i(t) = \sigma W_i(t)$, and $h_i(x_i) = x_i^2/2$ for $i = 1, 2$ (so $Q_i(y_i) = y_i$ by Example 1). ❧

**3.3. Analysis and results.** We now proceed to describe our main results for the stochastic dynamics (S-FTRL)—which, as we show shortly, reflect the dichotomy between bilinear and quadratic saddle-point problems that we noted in Section 3.1. To state them, it will be convenient to introduce a "primal-dual" generalization of the Euclidean distance that is more closely aligned with the regularization setup underlying the players' response scheme. Deferring the details to Appendix B, we define here the *Fenchel coupling* induced by the regularizer $h_i$ of player $i \in \mathcal{N}$ as

$$F_i(p_i, y_i) = h_i(p_i) + h_i^*(y_i) - \langle y_i, x_i\rangle \quad \text{for all } i \in \mathcal{N}, \text{ and all } p_i \in \mathcal{X}_i, y_i \in \mathcal{Y}_i \tag{13}$$

---

[3]The representation (12) of $M(t)$ via a Brownian integrator is not an assumption per se, but a consequence of the martingale representation theorem, which allows us to express any homogeneous square-integrable martingale in this form [58, Thm. 4.3.4]. Under this light, the loss in generality is negligible in our case.

where $h_i^*(y_i) := \max_{x_i \in \mathcal{X}_i}\{\langle y_i, x_i \rangle - h_i(x_i)\}$ denotes the convex conjugate of $h_i$. For example, in the unconstrained Euclidean case (Example 1), we recover the Euclidean distance squared, viz. $F_i(p_i, y_i) = \frac{1}{2}\|Q_i(y_i) - p_i\|^2$; by comparison, under entropic regularization on the simplex (Example 2), we get a "dualized" version of the Kullback–Leibler divergence, cf. Appendix B. In all cases, $F_i$ is *positive-semidefinite* in the sense that $F_i(p_i, y_i) \geq 0$ for all $y_i \in \mathcal{Y}_i$, with equality if and only if $Q_i(y_i) = p_i$. In view of this, the total coupling $F(x, y) := \sum_i F_i(p_i, y_i)$ is a valid measure of "divergence" between $p \in \mathcal{X}$ and $y \in \mathcal{Y}$, and we will use it freely in the sequel as such.

The last ingredient that we will need is two measures of the amount of randomness in (S-FTRL), viz.

$$\sigma_{\min}^2 := \min_{x \in \mathcal{X}} \lambda_{\min}(\Sigma(x)) \quad \text{and} \quad \sigma_{\max}^2 := \max_{x \in \mathcal{X}} \lambda_{\max}(\Sigma(x)) \tag{14}$$

where $\Sigma \equiv \sigma\sigma^\top$ denotes the quadratic covariation matrix of the martingale $M(t)$, and $\lambda_{\min}$ (resp. $\lambda_{\max}$) denotes the minimum (resp. maximum) eigenvalue thereof.

With all this in hand, we will focus on two broad classes of games, *null-monotone* and *strongly monotone*, of which the bilinear and quadratic examples of Section 3.1 are archetypal examples. To state our results, we will assume that (S-FTRL) is initialized at $x_0 \leftarrow Q(y_0) \in \operatorname{ri} \mathcal{X}$ for some $y_0 \in \mathcal{Y}$, and we will write $F_t \equiv F(x^*, Y(t))$ where $x^*$ is an equilibrium of the game. We then have:

**Theorem 1** (Null-monotone games). *Suppose that* (S-FTRL) *is run with a smooth mirror map $Q$ in a null-monotone game $\mathcal{G}$. Suppose further that the game admits an interior equilibrium $x^*$, and consider the hitting times $\tau_\varepsilon^- := \inf\{t > 0 : F_t \leq F_0 - \varepsilon\}$ and $\tau_\varepsilon^+ := \inf\{t > 0 : F_t \geq F_0 + \varepsilon\}$. If $\sigma_{\min}^2 > 0$ and $\varepsilon > 0$ is small enough, then*

$$\mathbb{E}_{x_0}[\tau_\varepsilon^-] = \infty \quad \text{and} \quad \mathbb{E}_{x_0}[\tau_\varepsilon^+] \leq 2\varepsilon/(\kappa\,\sigma_{\min}^2) \tag{15}$$

*for some constant $\kappa \equiv \kappa_\varepsilon > 0$; in addition, $X(t)$ does not admit a limiting distribution in this case.*

**Theorem 2** (Strongly monotone games). *Suppose that* (S-FTRL) *is run in an $\alpha$-strongly monotone game $\mathcal{G}$, and consider the hitting time*

$$\tau_r := \inf\{t > 0 : X(t) \in \mathbb{B}_r(x^*)\} \tag{16}$$

*where $\mathbb{B}_r(x^*) = \{x : \|x - x^*\| \leq r\}$ is a ball of radius $r$ centered on the (necessarily unique) equilibrium $x^*$ of $\mathcal{G}$. Then:*

$$\mathbb{E}_{x_0}[\tau_r] \leq (F_0/\alpha)/(r^2 - r_\sigma^2) \quad \text{for all } r > r_\sigma, \tag{17}$$

*where $r_\sigma := \sigma_{\max}/\sqrt{2K\alpha}$. If, in addition, $\sigma_{\min} > 0$ and $x^*$ is interior, $X(t)$ admits an invariant distribution concentrated in a ball of radius $\mathcal{O}(\sigma_{\max})$ around $x^*$, and we have*

$$\lim_{t \to \infty} \mu_t(\mathbb{B}_r(x^*)) \geq 1 - r_\sigma^2/r^2 \quad \text{for all } r > r_\sigma. \tag{18}$$

*Remark* 5. The bounds depend implicitly on the regularizer through its strong convexity modulus $K$ and they indicate a trade-off between the degree of concentration of the process around the radius beyond which the noise dominates the drift, and the time required to hit this region. ❦

*Remark* 6. The result of Theorem 2 holds for radius of concentration not sharper than $\mathcal{O}(\sigma)$. This coincides with the special case of Proposition 2, indicating that our bound is tight in this regard. ❦

Conceptually, Theorems 1 and 2 reflect the dichotomy between the bilinear and quadratic examples studied in detail in Section 3.1. Indeed, we see that:

1. In *null-monotone games*, the stochastic dynamics (S-FTRL) exhibit a consistent drift *away from equilibrium*, moving to greater distances in finite time, and requiring infinite time to return. As a result, if the game has an interior equilibrium, $X(t)$ becomes infinitely spread out in the long run, exhibiting no concentration in *any* region of $\mathcal{X}$ other than, possibly, its boundary (if $\mathcal{X}$ is constrained).

2. In *strongly monotone games*, the dynamics drift *toward equilibrium*, and they end up being concentrated around the game's (necessarily unique) equilibrium. However, the players' learning trajectories continue to fluctuate at a distance which scales as $\mathcal{O}(\sigma_{\max})$ and, with probability 1, they return arbitrarily close to where they started, infinitely often.

These properties paint a sharp separation between null- and strongly monotone games, with uncertainty carrying drastically different consequences in each case; for an illustration, see Fig. 1.

The proof of [Theorems 1](#) and [2](#) is detailed in [Appendix D](#). From a technical standpoint, our analysis hinges on the use of the Fenchel coupling [(13)](#) as a "mean" energy function for the dynamics. In the null-monotone case, the hitting time estimates [(15)](#) rely on an application of Dynkin's formula [58, Chap. 7.4], coupled with an eigenvalue estimation for the growth of $F$. Then, by descending to a specific quotient of $\mathcal{Y}$ that compactifies the sublevel sets of $F$, we are able to leverage the fact that $\mathbb{E}_{x_0}[\tau_r] = \infty$ for $r < F_0$ to show that the dynamics are *not* positively recurrent—and hence, they *do not* admit an invariant distribution. The analysis for the strongly monotone case has the same starting point, but it then branches out almost immediately: the hitting time estimate [(17)](#) is again obtained via Dynkin's stopping time formula, but positive recurrence can no longer be established in $\mathcal{X}$, because the infinitesimal generator of $X(t)$ is not uniformly elliptic (that is, its eigenvalues are not bounded away from zero). Instead, we work directly with the infinetisimal generator of the score process $Y(t)$ whose generator *is* uniformly elliptic after taking a specific quotient in $\mathcal{Y}$. This allows us to deduce positive recurrence in $\mathcal{Y}$, which we then push forward to $\mathcal{X}$ via $Q$, and leverage the convergence of the occupation measures to the invariant distribution of the process to derive the concentration bound [(18)](#). We detail these steps in a series of technical lemmas in [Appendix D](#).

*Remark* 7. If the game is neither null- nor strongly monotone, our analysis suggests that [(S-FTRL)](#) would tend to "wander around" the null-monotone directions, and be carried along the strongly monotone directions toward the game's set of equilibria. However, obtaining a precise version of such a result is quite involved, so we defer it to future work. ☙

## 4 Learning under uncertainty in discrete time

We now turn to the discrete-time setting, which is of more direct algorithmic relevance. Compared to [Section 3](#), the analysis here is considerably more involved due to the lack of closed-form solutions and the limited applicability of diffusion-based methods. Nevertheless, as we shall see later in this section, the structural insights gained from the continuous-time analysis remain highly valuable as they form the foundation of the tools and techniques developed here.

**4.1. Learning in discrete time.** In discrete time, the most widely used implementation of the FTRL template unfolds for $t = 0, 1, \dots$ as

$$Y_{i,t+1} = Y_{i,t} + \gamma \hat{v}_{i,t} \qquad X_{i,t+1} = Q_i(Y_{i,t+1}). \qquad \text{(FTRL)}$$

In addition to the notions already introduced and discussed in [Section 3.2](#), (*i*) $\hat{v}_{i,t}$ denotes here a stochastic estimate of the player's payoff gradient vector at $X_{i,t}$; and (*ii*) $\gamma > 0$ is a *step-size* parameter, interchangeably referred to as the *learning rate* of the process. We discuss these two new elements below.

**The feedback process.** In terms of feedback, we assume that, at every round $t = 0, 1, \dots$, each player $i \in \mathcal{N}$ receives stochastic gradient feedback of the form

$$\hat{v}_{i,t} = \mathsf{V}_i(X_t; \omega_t) \quad \text{or, aggregating over all players} \quad \hat{v}_t = \mathsf{V}(X_t; \omega_t) \qquad (19)$$

where $\hat{v}_t = (\hat{v}_{i,t})_{i \in \mathcal{N}}$ and $\mathsf{V}(x; \omega) = (\mathsf{V}_i(x; \omega))_{i \in \mathcal{N}}$ is a *stochastic first-order oracle* for $v(x)$, viz.

$$\mathsf{V}(x; \omega) = v(x) + \mathsf{U}(x; \omega). \qquad \text{(SFO)}$$

In the above, $\omega_t$, $t = 0, 1, \dots$, is an i.i.d. sequence of random seeds drawn from some complete probability space $\Omega$, and $\mathsf{U}(x; \omega)$ is a random $\mathcal{Y}$-valued vector satisfying the standard assumptions

$$\mathbb{E}_\omega[\mathsf{U}(x; \omega)] = 0 \qquad \text{and} \qquad \mathbb{E}_\omega[\|\mathsf{U}(x; \omega)\|_*^2] \le \sigma^2 \qquad (20)$$

for some $\sigma > 0$. In this way, letting $\mathcal{F}_t$, $t = 0, 1, \dots$, denote the history of the process up to time $t$, and writing $U_t := \mathsf{U}(X_t; \omega_t)$ for the noise in the players' gradient feedback at time $t$, we get

$$\hat{v}_t = v(X_t) + U_t \quad \text{with} \quad \mathbb{E}[U_t \mid \mathcal{F}_t] = 0 \text{ and } \mathbb{E}[\|U_t\|_*^2 \mid \mathcal{F}_t] \le \sigma^2. \qquad (21)$$

Following standard practice in the field—see e.g., [68, 71] and references therein—we further assume that the probability distribution $\nu_x$ of $\mathsf{U}(x)$ decomposes as $\nu_x = \nu_x^c + \nu_x^\perp$ where: (*a*) $\nu_x^\perp$ is singular relative to the Lebesgue measure $\lambda_\mathcal{Y}$ on $\mathcal{Y}$; (*b*) $\nu_x^c$ is absolutely continuous relative to $\lambda_\mathcal{Y}$; and (*c*) the density $p_x(y)$ of $\nu_x^c$ is jointly continuous in $x$ and $y$, and it satisfies $\inf_{x \in \mathcal{K}} p_x(y) > 0$ for every compact set $\mathcal{K} \subseteq \mathcal{X}$ and all $y \in \mathcal{Y}$. This last assumption is relatively mild and ensures that the noise retains a non-degenerate, smooth component across $\mathcal{X}$, much like the assumption $\sigma_{\min} > 0$ for the diffusion matrix of [(S-FTRL)](#) in [Section 3](#).[4]

---

[4]This condition is trivially satisfied by most continuous error distributions in practice, and it can always be enforced by injecting a small uniform Gaussian noise component into the process, a technique which is widely

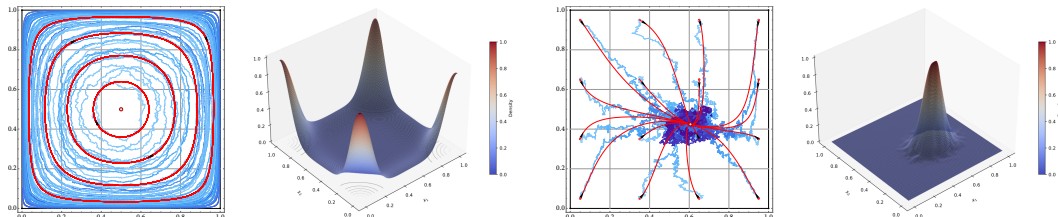

**Figure 1:** Trajectories and statistics of play under (FTRL) with entropic regularization in two min-max games over $\mathcal{X} = [0, 1]^2$, a bilinear and a quadratic one (left vs. right half respectively). Deterministic orbits are plotted in red and stochastic trajectories in shades of blue, with darker hues indicating later points in time; the density plots depict the resulting visitation frequency in $\mathcal{X}$. In tune with Theorems 3 and 4, we see that learning in null-monotone games drifts toward the *extremes* of $\mathcal{X}$; by contrast, in strongly monotone games, learning orbits drift toward equilibrium, but continue to fluctuate around it. More details are provided in Appendix F.

**The algorithm's learning rate.** The second feature which sets the discrete-time framework apart is the method's learning rate $\gamma$. Here and throughout, we consider a *constant* learning rate schedule; this should be contrasted to the stochastic approximation literature [6, 9, 37, 50, 54], where (FTRL) is run with a *vanishing* step-size $\gamma_t \to 0$, typically satisfying some form of the Robbins–Monro summability conditions $\sum_t \gamma_t = \infty$, $\sum_t \gamma_t^2 < \infty$.

In many cases, the use of a vanishing step-size enables convergence of the algorithm because it dampens the impact of the noise over time [50]; at the same time however, in many applied settings, algorithms are implemented with a constant—or, at the very least, non-vanishing—step-size. This choice is largely driven by practical considerations: constant step-size schedules are easier to calibrate and maintain, particularly in large-scale systems where adaptivity and simplicity are critical. Moreover, vanishing step-size schedules often exhibit prolonged transient phases and converge slowly toward equilibrium neighborhoods; by contrast, constant step-size methods tend to reach near-stationary regions much faster, even within 0.1% accuracy or lower [16]. This behavior underlies their widespread use in modern machine learning pipelines, where learning rates are kept effectively constant throughout training, even for models trained over billions of samples and/or hundreds of billions of tokens [15].

**4.2. Analysis and results.** We now have the necessary machinery in place to present our results for (FTRL). Before doing so, we should only stress that the discrete-time analysis is, by necessity, more qualitative than the more explicit, continuous-time results presented in Section 3. This gap is difficult to avoid: in continuous time, the rules of stochastic calculus comprise a very sharp set of tools with which to obtain closed-form estimates for the processes involved; on the other hand, in discrete time, even the most basic tools of stochastic analysis—like Dynkin's formula—are dulled down because of measurability and subsampling issues.

As before, we split our focus between null- and strongly monotone games.

**The null-monotone regime.** A key take-away from the analysis of Section 3 is that, in null-monotone games, uncertainty causes the dynamics of regularized learning to spread out, diverging to infinity on average, without concentrating at any region of $\mathcal{X}$ other than its boundary. Our first result below shows that a version of this tenet continues to hold in discrete time:

**Theorem 3** (Null-monotone games)**.** *Suppose that* (FTRL) *is run in a null-monotone game* $\mathcal{G}$*, and let* $x^*$ *be an equilibrium of* $\mathcal{G}$*. Suppose further that* $h^*$ *is strongly convex, and let* $F_t = F(x^*, Y_t)$*, where* $F$ *is the induced Fenchel coupling* (B.22)*. Then* $\lim_{t \to \infty} \mathbb{E}[F_t] = \infty$*.*

This result shows that (FTRL) drifts away to infinity on average—though, of course, as in the continuous-time case, this does not mean that this occurs with probability 1. What is missing from Theorem 3 relative to Theorem 1 is a bound on the mean time required for $F_t$ to increase or decrease by $\varepsilon$. In the absence of a *consistent* drift component, our continuous-time estimates were only made possible through the use of stochastic calculus. In discrete time however, $X_t$ evolves in *discrete*, driftless jumps, introducing overshoots and upcrossings that render this question significantly harder.

---

used in both optimization and reinforcement learning to promote sufficient exploration and avoid degeneracy issues and saddle-points [6, 22, 39, 70]. In such cases, the density of the absolutely continuous component is strictly positive everywhere and independent of $x \in \mathcal{X}$, so the uniform lower bound condition holds trivially.

We conjecture that similar bounds do hold in discrete time, but we leave this open as a conjecture. [We only note here for completeness that a similar result holds for any decaying step-size sequence $\gamma_t$ with $\sum_t \gamma_t^2 = \infty$.]

**The strongly monotone regime.** We now turn to the long-run behavior of (FTRL) in strongly monotone games. Based in no small part on the continuous-time analysis of the previous section, our goal will be to understand the distributional properties of the dynamics, with a particular focus on (*a*) the existence and uniqueness of an invariant measure; and (*b*) the extent to which this measure is concentrated around the game's equilibrium—which, in turn, quantifies the long-run proximity of the iterates of (FTRL) to equilibrium. With all this in mind, our results can be stated as follows:

**Theorem 4** (Strongly monotone games). *Suppose that* (FTRL) *is run in an $\alpha$-strongly monotone game $\mathcal{G}$, and consider the hitting time*

$$\tau_r := \inf\{t > 0 : X(t) \in \mathbb{B}_r(x^*)\} \tag{22}$$

*where $\mathbb{B}_r(x^*) = \{x : \|x - x^*\| \le r\}$ is a ball of radius $r$ centered on the* (*necessarily unique*) *equilibrium $x^*$ of $\mathcal{G}$. Then, for all $r > r_\sigma := \sqrt{\gamma(\sigma^2 + \beta^2)/(\alpha K)}$, we have*

$$\mathbb{E}[\tau_r] \le \frac{1}{\alpha\gamma(r^2 - r_\sigma^2)} \times \begin{cases} F_0 & \text{if } X_0 \notin \mathbb{B}_r(x^*), \\ F_0 + \alpha\gamma r^2 & \text{if } X_0 \in \mathbb{B}_r(x^*), \end{cases} \tag{23}$$

*where $F_0 = F(x^*, Y_0)$. If, in addition, $x^*$ is interior, $X_t$ admits a unique invariant distribution to which it converges in total variation, and we have*

$$\lim_{t\to\infty} \frac{1}{t} \mathbb{E}\left[ \sum_{s=0}^{t} \mathbb{1}\{X_t \in \mathbb{B}_r(x^*)\} \right] \ge 1 - r_\sigma^2/r^2 \tag{24}$$

*for all $r > r_\sigma$ such that $\mathbb{B}_r(x^*) \subseteq \operatorname{ri} \mathcal{X}$.*

*Remark.* Unlike the continuous-time setting of Section 3, we must treat the cases $X_0 \in \mathbb{B}_r(x^*)$ and $X_0 \notin \mathbb{B}_r(x^*)$ separately. This distinction arises only in discrete time, because the iterates may exhibit large jumps—so, returning to $\mathbb{B}_r(x^*)$ is not guaranteed, even if the process is initialized within. ☕

We prove Theorem 4 in Appendix E following the strategy outlined below. First, shadowing the continuous-time analysis of Section 3, we reduce the dynamics to a suitable quotient space of $\mathcal{Y}$, eliminating redundant directions and ensuring that the process evolves in a minimal, non-degenerate domain. Building on this, we then show that the induced dynamics are Lebesgue-irreducible, i.e., every measurable set with positive Lebesgue measure is reachable with positive probability under the transition kernel of the process. Moreover, invoking (23), we further deduce that $\mathbb{P}(\tau_r < \infty) = 1$ for any initial condition, implying that $\mathbb{B}_r(x^*)$ is visited infinitely often. Finally, we also show that $\mathbb{B}_r(x^*)$ satisfies a minorization condition, meaning that the transition kernel from any point in the ball dominates a fixed reference measure. In turn, this implies that, upon returning to $\mathbb{B}_r(x^*)$, the process has a nonzero chance of "forgetting" its past, allowing us to construct a regeneration structure via a coupling argument. Then, leveraging the continuity of $F(x^*, y)$, we obtain a uniform bound on the expected return times $\mathbb{E}[\tau_r]$ over any initialization in $\mathbb{B}_r(x^*)$, which allows us to conclude that the process $Y_t$ is positive Harris recurrent. As a result, it can be shown that the iterates of (FTRL) converge to a unique invariant measure, and we obtain quantitative control over their long-run concentration by means of our previous estimates.

## 5  Concluding remarks

Our aim in this paper was to quantify the impact of noise and uncertainty on the dynamics of multi-agent regularized learning. Our findings reveal a sharp separation between games that are *null-monotone* (like bilinear min-max games), and *strongly monotone* games (like Kelly auctions or Cournot competitions). In the former case, the quasi-periodic profile of the deterministic dynamics is destroyed, and learning under uncertainty drifts away on average toward extreme points (or escapes to infinity); in the latter, the sharp convergence guarantees of the deterministic dynamics are diluted by noise, and the resulting dynamics end up concentrated in a region around the game's equilibrium (which we estimate). This paves the way for further explorations of the long-run statistics of regularized learning in games—especially pertaining to the invariant measure of the process—a topic which we find particularly promising for advancing our understanding of the field.

## Acknowledgments and Disclosure of Funding

Jose Blanchet gratefully acknowledges support from the Department of Defense through the Air Force Office of Scientific Research (Award FA9550-20-1-0397) and the Office of Naval Research (Grant 1398311), as well as from the National Science Foundation (Grants 2229012, 2312204, and 2403007). Panayotis Mertikopoulos is also a member of Archimedes/Athena RC and acknowledges financial support by the French National Research Agency (ANR) in the framework of the PEPR IA FOUNDRY project (ANR-23-PEIA-0003), the project IRGA-SPICE (G7H-IRG24E90), and project MIS 5154714 of the National Recovery and Resilience Plan Greece 2.0, funded by the European Union under the NextGenerationEU Program.

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

# A  Examples

In this appendix, we present several examples of games satisfying the standing assumptions we outlined in Section 2. Overall, these assumptions are quite standard in the study of online learning and games with continuous action spaces, and most of the positive results in the literature hinge on precisely these assumptions or close variants thereof.

To provide some context, the monotonicity assumption (cf. Definition 1) provides an amenable "convex structure", which is essential for establishing the existence and characterization of equilibria via first-order variational condtions. Without a structural characteriztion of this type, even defining a meaningful solution concept becomes unclear, and global convergence cannot be expected—at least in general. In a sense, these assumptions parallel the convex/non-convex separation in optimization—but with the added challenge that, in non-convex games, equilibria may fail to exist altogether, unlike minimizers (either local or global) in non-convex minimization problems.

Our regularity assumptions—closed action sets, Lipschitz smoothness, etc.—are largely technical, standard in practice and, as such, nearly universal in the literature. They could be relaxed, for instance, by assuming local or relative smoothness, Hölder continuity, or something of the sort—though the resulting analysis would be considerably more involved. Relaxing monotonicity, however, is considerably trickier: some of our results would go through as long as the game's equilibrium admits a global variational characterization, e.g., in the spirit of variational stability or a Minty-type condition, cf. [50, 53, 73] and references therein. If, however, the game admits distinct components of equilibria, all bets are off: in that case, FTRL could transit in perpetuity between the game's different equilibrium components, and characterizing the mean sojourn and transition times of the process would only be possible in very special cases.

All in all, the set of assumptions that we consider represents a certain "sweet spot" between theoretical tractability and practical relevance, which explains their prevalence in the literature. The examples below illustrate the range of settings where these assumptions arise naturally.

**Example A.1** (Zero-sum bimatrix games). A bimatrix game consists of two players, each with a finite set of actions $\mathcal{A}_i$, $i = 1, 2$, and a min-max objective function $L\colon \mathcal{A}_1 \times \mathcal{A}_2 \to \mathbb{R}$, typically encoded in a matrix $M \in \mathbb{R}^{\mathcal{A}_1 \times \mathcal{A}_2}$ with $M_{\alpha\beta} = L(\alpha, \beta)$ for all $\alpha \in \mathcal{A}_1$, $\beta \in \mathcal{A}_2$. The first player is cast in the role of the minimizer and the second player in that of the maximizer, so their corresponding payoff functions are defined as $u_1 = -L = -u_2$.

In the mixed extension of the game, each player can mix their actions by selecting a probability distribution—a *mixed strategy*—over $\mathcal{A}_i$, that is, an element $x_i$ of the probability simplex $\mathcal{X}_i \equiv \Delta(\mathcal{A}_i) = \{x_i \in \mathbb{R}_+^{\mathcal{A}_i} : \|x_i\|_1 = 1\}$. Accordingly, in matrix notation, the players' corresponding mixed payoffs are given by

$$u_1(x_1, x_2) = -x_1^\top M x_2 = -u_2(x_1, x_2) \tag{A.1}$$

so their individual gradient fields can be expressed as

$$v_1(x_1, x_2) = -M x_2 \quad \text{and} \quad v_2(x_1, x_2) = M^\top x_1 \tag{A.2}$$

for all $x_1 \in \mathcal{X}_1$ and all $x_2 \in \mathcal{X}_2$.

By definition, a mixed-strategy Nash equilibrium of a bimatrix zero-sum game satisfies

$$L(x_1^*, x_2) \le L(x_1^*, x_2^*) \le L(x_1, x_2^*) \quad \text{for all } x_1 \in \mathcal{X}_1, x_2 \in \mathcal{X}_2. \tag{A.3}$$

If, in addition, $x_1^*, x_2^*$ both have full support—that is, $x_1^* \in \mathrm{ri}\,\mathcal{X}_1$ and $x_2^* \in \mathrm{ri}\,\mathcal{X}_2$—we also have the "equalizing payoffs" condition

$$L(x_1^*, x_2) = L(x_1, x_2^*) \quad \text{for all } x_1 \in \mathcal{X}_1, x_2 \in \mathcal{X}_2 \tag{A.4}$$

which means that (A.3) binds identically. In this case, we readily get

$$\langle v_1(x_1, x_2), x_1 - x_1^* \rangle + \langle v_2(x_1, x_2), x_2 - x_2^* \rangle$$
$$= u_1(x_1, x_2) - u_1(x_1^*, x_2) + u_2(x_1, x_2) - u_2(x_1, x_2^*) = 0 \tag{A.5}$$

for all $x_1 \in \mathcal{X}_1$, $x_2 \in \mathcal{X}_2$, i.e., the game is null-monotone in the sense of Definition 1. ☙

**Example A.2** (Cournot competition). In the standard Cournot competition model, there is a finite set of *firms*, indexed by $i \in \mathcal{N} = \{1, \ldots, N\}$, each providing the market with a quantity $x_i \in [0, B_i]$ of some good (or service) up to the firm's production budget $B_i$. Following the law of supply and demand, this good is priced following the simple linear model $P(x) = a - b \sum_i x_i$, i.e., as a linearly decreasing function of the total supply. Accordingly, in this model, the utility of firm $i$ is given by

$$u_i(x) = x_i P(x) - c_i x_i = \left[ a - b \sum_{j \in \mathcal{N}} x_j - c_i \right] x_i, \tag{A.6}$$

where $c_i$ represents the marginal production cost of firm $i$.

By a straightforward derivation, the players' individual payoff gradients are given by

$$v_i(x) = \frac{\partial u_i}{\partial x_i} = \left[ a - b \sum_{j \in \mathcal{N}} x_j - c_i \right] - b x_i \tag{A.7}$$

and hence, the Hessian matrix of the game will be

$$H_{ij}(x) := \frac{1}{2} \frac{\partial^2 u_i}{\partial x_j \partial x_i} + \frac{1}{2} \frac{\partial^2 u_j}{\partial x_i \partial x_j} = -b - b\delta_{ij} \tag{A.8}$$

where $\delta_{ij}$ is the standard Kronecker delta. Since $H$ is circulant, standard linear algebra considerations show that its eigenvalues are $-b$ and $-(N+1)b$ (with multiplicity $N-1$ and $1$ respectively), so it follows by a well-known second-order criterion that the Cournot competition game is $b$-strongly monotone [50, 62]. ❧

**Example A.3** (Signal covariance optimization). Consider a vector Gaussian channel of the form

$$\mathbf{y} = \sum_{i \in \mathcal{N}} \mathbf{H}_i \mathbf{x}_i + \mathbf{z} \tag{A.9}$$

where $\mathbf{x}_i \in \mathbb{C}^{m_i}$ is the (complex-valued) signal transmitted by the $i$-th user of the channel, $\mathbf{H} \in \mathbb{C}^{n \times m_i}$ is the transfer matrix of the channel, $\mathbf{z} \in \mathbb{C}^n$ is the noise in the channel (assumed zero-mean Gaussian and, without loss of generality, with unit covariance), and $\mathbf{y} \in \mathbb{C}^n$ is the aggregate signal output of the channel [72]. In this context, each user $i \in \mathcal{N}$ controls the covariance matrix $\mathbf{X}_i = \mathbb{E}[\mathbf{x}_i \mathbf{x}_i^\dagger]$ subject to the power constraint $\mathrm{tr}(\mathbf{X}_i) = \mathbb{E}[\|\mathbf{x}_i\|^2] \leq P_i$, where $P_i$ denotes the user's maximum transmit power. In this case, by the celebrated Shannon–Telatar formula [66], and assuming a single-user decoding scheme at the receiver, the achievable rate of the $i$-th user is

$$u_i(\mathbf{X}_i; \mathbf{X}_{-i}) = \log \det\left( \mathbf{I} + \sum_j \mathbf{H}_j \mathbf{X}_j \mathbf{H}_j^\dagger \right) - \log \det\left( \mathbf{I} + \sum_{j \neq i} \mathbf{H}_j \mathbf{X}_j \mathbf{H}_j^\dagger \right). \tag{A.10}$$

Putting everything together, this defines a continuous game with players $i \in \mathcal{N} = \{1, \ldots, N\}$, spectrahedral action sets of the form

$$\mathcal{Q}_i = \{ \mathbf{X}_i \in \mathbb{C}^{m_i \times m_i} : \mathbf{X}_i \succcurlyeq 0 \text{ and } \mathrm{tr} \, \mathbf{X}_i \leq P_i \} \tag{A.11}$$

for all $i \in \mathcal{N}$, and payoff functions given by (A.10). By a calculation of Belmega et al. [5], it is known that this game is concave and monotone—and, in fact, *strongly monotone* if the linear mapping $(\mathbf{X}_1, \ldots, \mathbf{X}_N) \mapsto \sum_i \mathbf{H}_i \mathbf{X}_i \mathbf{H}_i^\dagger$ is not rank-deficient. ❧

Examples that are closer to signal processing and data science include distributed metric learning, multimedia classification, etc. For a range of applications along these lines, we refer the reader to [51, 63] and references therein.

# B  Mirror maps and regularization

In this appendix, we collect some background material, properties and examples regarding the regularization machinery underlying (FTRL) and (S-FTRL). To lighten notation—especially with respect to the player index $i \in \mathcal{N}$—we base everything in this appendix on an abstract closed convex subset of some $d$-dimensional vector space, which could either be $\mathcal{X}_i$ or $\mathcal{X}$, depending on the context.

The results presented below (or a version thereof) are known in the literature; nevertheless, we provide detailed proofs for completeness and to resolve any conflicts or ambiguities with different conventions in the literature.

**B.1. Preliminaries.** Let $\mathcal{V}$ be a $d$-dimensional normed space, let $\mathcal{Y} := \mathcal{V}^*$ denote the (algebraic) dual of $\mathcal{V}$, and let $\langle y, x \rangle$ denote the canonical bilinear pairing between $x \in \mathcal{V}$ and $y \in \mathcal{V}^*$. If $\|\cdot\|$ is a norm on $\mathcal{V}$ will also write

$$\|y\|_* = \max\{\langle y, x \rangle : \|x\| \leq 1\} \tag{B.1}$$

for the induced dual norm on $\mathcal{Y}$, so $|\langle y, x \rangle| \leq \|x\| \|y\|_*$ for all $x \in \mathcal{V}$ and all $y \in \mathcal{Y}$ by construction.

Given a closed convex subset $\mathcal{C}$ of $\mathcal{V}$, we also define:

1. The *tangent cone* to $\mathcal{C}$ at $p \in \mathcal{C}$ as

$$\mathrm{TC}(p) = \mathrm{cl}\{z \in \mathcal{V} : p + tz \in \mathcal{C} \text{ for some } t > 0\} \tag{B.2}$$

   i.e., as the closure of the set of rays emanating from $p$ and meeting $\mathcal{C}$ in at least one other point.
2. The *dual cone* to $\mathcal{C}$ at $p \in \mathcal{C}$ as

$$\mathrm{TC}^*(p) = \{w \in \mathcal{Y} : \langle w, z \rangle \geq 0 \text{ for all } z \in \mathrm{TC}(p)\} \tag{B.3}$$

3. The *polar cone* to $\mathcal{C}$ at $p \in \mathcal{C}$ as

$$\mathrm{PC}(p) = \{w \in \mathcal{Y} : \langle w, z \rangle \leq 0 \text{ for all } z \in \mathrm{TC}(p)\} \tag{B.4}$$

Following standard conventions in the field [60], convex functions will be allowed to take values in the extended real line $\mathbb{R} \cup \{\infty\}$, and we will denote the *effective domain* of a convex function $f : \mathcal{V} \to \mathbb{R} \cup \{\infty\}$ as

$$\mathrm{dom}\, f := \{x \in \mathcal{V} : f(x) < \infty\}. \tag{B.5}$$

When there is no danger of confusion, we will identify a convex function $f : \mathcal{V} \to \mathbb{R}$ with its restriction on $\mathrm{dom}\, f$; in other words, we will treat $f$ interchangeably as a function on $\mathrm{dom}\, f$ with values in $\mathbb{R}$, or as a function on $\mathcal{V}$ with values in $\mathbb{R} \cup \{\infty\}$ (and finite on $\mathrm{dom}\, f$).

Throughout the sequel, we will assume that all functions under study are *proper*, that is, $\mathrm{dom}\, f \neq \varnothing$. Then, given a proper function $f : \mathcal{V} \to \mathbb{R} \cup \{\infty\}$, the *subdifferential* of $f$ at $x \in \mathrm{dom}\, f$ is defined as

$$\partial f(x) := \{y \in \mathcal{Y} : f(x') \geq f(x) + \langle y, x' - x \rangle \text{ for all } x' \in \mathcal{V}\} \tag{B.6}$$

and we denote the *domain of subdifferentiability* of $f$ as

$$\mathrm{dom}\, \partial f = \{x \in \mathcal{V} : \partial f(x) \neq \varnothing\}. \tag{B.7}$$

With all this in hand, a *regularizer* on a closed convex subset $\mathcal{C}$ of $\mathcal{V}$ is a continuous function $h : \mathcal{C} \to \mathbb{R}$ which is *strongly convex*, i.e., there exists some $K > 0$ such that

$$h(\lambda x + (1 - \lambda)x') \leq t h(x) + (1 - \lambda)h(x') - \frac{K}{2}\lambda(1 - \lambda)\|x' - x\|^2 \tag{B.8}$$

for all $x, x' \in \mathcal{C}$ and for all $\lambda \in [0, 1]$. By standard arguments [7, 61], this immediately implies that

$$h(x') \geq h(x) + \partial h(x; x' - x) + \frac{K}{2}\|x' - x\|^2 \quad \text{for all } x, x' \in \mathcal{X}, \tag{B.9}$$

where

$$\partial h(x; x' - x) = \lim_{\theta \to 0^+} [h(x + \theta(x' - x)) - h(x)]/\theta \tag{B.10}$$

denotes the one-sided directional derivative of $h$ at $x$ along the direction of $x' - x$. In addition, we also define the following objects associated to $h$:

1. The *prox-domain* of $h$:

$$\mathcal{C}_h := \mathrm{dom}\, \partial h \tag{B.11}$$

2. The *mirror map* $Q : \mathcal{Y} \to \mathcal{X}$ induced by $h$:

$$Q(y) := \arg\max_{x \in \mathcal{X}}\{\langle y, x \rangle - h(x)\} \qquad \text{for all } y \in \mathcal{Y}. \tag{B.12}$$

3. The *convex conjugate* $h^* : \mathcal{Y} \to \mathbb{R}$ of $h$:

$$h^*(y) := \max_{x \in \mathcal{X}}\{\langle y, x \rangle - h(x)\} \qquad \text{for all } y \in \mathcal{Y}. \tag{B.13}$$

The proposition below provides some basic properties linking all the above:

**Proposition B.1.** *Let h be a K-strongly convex regularizer on C. Then:*

(a) *Q is single-valued on $\mathcal{Y}$.*

(b) *For all $x \in \mathcal{C}_h$ and all $y \in \mathcal{Y}$, we have*

$$x = Q(y) \quad \text{if and only if} \quad y \in \partial h(x). \tag{B.14}$$

(c) *The image $\operatorname{im} Q$ of $Q$ is equal to the prox-domain of h, and we have*

$$\operatorname{ri} \mathcal{C} \subseteq \operatorname{im} Q = \mathcal{C}_h \subseteq \mathcal{C}. \tag{B.15}$$

(d) *The convex conjugate $h^* \colon \mathcal{Y} \to \mathbb{R}$ of h is differentiable and satisfies*

$$Q(y) = \nabla h^*(y) \quad \text{for all } y \in \mathcal{Y}. \tag{B.16}$$

(e) *Q is $(1/K)$-Lipschitz continuous, that is,*

$$\|Q(y') - Q(y)\| \le (1/K)\|y' - y\|_* \quad \text{for all } y, y' \in \mathcal{Y}. \tag{B.17}$$

(f) *Fix some $y \in \mathcal{Y}$ and let $x = Q(y)$. Then, for all $x' \in \mathcal{X}$ we have:*

$$\partial h(x; x' - x) \ge \langle y, x' - x \rangle. \tag{B.18}$$

(g) *Fix some $y \in \mathcal{Y}$, and let $x = Q(y)$. Then $Q(y + w) = x$ for all $w \in \mathrm{PC}(x)$.*

*Proof.* For the most part, these properties are well known in the literature (except possibly the last one), so we only provide a pointer or a short sketch for most of them.

(a) This readily follows from the fact that $h$ is strongly convex, so the arg max in (B.12) is attained and is unique for all $y \in \mathcal{Y}$.

(b) By Fermat's rule [60, Chap. 26], we readily see that $x$ solves (B.12) if and only if $y - \partial h(x) \ni 0$, that is, if and only if $y \in \partial h(x)$. Since this implies that $\partial h$, our claim follows.

(c) By (B.14), we readily get $\operatorname{im} Q = \mathcal{C}_h$. As for the second part of our claim, it follows from basic properties of the subdifferential, cf. Rockafellar [60, Chap. 26].

(d) This is simply Danskin's theorem, see e.g., Bertsekas [7, Proposition 5.4.8, Appendix B].

(e) This is a consequence of the fact that $h^*$ is $(1/K)$-Lipschitz smooth, cf. Rockafellar & Wets [61, Theorem 12.60(b)].

(f) Since $y \in \partial h(x)$ by (B.14), we readily get that

$$h(x + \theta(x' - x)) \ge h(x) + \theta \langle y, x' - x \rangle \quad \text{for all } \theta \in [0, 1]. \tag{B.19}$$

Hence, by rearranging and taking the limit $\theta \to 0^+$, we conclude that

$$\partial h(x; x' - x) = \lim_{\theta \to 0^+} \frac{h(x + \theta(x' - x)) - h(x)}{\theta} \ge \langle y, x' - x \rangle \tag{B.20}$$

as claimed.[5]

(g) By (B.14) it suffices to show that $y + w \in \partial h(x)$ for all $w \in \mathrm{PC}(x)$. However, if $w \in \mathrm{PC}(x)$, we also have $\langle w, x' - x \rangle \le 0$ for all $x' \in \mathcal{X}$, and hence, with $y \in \partial h(x)$, we readily get

$$h(x') \ge h(x) + \langle y, x' - x \rangle$$
$$\ge h(x) + \langle y + w, x' - x \rangle \quad \text{for all } x' \in \mathcal{X}. \tag{B.21}$$

This shows that $y + w \in \partial h(x)$ and completes our proof. $\blacksquare$

Following [45, 50], we also define the *Fenchel coupling* associated to $h$ as

$$F(p, y) = h(p) + h^*(y) - \langle y, p \rangle \quad \text{for all } p \in \mathcal{X}, y \in \mathcal{Y}. \tag{B.22}$$

The next proposition shows that the Fenchel coupling can be seen as a "primal-dual" measure of divergence between $p \in \mathcal{C}$ and $y \in \mathcal{Y}$:

---

[5]The existence of the limit is guaranteed by elementary convex analysis arguments, cf. Bertsekas [7, App. B].

**Proposition B.2.** *Let $h$ be a $K$-strongly convex regularizer on $\mathcal{C}$. Then, for all $p \in \mathcal{X}$ and all $y \in \mathcal{Y}$, we have:*

$(a)$ $F(p, y) \geq 0$ *with equality if and only if* $p = Q(y)$. $\hspace{2cm}$ (B.23a)

$(b)$ $F(p, y) \geq \frac{1}{2} K \, \|Q(y) - p\|^2$. $\hspace{3.5cm}$ (B.23b)

*Proof.* These properties are known in the literature, but we provide a quick proof for completeness.

$(a)$ By the Fenchel–Young inequality, we have $h(p) + h^*(y) \geq \langle y, p \rangle$ for all $p \in \mathcal{X}$, $y \in \mathcal{Y}$, with equality if and only if $y \in \partial h(p)$. Our claim then follows from (B.14).

$(b)$ Let $x = Q(y)$ so $y \in \partial h(x)$ by (B.14). Then, by the definition of $F$, we have

$$
\begin{aligned}
F(p, y) &= h(p) + h^*(y) - \langle y, p \rangle \\
&= h(p) + \langle y, x \rangle - h(x) - \langle y, p \rangle && \text{\# since } y \in \partial h(x) \\
&\geq h(p) - h(x) - \partial h(x; p - x) && \text{\# by Proposition B.1} \\
&\geq \tfrac{1}{2} K \|x - p\|^2 && \text{\# by (B.8)}
\end{aligned}
$$

so our proof is complete. $\hspace{10cm}$ ∎

Our last result at this point is a useful differentiation formula for the Fenchel coupling:

**Lemma B.1.** *For all $p \in \mathcal{X}$ and all $y \in \mathcal{Y}$, we have:*

$$
\nabla_y F(p, y) = Q(y) - p. \tag{B.24}
$$

*Proof.* The proof follows immediately from Danskin's theorem, cf. Eq. (B.16) of Proposition B.1. $\hspace{0.3cm}$ ∎

**B.2. Update lemmas.** Moving forward, we note that the basic update step of (FTRL) can be written as

$$
y^+ = y + w \quad \text{and} \quad x^+ = Q(y^+) \tag{B.25}
$$

for some $y, w \in \mathcal{Y}$. With this in mind, we state below a series of identities and estimates for the Fenchel coupling before and after an update of the form (B.25).

The first is a primal-dual version of the so-called "three-point identity" for Bregman functions [14]:

**Lemma B.2.** *Fix some $p \in \mathcal{X}$, $y \in \mathcal{Y}$, and let $x = Q(y)$. Then, for all $y^+ \in \mathcal{Y}$, we have:*

$$
F(p, y^+) = F(p, y) + F(x, y^+) + \langle y^+ - y, x - p \rangle. \tag{B.26}
$$

*Proof.* By definition, we have:

$$
\begin{aligned}
F(p, y^+) &= h(p) + h^*(y^+) - \langle y^+, p \rangle \tag{B.27a} \\
F(p, y) &= h(p) + h^*(y) - \langle y, p \rangle \tag{B.27b} \\
F(x, y^+) &= h(x) + h^*(y^+) - \langle y^+, x \rangle \tag{B.27c}
\end{aligned}
$$

Thus, subtracting (B.27b) and (B.27c) from (B.27a), and rearranging, we get

$$
F(p, y^+) = F(p, y) + F(x, y^+) - h(x) - h^*(y) + \langle y^+, x \rangle - \langle y^+ - y, p \rangle. \tag{B.28}
$$

Our assertion then follows by recalling that $x = Q(y)$, so $h(x) + h^*(y) = \langle y, x \rangle$. $\hspace{1.5cm}$ ∎

The next result we present concerns the Fenchel coupling before and after a direct update step; similar results exist in the literature, but we again provide a proof for completeness.

**Lemma B.3.** *Fix some $p \in \mathcal{X}$ and $y, w \in \mathcal{Y}$. Then, letting $x = Q(y)$, $y^+ = y + w$, and $x^+ = Q(y^+)$ as per (B.25), we have:*

$$
F(p, y^+) = F(p, y) + \langle w, x^+ - p \rangle - F(x^+, y) \tag{B.29a}
$$

$$
\leq F(p, x) + \langle w, x - p \rangle + \frac{1}{2K} \|w\|_*^2. \tag{B.29b}
$$

*Proof.* By the three-point identity (B.26), we have

$$F(x, y) = F(x, y^+) + F(x^+, x) + \langle y - y^+, x^+ - p \rangle \tag{B.30}$$

so our first claim is immediate. For our second claim, rearranging terms and employing the Fenchel–Young inequality gives

$$\begin{aligned}
F(p, y) &+ \langle w, x^+ - p \rangle - F(x^+, y) \\
&= F(p, y) + \langle w, x - p \rangle + \langle w, x^+ - x \rangle - F(p, y) \\
&\leq F(p, y) + \langle w, x - p \rangle + \frac{1}{2K} \|w\|_*^2 + \frac{K}{2} \|x - p\|^2 - F(p, y)
\end{aligned} \tag{B.31}$$

so our claim follows from Proposition B.2. ∎

## C  A short primer on stochastic analysis

In this appendix, we collect some standard results from stochastic analysis in order to provide a degree of self-completeness to the main text. For an introduction to stochastic analysis and the theory of SDEs, we refer the reader to the masterful accounts of Øksendal [58] and Kuo [36].

The main focus of the theory is the study of ordinary differential equations (ODEs) perturbed by noise, modeled informally after the *Langevin equation*

$$\frac{dZ}{dt} = b(Z(t)) + \eta(t) \tag{LE}$$

where $Z(t)$ is a stochastic process in $\mathbb{R}$, $b \colon \mathbb{R} \to \mathbb{R}$ is the *drift* of the process, and $\eta(t)$ is the "noise" perturbing the deterministic ODE $\dot{z} = b(z)$. Unfortunately, albeit natural, the problem with (LE) is that any reasonable continuous-time model of noise would lead to trajectories that are almost nowhere differentiable, so the meaning of "$dZ/dt$" in (LE) is rather precarious.[6]

In lieu of this, to give formal meaning to (LE), we consider instead the *stochastic differential equation*

$$dZ(t) = b(Z(t)) \, dt + \sigma(Z(t)) \, dW(t) \tag{SDE}$$

which is shorthand for the integral equation

$$Z(t) = \int_0^t b(Z(s)) \, ds + \int_0^t \sigma(Z(s)) \, dW(s). \tag{C.1}$$

for some state-dependent *diffusion coefficient* $\sigma \colon \mathbb{R} \to \mathbb{R}$. The key element in the above formulation is the so-called *Itô integral* that appears in the right-hand side of (SDE), and which is defined relative to what is known as a standard *Brownian motion* on $\mathbb{R}$. Intuitively, what this means is that the integral $\int_0^t \sigma(Z(s)) \, dW(s)$ is obtained in the limit $\delta t = t_{k+1} - t_k \to 0$ of the discrete-time approximation

$$\int_0^t \sigma(Z(s)) \, dW(s) \approx \sum_{k=1}^{\lceil t/\delta t \rceil} \sigma(Z(t_k)) \left[ W(t_{k+1}) - W(t_k) \right] \tag{C.2}$$

where $W(t)$ is some stochastic process that satisfies what one would expect from a "white noise" process (zero-mean, with independent increments), but is still "regular enough" to possess a reasonable behavior in the limit $\delta t \to 0$. These considerations lead to the formal definition of a Brownian motion—or, more precisely, the *Wiener process*—which is characterized by the following properties:

1. The increments of $W$ are *independent*, that is, for all $t, \tau > 0$, the future increments $W(t+\tau) - W(t)$ of $W$ are independent of its past values $W(s)$, $s < t$.
2. The increments of $W$ are *Gaussian*, that is, for all $t, \tau > 0$, the future increments $W(t + \tau) - W(t)$ of $W$ are normally distributed with mean 0 and variance $\tau$, i.e., $W(t + \tau) - W(t) \sim \mathcal{N}(0, \tau)$.
3. The sample paths of $W$ are *continuous* (a.s.), i.e., $W(t)$ is a continuous function of $t$ for almost every realization of $W$.

---

[6]In particular, consider a noise process $\eta(t)$ which is *a) zero-mean:* $\mathbb{E}[\eta(t)] = 0$; *b) uncorrelated:* $\mathbb{E}[\eta(t_1)\eta(t_2)] = 0$ if $t_2 \neq t_1$; and *c) stationary*, in the sense that $\eta(t + s)$ and $\eta(t)$ are identically distributed for all $s > 0$. Then, *any* such process does not have continuous paths [58, p. 21].

The existence of a process with the above properties is by no means a trivial affair, but it can constructed e.g., as the scaling limit of a random walk, or some other discrete-time stochastic processes with stationary independent increments.

Providing a more detailed account of the definition of $W(t)$ and the associated stochastic integral which appears in (SDE) is well beyond the scope of our paper; for an accessible introduction, we refer the reader to Øksendal [58, Chap. 2]. What is more important for our purposes is that, albeit non-differentiable, the solution $Z(t)$ still satisfies a certain version of the chain rule, known as *Itô's formula* [29]. Specifically, for any $C^2$ function $f \colon \mathbb{R} \to \mathbb{R}$, we have

$$df(Z(t)) = f'(Z(t))b(Z(t))\, dt + \tfrac{1}{2}f''(Z(t))\sigma^2(t)\, dt + f'(Z(t))\, \sigma(Z(t))\, dW(t) \qquad \text{(C.3)}$$

or, more compactly:

$$df(Z(t)) = f'(Z(t))\, dZ(t) + \tfrac{1}{2}f''(Z(t))\, dZ(t) \cdot dZ(t) \qquad \text{(C.4)}$$

where the product $dZ \cdot dZ$ is computed according to the rules of stochastic calculus [58]:

$$dt \cdot dt = 0 \quad dt \cdot dW(t) = 0 \quad \text{and} \quad dW(t) \cdot dW(t) = dt\,. \qquad \text{(C.5)}$$

Thanks to Itô's formula, we can still do calculus with stochastic processes satisfying (SDE); the resulting set of differentiation rules is known as *Itô—or stochastic—calculus*.

For our purposes, we will consider multi-dimensional analogues of (SDE) where, mutatis mutandis, (*i*) $Z(t)$ evolves in $\mathbb{R}^n$; (*ii*) the drift of the process is given by a vector field $b \colon \mathbb{R}^n \to \mathbb{R}^n$; (*iii*) $W(t)$ is an $m$-dimensional Brownian motion evolving in $\mathbb{R}^m$; and (*iv*) $\sigma \colon \mathbb{R}^n \to \mathbb{R}^{n \times m}$ is the *diffusion matrix* of the SDE. In this case, Itô's formula for a $C^2$ function $f \colon \mathbb{R}^n \to \mathbb{R}$ becomes

$$df(Z(t)) = \sum_{i=1}^{n} b_i(Z(t))\frac{\partial f}{\partial z_i}\, dt + \frac{1}{2}\sum_{i,j=1}^{n}\sum_{k=1}^{m}\sigma_{ik}(Z(t))\sigma_{jk}(Z(t))\frac{\partial^2 f}{\partial z_i \partial z_j}\, dt$$
$$+ \sum_{i=1}^{n}\sum_{k=1}^{m}\sigma_{ik}(Z(t))\frac{\partial f}{\partial z_i}\, dW_k(t) \qquad \text{(C.6)}$$

In our analysis, we will also require a weaker version of Itô's formula for convex functions $f \colon \mathbb{R}^n \to \mathbb{R}$ that are not $C^2$ but are only $L$-Lipschitz smooth, i.e., $C^1$-smooth with $L$-Lipschitz continuous derivatives. We borrow the precise statement from [47, Proposition C.2] which, in our notation, gives

$$f(Z(t)) \le f(Z(0)) + \sum_{i=1}^{n}\int_0^t \partial_{z_i} f(Z(s))\, dZ_i(s) + \frac{L}{2}\int_0^s \mathrm{tr}[\sigma(Z(s))\,\sigma(Z(s))^\top]\, ds \qquad \text{(C.7)}$$

or, more explicitly,

$$f(Z(t)) \le f(Z(0)) + \sum_{i=1}^{n}\int_0^t b_k(Z(s))\, \partial_{z_i} f(Z(s))\, ds + \frac{L}{2}\int_0^s \mathrm{tr}[\sigma(Z(s))\,\sigma(Z(s))^\top]\, ds$$
$$+ \sum_{i=1}^{n}\sum_{k=1}^{m}\int_0^t \sigma_{ik}(Z(s))\, \partial_{z_i} f(Z(s))\, dW_k(s)\,. \qquad \text{(C.8)}$$

The deterministic part of (the strong version of) Itô's formula for $C^2$-smooth functions is captured by the so-called *infinitesimal generator* of (SDE), defined here as the differential operator $\mathcal{L}$ whose action on $f$ is given by

$$\mathcal{L}f(x) = \sum_{i=1}^{n} b_i(z)\frac{\partial f}{\partial z_i} + \frac{1}{2}\sum_{i,j=1}^{n}\sum_{k=1}^{m}\sigma_{ik}(z)\sigma_{jk}(z)\frac{\partial^2 f}{\partial z_i \partial z_j} \quad \text{for all } z \in \mathbb{R}^n. \qquad \text{(C.9)}$$

Accordingly, Itô's formula can be written more compactly as

$$df(Z(t)) = \mathcal{L}f(Z(t))\, dt + \nabla_z f(Z(t))^\top \sigma(Z(t))\, dW(t)\,. \qquad \text{(C.10)}$$

Thus, letting $\mathbb{P}_z(\cdot)$ denote the law of $Z$ initialized at $Z(0) \leftarrow z \in \mathbb{R}^n$, and writing $\mathbb{E}_z[\cdot]$ for the corresponding expectation, we readily get

$$\mathbb{E}_z[f(Z(t))] = f(z) + \mathbb{E}_z\left[\int_0^t \mathcal{L}f(Z(s))\, ds\right] \quad \text{for all } t \ge 0. \qquad \text{(C.11)}$$

This shows that the infinitesimal generator of $Z$ captures precisely the mean part of the evolution of $f(Z(t))$ under (SDE). In fact, this simple expression admits a far-reaching generalization known as *Dynkin's formula* [58, Chap. 7.4]:

**Proposition C.1** (Dynkin's formula). *Suppose that $Z(t)$ is initialized at $Z(0) \leftarrow z \in \mathbb{R}^n$. Then, for every bounded stopping time $\tau$ and every $C^2$-smooth function $f : \mathbb{R}^n \to \mathbb{R}$, we have*

$$\mathbb{E}_z[f(Z(\tau))] = f(z) + \mathbb{E}_z\left[\int_0^\tau \mathcal{L}f(Z(s))\,ds\right]. \tag{C.12}$$

Moving forward, the matrix

$$A(z) = \sigma(z)\sigma(z)^\top \tag{C.13}$$

or, in components,

$$A_{ij}(z) = \sum_{k=1}^m \sigma_{ik}(z)\sigma_{jk}(z) \quad i, j = 1, \ldots, n, \tag{C.14}$$

is known as the *principal symbol* of $\mathcal{L}$, and we say that $\mathcal{L}$ is *uniformly elliptic* if there exists some $c > 0$ such that $u^\top A(z)u \geq c\|u\|^2$ for all $z, u \in \mathbb{R}^n$ (that is, if the eigenvalues of $A(z)$ are positive and uniformly bounded away from $0$). If this is the case, the noise in (SDE) is "uniformly exciting" in the sense that it does not vanish along any direction at any point of the state space of the process. Concretely, by standard results—see e.g., [56, Sec. 3.3.6.1] and references therein—this implies that every region of $\mathbb{R}^n$ is visited by $Z(t)$ with positive probability, viz.

$$\mathbb{P}_z(Z(t) = z' \text{ for some } t > 0) > 0 \quad \text{for all } z, z' \in \mathbb{R}^n. \tag{C.15}$$

If (SDE) is uniformly elliptic—i.e., if the infinitesimal generator thereof is uniformly elliptic—the behavior of $Z(t)$ can be further classified as *transient* or *recurrent*. Formally, these two fundamental notions are defined as follows:

**Definition C.1.** Suppose that (SDE) is initialized at some $z \in \mathbb{R}^n$. Then:

1. $Z(t)$ is *transient* from $z \in \mathbb{R}^n$ if it escapes every compact subset $\mathcal{K}$ of $\mathbb{R}^n$ in finite time, i.e., there exists some (possibly random) $T_\mathcal{K} < \infty$ such that

$$\mathbb{P}_z(Z(t) \notin \mathcal{K} \text{ for all } t \geq T_\mathcal{K}) = 1. \tag{C.16}$$

2. $Z(t)$ is *recurrent* relative to a compact subset $\mathcal{K}$ of $\mathbb{R}^n$ if the hitting time

$$\tau_\mathcal{K} = \inf\{t > 0 : Z(t) \in \mathcal{K}\} \tag{C.17}$$

is finite (a.s.). If, in addition, $\mathbb{E}[\tau_\mathcal{K}] < \infty$, we will say that $Z(t)$ is *positive recurrent*; otherwise, $Z(t)$ will be called *null recurrent*.

If (SDE) is uniformly elliptic, we have the following fundamental dichotomy:

**Theorem C.1** (Transience / recurrence dichotomy). *Suppose that (SDE) is uniformly elliptic. Then:*

*1. If (SDE) is positive recurrent (resp. null recurrent) for some initial condition $z \in \mathbb{R}^n$ and some compact subset $\mathcal{K}$ of $\mathbb{R}^n$, then it is positive recurrent (resp. null recurrent) for every initial condition and every compact subset of $\mathbb{R}^n$.*

*2. If (SDE) is transient from some initial condition $z \in \mathbb{R}^n$, it is transient from every initial condition.*

For a more detailed version of Theorem C.1, we refer the reader to Bhattacharya [8, Proposition 3.1] who, to the best of our knowledge, was the first to state and prove this criterion. In words, Theorem C.1 simply states that, as long as (SDE) is uniformly elliptic, then it is *either transient or recurrent*; and if it is recurrent, it is *either positive or null recurrent*; no other outcome is possible. The choice of initialization or compact set in Definition C.1 does not matter (so, in particular, $Z$ cannot be transient from some region of $\mathbb{R}^n$ and recurrent from another). This crisp separation of regimes will play a major role in our analysis, and we will refer to it as the *transience / recurrence dichotomy*.

An important consequence of positive recurrence is that, under uniform ellipticity, $Z(t)$ admits a unique *invariant measure*, that is, a probability measure $\nu$ on $\mathbb{R}^n$ such that $Z(t) \sim \nu$ for all $t \geq 0$ whenever $Z(0) \sim \nu$. Importantly, the proviso that $\nu$ is a probability measure implies that $\nu(\mathbb{R}^n) < \infty$; if the process is null-recurrent, the semigroup of flows of (SDE) still admits an invariant meassure in the sense of Khasminskii [32], but this measure is no longer finite, i.e., $\nu(\mathbb{R}^n) = \infty$. Finally, if the process is transient, (SDE) does not admit such a measure.

# D   Analysis and results in continuous time

We now proceed to prove the continuous-time results for (S-FTRL) that we presented in Section 3.

**D.1. Proofs omitted from Section 3.1.** We begin with the "gentle start" results of Section 3.1, which we restate below for convenience.

**Proposition 1.** *Suppose that* (S-GDA) *is run on the game* (6a) *with initial condition* $x_0 \in \mathbb{R}^2$. *Then:*

*1.* $\lim_{t \to \infty} \mathbb{E}_{x_0} [\|X(t)\|_2^2] = \infty$, *i.e.,* $X(t)$ *escapes to infinity in mean square.*

*2.* $\mathbb{E}_{x_0} [\tau_r] = \infty$ *if* $r < \|x_0\|$, *i.e.,* $X(t)$ *takes infinite time on average to get closer to equilibrium.*

*3. The limit* $\mathcal{P}_\infty(x) = \lim_{t \to \infty} \mathcal{P}(x, t)$ *does not exist, i.e.,* $X$ *does not admit an invariant distribution.*

**Proposition 2.** *Suppose that* (S-GDA) *is run on the game* (6b) *with initial condition* $x_0 \in \mathbb{R}^2$. *Then:*

*1.* $\lim_{t \to \infty} \mathbb{E}_{x_0} [\|X(t)\|_2^2] = \sigma^2$, *i.e., the dynamics fluctuate at mean distance* $\sigma$ *from equilibrium.*

*2. The mean time required to get within distance* $r$ *of the game's equilibrium is bounded as*

$$\mathbb{E}_{x_0} [\tau_r] \leq \frac{1}{2} \frac{\|x_0\|_2^2 - r^2}{r^2 - \sigma^2} \qquad \text{for all } \sigma < r < \|x_0\|_2. \tag{10}$$

*3. The density of* $X(t)$ *is* $\mathcal{P}(x, t) = [\pi\sigma^2(1 - e^{-2t})]^{-1} \exp\left(-\frac{\|x - e^{-t}x_0\|_2^2}{(1 - e^{-2t})\sigma^2}\right)$. *In particular,* $X(t)$ *converges in distribution to a Gaussian random variable centered at* 0, *viz.*

$$\mathcal{P}_\infty(x) \equiv \lim_{t \to \infty} \mathcal{P}(x, t) = 1/(\pi\sigma^2) \cdot e^{-\|x\|_2^2/\sigma^2}. \tag{11}$$

*Proof of Proposition 1.* For our first claim, note that Itô's formula (C.6) applied to the function $f(x) = \|x\|_2^2$ under the dynamics (S-GDA) for the game (6a) readily yields the expression

$$d(\|X(t)\|_2^2) = 2X(t) \cdot dX(t) + dX(t) \cdot dX(t) = 2\sigma^2 \, dt + \sigma X(t) \cdot dW(t). \tag{D.1}$$

Hence, by (C.11), we get

$$\mathbb{E}_{x_0} [\|X(t)\|_2^2] = 2\sigma^2 t, \tag{D.2}$$

which proves our claim.

For our second claim, consider the hitting time $\tau = \inf\{t > 0 : \|X(t)\|_2 \leq r\}$ with $r < \|x_0\|_2$, and assume that $\mathbb{E}[\tau] < \infty$. Then, by Dynkin's formula (Proposition C.1) applied to $f(x) = \|x\|_2^2$ and $\tau$, we readily get

$$\mathbb{E}_{x_0} [f(X(\tau))] = f(x_0) + \mathbb{E}_{x_0} \left[ \int_0^\tau 2\sigma^2 \, ds \right] = f(x_0) + 2\sigma^2 \mathbb{E}_{x_0} [\tau] \geq \|x_0\|_2^2. \tag{D.3}$$

However, since $f(X(\tau)) = r^2$ by construction, we readily get $r^2 \geq \|x_0\|_2^2$, a contradiction. This shows that $\mathbb{E}_{x_0} [\tau] = \infty$, as asserted.

Finally, for our third claim, it is easy to check that (S-GDA) is uniformly elliptic under the stated assumptions. Thus, by Theorem C.1 and the fact that $\mathbb{E}_{x_0} [\tau] = \infty$, it follows that $X(t)$ cannot be positive recurrent. By the discussion following Theorem C.1, this implies that $X(t)$ does not admit an inveriant measure, so the density $\mathcal{P}(x, t)$ of $X(t)$ does not converge to a limit either. ∎

*Proof of Proposition 2.* Under the dynamics (S-GDA) for the game (6b), each coordinate of $X(t)$ evolves as an Ornstein–Uhlenbeck process, viz.

$$dX_i(t) = -X_i(t) \, dt + \sigma \, dW_i(t) \quad \text{for } i = 1, 2. \tag{D.4}$$

Since the processes are decoupled, we conclude by standard stochastic analysis arguments [36, Example 7.4.5] that

$$X_i(t) = X_i(0)e^{-t} + \sigma \int_0^t e^{-(t-s)} \, dW_i(s). \tag{D.5}$$

In turn, by [36, Theorem 7.4.7], this implies that the transition probability kernel of $X_i(t)$ is given by

$$\mathcal{P}_i(x_i, t) = \frac{1}{\sigma\sqrt{\pi(1 - e^{-2t})}} \exp\left(-\frac{(x_i - e^{-t}x_{i,0})^2}{(1 - e^{-2t})\sigma^2}\right) \quad \text{for } i = 1, 2, \tag{D.6}$$

that is, $X_i(t)$ follows a Gaussian distribution with mean $\mathbb{E}_{x_{i,0}}[X_i(t)] = x_{i,0}e^{-t}$ and variance

$$\mathbb{E}[X_i^2(t)] = \frac{\sigma^2}{2}\left(1 - e^{-2t}\right). \tag{D.7}$$

Our first and third claims then follow immediately.

For our second claim, note that the infinitesimal generator of $X(t)$ is now given by

$$\mathcal{L}f(x) = -\langle \nabla f(x), x \rangle + \frac{1}{2}\sigma^2 \Delta f(x), \tag{D.8}$$

where $\Delta f \equiv \operatorname{tr} \nabla^2 f$ denotes the Laplacian of $f$. Then, Dynkin's formula (Proposition C.1) applied to $f(x) = \|x\|_2^2$ at the truncated hitting time $\tau_r \wedge t \equiv \min\{\tau_r, t\}$, $t > 0$, readily yields

$$
\begin{aligned}
\mathbb{E}_{x_0}[\|X(\tau_r \wedge t)\|_2^2] &= \|x_0\|_2^2 + \mathbb{E}_{x_0}\left[\int_0^{\tau_r \wedge t} 2[\sigma^2 - \|X(s)\|_2^2]\, ds\right] \\
&\leq \|x_0\|_2^2 + \mathbb{E}_{x_0}\left[\int_0^{\tau_r \wedge t} 2(\sigma^2 - r^2)\, ds\right] \\
&= \|x_0\|_2^2 + 2(\sigma^2 - r^2)\, \mathbb{E}_{x_0}[\tau_r \wedge t].
\end{aligned} \tag{D.9}
$$

Since $\|X(\tau_r \wedge r)\|_2^2 \geq r^2$ by construction (recall that $\|x_0\|_2 > r$), we get

$$\mathbb{E}_{x_0}[\tau_r \wedge t] \leq \frac{\|x_0\|_2^2 - r^2}{2(r^2 - \sigma^2)} \quad \text{for all } t > 0. \tag{D.10}$$

Since $\mathbb{E}_{x_0}[\tau_r \wedge t]$ is uniformly bounded, our claim follows by taking the limit $t \to \infty$ (so $\tau_r \wedge t \to \tau_r$ pointwise), and invoking the dominated convergence theorem. $\blacksquare$

**D.2. General properties of the dynamics (S-FTRL).** We now proceed to establish the properties of the stochastic dynamics (S-FTRL) in the general case, for null- and strongly monotone games respectively. Before doing so, we begin with a result of a book-keeping nature (which is, however, necessary to ensure that the ensuing questions are meaningful).

**Proposition D.1.** *Suppose that $\sigma$ is Lipschitz continuous. Then, for every initial condition $X(0) \leftarrow x_0 = Q(y_0) \in \mathcal{X}$, the dynamics* (S-FTRL) *admit a unique strong solution that exists for all time.*

*Proof.* Note that the dynamics (S-FTRL) can be recast in fully autonomous form as

$$dY(t) = v(Q(Y(t)))\, dt + \sigma(Q(Y(t))) \cdot dW(t). \tag{D.11}$$

Note further that $v$, $\sigma$ and $Q$ are all Lipschitz, by our standing assumptions for the game, our assumptions here, and Proposition B.1 respectively. In turn, this implies that the compositions $\tilde{v} = v \circ Q$ and $\tilde{\sigma} = \sigma \circ Q$ are likewise Lipschitz continuous, so our claim follows from the existence and uniqueness theorem for SDEs with Lipschitz data, see e.g., [58, Theorem 5.2.1]. $\blacksquare$

Our next result is an ancillary calculation responsible for much of the heavy lifting in the upcoming analysis.

**Proposition D.2.** *Fix a base point $p \in \mathcal{X}$ and consider the energy function*

$$E(y) := F(p, y) = h(p) + h^*(y) - \langle y, p \rangle \quad \text{for } y \in \mathcal{Y}. \tag{D.12}$$

*Then, for every stopping time $\tau \geq 0$, we have*

$$
\begin{aligned}
E(Y(\tau)) - E(Y(0)) &\leq \int_0^\tau \langle v(X(s)), X(s) - p \rangle\, ds + \frac{\sigma_{\max}^2}{2K}\tau \\
&\quad + \int_0^\tau (X(s) - p)^\top \sigma(X(s))\, dW(s).
\end{aligned} \tag{D.13}
$$

*If, in particular, $Q$ is smooth, we have*

$$
\begin{aligned}
E(Y(\tau)) - E(Y(0)) &= \int_0^\tau \langle v(X(s)), X(s) - p \rangle\, ds \\
&\quad + \frac{1}{2}\int_0^\tau \operatorname{tr}[\Sigma(X(s))\, \operatorname{Jac} Q(Y(s))]\, ds \\
&\quad + \int_0^\tau (X(s) - p)^\top \sigma(X(s))\, dW(s).
\end{aligned} \tag{D.14}
$$

*Proof.* Assume first that $Q$ is $C^1$-smooth; In this case, by Lemma B.1, we have $\nabla F(p, y) = Q(y) - p$, and hence,

$$\nabla^2 E(y) = \nabla^2 h^*(y) = \text{Jac } Q(y).$$  (D.15)

Thus, by Itô's formula (C.6), we readily get

$$\begin{aligned}
dE(Y(t)) &= (X(t) - p) \cdot dY(t) + \frac{1}{2} \text{tr}[\sigma^\top(X(t)) \nabla^2 E(Y(t)) \sigma(X(t))] \, dt \\
&= \langle v(X(t)), X(t) - p \rangle \, dt \\
&\quad + \frac{1}{2} \text{tr}[\Sigma(X(t)) \text{ Jac } Q(Y(t))] \, dt \\
&\quad + (X(t) - p)^\top \sigma(X(t)) \, dW(t)
\end{aligned}$$  (D.16)

so (D.14) follows.

Now, if $Q$ is not smooth, Proposition B.1 shows that it is still $(1/K)$-Lipschitz continuous, which, equivalently, means that $h^*$ is $(1/K)$-Lipschitz smooth. Thus, (D.13) follows by the weak Itô formula for Lipschitz smooth functions (C.7) applied to $h^*$, and noting that $\text{tr}[\sigma(x)\sigma(x)^\top] \leq d\sigma_{\max}^2$. ∎

**D.3. The null-monotone case.** We begin our analysis proper with our result for null-monotone games, which we restate below for convenience.

**Theorem 1** (Null-monotone games). *Suppose that* (S-FTRL) *is run with a smooth mirror map $Q$ in a null-monotone game $\mathcal{G}$. Suppose further that the game admits an interior equilibrium $x^*$, and consider the hitting times $\tau_\varepsilon^- := \inf\{t > 0 : F_t \leq F_0 - \varepsilon\}$ and $\tau_\varepsilon^+ := \inf\{t > 0 : F_t \geq F_0 + \varepsilon\}$. If $\sigma_{\min}^2 > 0$ and $\varepsilon > 0$ is small enough, then*

$$\mathbb{E}_{x_0}[\tau_\varepsilon^-] = \infty \quad \text{and} \quad \mathbb{E}_{x_0}[\tau_\varepsilon^+] \leq 2\varepsilon / (\kappa \sigma_{\min}^2)$$  (15)

*for some constant $\kappa \equiv \kappa_\varepsilon > 0$; in addition, $X(t)$ does not admit a limiting distribution in this case.*

*Proof.* We start with the decreasing case, where we argue by contradiction. Specifically, let $x^*$ be an equilibrium of $\mathcal{G}$, and assume that $\mathbb{E}_{x_0}[\tau_\varepsilon^-] < \infty$. Then, by applying Dynkin's formula to the energy function $E(y)$ at $\tau_\varepsilon^-$ for $p \leftarrow x^*$ (cf. Propositions C.1 and D.2), we readily get

$$\begin{aligned}
\mathbb{E}_{x_0}[E(Y(\tau_\varepsilon^-))] &= E(y_0) + \mathbb{E}_{x_0}\left[\int_0^{\tau_\varepsilon^-} \left(\langle v(X(s)), X(s) - x^* \rangle + \frac{1}{2} \text{tr}[\Sigma(X(s)) \text{ Jac } Q(Y(s))]\right) ds\right] \\
&= F_0 + \frac{1}{2} \mathbb{E}_{x_0}\left[\int_0^{\tau_\varepsilon^-} \text{tr}[\Sigma(X(s)) \text{ Jac } Q(Y(s))] \, ds\right] \qquad \text{\# by null monotonicity} \\
&\geq F_0
\end{aligned}$$  (D.17)

where the last line follows from the fact that $\Sigma$ and $\text{Jac } Q$ are both positive-semidefinite. However, since $\mathbb{E}_{x_0}[E(Y(\tau_\varepsilon^-))] = F_0 - \varepsilon$ by the definition of $\tau_\varepsilon^-$, we get $F_0 - \varepsilon \geq F_0$, a contradiction which establishes our claim.

Since $\sigma_{\min} > 0$, we further conclude that $Y(t)$ is uniformly elliptic. Thus, for any compact set $\mathcal{K} \subseteq \{y \in \mathcal{Y} : F(x^*, y) \leq F_0 - \varepsilon\}$, the hitting time $\tau_\mathcal{K} = \inf\{t > 0 : Y(t) \in \mathcal{K}\}$ will be infinite on average (because $\mathbb{E}_{x_0}[\tau_\mathcal{K}] \geq \mathbb{E}_{x_0}[\tau_\varepsilon^-] = \infty$), so, by Theorem C.1, $Y(t)$ cannot be positive recurrent. In turn, this implies that $Y(t)$ does not admit an invariant measure on $\mathcal{Y}$, which proves our claim.

Finally, for the second part of (15), applying Dynkin's formula to the energy function $E(y)$ for $p \leftarrow x^*$ at the truncated hitting times $\tau_\varepsilon^+ \wedge t$, $t > 0$, we get:

$$\begin{aligned}
\mathbb{E}_{x_0}[E(Y(\tau_\varepsilon^+) \wedge t)] &= E(y_0) + \mathbb{E}_{x_0}\left[\int_0^{\tau_\varepsilon^+ \wedge t} \left(\langle v(X(s)), X(s) - x^* \rangle + \frac{1}{2} \text{tr}[\Sigma(X(s)) \text{ Jac } Q(Y(s))]\right) ds\right] \\
&= F_0 + \frac{1}{2} \mathbb{E}_{x_0}\left[\int_0^{\tau_\varepsilon^+ \wedge t} \text{tr}[\Sigma(X(s)) \text{ Jac } Q(Y(s))] \, ds\right] \qquad \text{\# by null monotonicity} \\
&\geq F_0 + \frac{\sigma_{\min}^2}{2} \mathbb{E}_{x_0}\left[\int_0^{\tau_\varepsilon^+ \wedge t} \text{tr}[\text{Jac } Q(Y(s))] \, ds\right]
\end{aligned}$$  (D.18)

where the last line follows from the estimate

$$
\begin{aligned}
\operatorname{tr}[\Sigma \operatorname{Jac} Q] &= \operatorname{tr}[(\operatorname{Jac} Q)^{1/2} \Sigma (\operatorname{Jac} Q)^{1/2}] \\
&= (1, \ldots, 1) \cdot (\operatorname{Jac} Q)^{1/2} \Sigma (\operatorname{Jac} Q)^{1/2} \cdot (1, \ldots, 1)^{\top} \\
&\geq \sigma_{\min}^2 (1, \ldots, 1) \cdot (\operatorname{Jac} Q)^{1/2} \cdot (\operatorname{Jac} Q)^{1/2} \cdot (1, \ldots, 1)^{\top} \\
&= \sigma_{\min}^2 \operatorname{tr}[\operatorname{Jac} Q] .
\end{aligned}
\tag{D.19}
$$

By the assumptions of the theorem (smooth $Q$ and interior initialization), it follows that the (necessarily compact) set $\mathcal{D}_\varepsilon := \{x = Q(y) : F(x^*, y) \leq F_0 + \varepsilon\}$ is contained in the relative interior $\operatorname{ri} \mathcal{X}$ of $\mathcal{X}$. In turn, this implies that $\kappa \equiv \kappa_\varepsilon := \min\{\operatorname{tr}[\operatorname{Jac} Q(y)] : F(x^*, y) \leq F_0 + \varepsilon\} > 0$, so (D.18) becomes

$$
F_0 + \varepsilon \geq F_0 + \frac{1}{2} \kappa \sigma_{\min}^2 \mathbb{E}_{x_0}[\tau_\varepsilon^+ \wedge t]
\tag{D.20}
$$

and hence

$$
\mathbb{E}_{x_0}[\tau_\varepsilon^+ \wedge t] \leq \frac{2\varepsilon}{\kappa \sigma_{\min}^2} \quad \text{for all } t \geq 0.
\tag{D.21}
$$

This shows that $\mathbb{E}_{x_0}[\tau_\varepsilon^+ \wedge t]$ is uniformly bounded, so the upper bound in (15) follows by letting $t \to \infty$ (which implies $\tau_\varepsilon^+ \wedge t \to \tau_\varepsilon^+$ pointwise), and invoking the dominated convergence theorem. ∎

**D.4. The strongly monotone case.** We now turn to our main result for strongly monotone games. Our proof strategy draws on methods related to the analysis of (S-FTRL) in the context of convex minimization, as explored by [47], and incorporating ideas that can be traced back to [28].

For convenience, we begin by restating Theorem 2.

**Theorem 2** (Strongly monotone games). *Suppose that* (S-FTRL) *is run in an $\alpha$-strongly monotone game $\mathcal{G}$, and consider the hitting time*

$$
\tau_r := \inf\{t > 0 : X(t) \in \mathbb{B}_r(x^*)\}
\tag{16}
$$

*where $\mathbb{B}_r(x^*) = \{x : \|x - x^*\| \leq r\}$ is a ball of radius $r$ centered on the (necessarily unique) equilibrium $x^*$ of $\mathcal{G}$. Then:*

$$
\mathbb{E}_{x_0}[\tau_r] \leq (F_0/\alpha)/(r^2 - r_\sigma^2) \quad \text{for all } r > r_\sigma,
\tag{17}
$$

*where $r_\sigma := \sigma_{\max}/\sqrt{2K\alpha}$. If, in addition, $\sigma_{\min} > 0$ and $x^*$ is interior, $X(t)$ admits an invariant distribution concentrated in a ball of radius $\mathcal{O}(\sigma_{\max})$ around $x^*$, and we have*

$$
\lim_{t \to \infty} \mu_t(\mathbb{B}_r(x^*)) \geq 1 - r_\sigma^2/r^2 \quad \text{for all } r > r_\sigma.
\tag{18}
$$

*Proof.* Our proof proceeds along the following basic steps:

*Step 1.* Deriving an estimate for the mean hitting time $\mathbb{E}_{x_0}[\tau_r]$.

*Step 2.* Descending to a restricted process $\tilde{Y}(t)$ where any redundant degrees of freedom in $Y(t)$ have been "modded out".

*Step 3.* Showing that the restricted process is positive recurrent.

*Step 4.* Estimating the resulting invariant distribution and pushing the result forward to $X(t)$.

In what follows, we go through the steps outlined above, one at a time.

**Step 1: Estimating the hitting time.** We begin with the hitting time estimate (17). To that end, setting $p \leftarrow x^*$ in Proposition D.2, we get

$$
\begin{aligned}
E(Y(\tau)) - E(y_0) &= \int_0^\tau \langle v(X(s)), X(s) - x^* \rangle \, ds + \frac{1}{2K} \int_0^\tau \operatorname{tr}[\Sigma(X(s))] \, ds \\
&\quad + \int_0^\tau (X(s) - x^*)^{\top} \sigma(X(s)) \, dW(s) \\
&\leq -\alpha \int_0^\tau \|X(s) - x^*\|^2 \, ds + \frac{\sigma_{\max}^2}{2K} \tau + M(\tau) .
\end{aligned}
\tag{D.22}
$$

where we set

$$M(t) = \int_0^t (X(s) - x^*)^\top \sigma(X(s)) \, dW(s) \,. \tag{D.23}$$

Thus, by a quick rearrangement, we obtain

$$\alpha \int_0^\tau \|X(s) - x^*\|^2 \, ds \le E(y_0) - E(Y(\tau)) + \frac{\sigma_{\max}^2 \tau}{2K} + M(\tau) \tag{D.24}$$

and hence, with $E \ge 0$:

$$\int_0^\tau \|X(s) - x^*\|^2 \, ds \le \frac{F_0}{\alpha} + \frac{\sigma_{\max}^2 \tau}{2\alpha K} + \frac{M(\tau)}{\alpha} \,. \tag{D.25}$$

Thus, applying the above to the truncated hitting time $\tau \leftarrow \tau_r \wedge t \equiv \min\{\tau_r, t\}$, $t > 0$, we get

$$r^2 \, \mathbb{E}_{x_0}[\tau_r \wedge t] \le \mathbb{E}_{x_0}\left[\int_0^{\tau_r \wedge t} \|X(s) - x^*\|^2 \, ds\right] \qquad \text{\# b/c } \|X(s) - x^*\| \ge r \text{ for } s \le \tau_r \wedge t$$

$$\le \frac{F_0}{\alpha} + r_\sigma^2 \, \mathbb{E}_{x_0}[\tau_r \wedge t] + \frac{1}{\alpha} \, \mathbb{E}_{x_0}[M(\tau_r \wedge t)] \,. \tag{D.26}$$

Since $\tau_r \wedge t \le t$ is uniformly bounded, we will have $\mathbb{E}_{x_0}[M(\tau_r \wedge t)] = \mathbb{E}[M(0)] = 0$ by the optional sampling theorem for continuous-time martingales [31, Theorem 3.22]. Thus, a simple rearrangement gives

$$\mathbb{E}_{x_0}[\tau_r \wedge t] \le \frac{F_0/\alpha}{r^2 - r_\sigma^2} \quad \text{for all } t \ge 0. \tag{D.27}$$

This shows that $\mathbb{E}_{x_0}[\tau_r \wedge t]$ is uniformly bounded, so the bound (17) follows by letting $t \to \infty$ (which implies $\tau_r \wedge t \to \tau_r$ pointwise), and invoking the dominated convergence theorem.

**Step 2: Descending to the restricted process.** We now proceed to examine the recurrence properties of $X(t)$. To that end, note first that the assumption $\sigma_{\min} > 0$ directly implies that (S-FTRL) is uniformly elliptic in the sense discussed in Appendix C. As such, consider the set

$$\mathcal{D}_r := Q^{-1}(\mathbb{B}_r(x^*)) = \{y \in \mathcal{Y} : \|Q(y) - x^*\| \le r\} \,. \tag{D.28}$$

and note that

$$\tau_r = \inf\{t > 0 : X(t) \in \mathbb{B}_r(x^*)\} = \inf\{t > 0 : Y(t) \in \mathcal{D}_r\} \tag{D.29}$$

so $Y(t)$ is positive recurrent relative to $\mathcal{D}_r$. Thus, if $\mathcal{D}_r$ is compact, Theorem C.1 immediately shows that $Y(t)$ is positive recurrent, and hence admits an invariant measure $\nu$ on $\mathcal{Y}$. In general however, $\mathcal{D}_r$ *need not be* compact, so we cannot conclude that $Y(t)$ is recurrent from the fact that it hits $\mathcal{D}_r$ in finite time on average.

To circumvent this difficulty, we will consider a "restricted" process which *is* positive recurrent, while retaining all information present in $Y(t)$. The main idea here will be to "collapse" the fibers of $Q$, that is, those directions in $\mathcal{Y}$ which map to the same point in $\mathcal{X}$ under $Q$: since the dynamics (S-FTRL) factor through $X(t) = Q(Y(t))$, these directions carry no relevant information, so they can be effectively discarded.

To carry all this out, let $\tilde{\mathcal{V}}$ denote the *"tangent hull"* of $\mathcal{X}$ in $\mathcal{V}$, viz.

$$\tilde{\mathcal{V}} := \text{aff}(\mathcal{X} - \mathcal{X}) = \{z \in \mathcal{V} : x + tz \in \mathcal{X} \text{ for all sufficiently small } t > 0 \text{ and all } x \in \text{ri } \mathcal{X}\} \,. \tag{D.30}$$

In words, $\tilde{\mathcal{V}}$ is the smallest subspace of $\mathcal{V}$ which contains $\mathcal{X}$ when the latter is translated to the origin so, by construction, $\mathcal{X}$ is full-dimensional when viewed as a subset of $\tilde{\mathcal{V}}$.[7] In this sense, $\tilde{\mathcal{V}}$ contains all the "essential" directions of motion of the problem.

Dually to the above, we also consider the corresponding dual space $\tilde{\mathcal{Y}} \equiv \tilde{\mathcal{V}}^*$ of $\tilde{\mathcal{V}}$; this is not a subspace of $\mathcal{Y}$, but there exists a canonical surjection $\Pi\colon \mathcal{Y} \twoheadrightarrow \tilde{\mathcal{Y}}$ defined by restricting the action of $y \in \mathcal{Y}$ to $\tilde{\mathcal{V}}$, that is,

$$\langle \Pi(y), z \rangle = \langle y, z \rangle \quad \text{for all } z \in \tilde{\mathcal{V}}. \tag{D.31}$$

The kernel of $\Pi$ is precisely the *annihilator* $\text{Ann}(\tilde{\mathcal{V}})$ of $\tilde{\mathcal{V}}$, i.e.,

$$\ker \Pi = \text{Ann}(\tilde{\mathcal{V}}) \equiv \{w \in \mathcal{Y} : \langle w, z \rangle = 0 \text{ for all } z \in \tilde{\mathcal{V}}\} \tag{D.32}$$

---

[7]Specifically, unless $\mathcal{X}$ is a singleton, it has nonempty topological interior when viewed as a subset of $\tilde{\mathcal{V}}$.

so, by the first isomorphism theorem, we get a canonical identification $(\mathcal{V}/\tilde{\mathcal{V}})^* \cong \ker\Pi$.

The main reason for descending from $\mathcal{Y}$ to $\tilde{\mathcal{Y}}$ is the following: in the original space $\mathcal{Y}$, we have $Q(y + w) = Q(y)$ whenever $w$ annihilates $\tilde{\mathcal{V}}$, cf. Proposition B.1. As a result, the inverse image of *any* compact subset of $\mathcal{X}$ under $Q$ will always contain a copy of $\mathrm{Ann}(\tilde{\mathcal{V}})$, so it can never be compact itself. By contrast, by "modding out" $\mathrm{Ann}(\tilde{\mathcal{V}})$ and descending to the restricted space $\tilde{\mathcal{Y}}$, this is no longer the case.

To move forward, consider the *restricted mirror map* $\tilde{Q}\colon \tilde{\mathcal{Y}} \to \mathcal{X}$ given by

$$\tilde{Q}(\tilde{y}) = Q(y) \quad \text{whenever } \Pi(y) = \tilde{y}. \tag{D.33}$$

By the last item of Proposition B.1 we have $Q(y) = Q(y + w)$ whenever $w \in \mathrm{Ann}(\tilde{\mathcal{V}})$; this means that the choice of representative in (D.33) does not matter, so $\tilde{Q}$ is well-defined. Accordingly, letting

$$\tilde{Y}(t) = \Pi(Y(t)) \tag{D.34}$$

and applying $\Pi$ to (S-FTRL) yields the "restricted" dynamics

$$d\tilde{Y}(t) = d(\Pi \cdot Y(t)) = \Pi \cdot v(X(t))\, dt + \Pi \cdot \sigma(X(t)) \cdot dW(t) \tag{D.35}$$

where $X(t) = Q(Y(t)) = \tilde{Q}(\tilde{Y}(t))$ and, in a slight abuse of notation, we are overloading the symbol $\Pi$ to denote both the linear map $\Pi\colon \mathcal{Y} \to \tilde{\mathcal{Y}}$ and its representation as a matrix. These dynamics represent a time-homogeneous SDE in terms of $\tilde{Y}$, and they will be our main object of study in the rest of our proof.

**Step 3: Positive recurrence of the restricted process.** With all this in hand, positive recurrence for the restricted process $\tilde{Y}(t)$ boils down to the following: *a)* verifying that the infinitesimal generator of $\tilde{Y}$ is uniformly elliptic; and *b)* showing that the mean time required for $\tilde{Y}(t)$ to reach some compact set of $\tilde{\mathcal{Y}}$ is finite.

We begin by establishing uniform ellipticity. In view of (D.35), the principal symbol (C.13) of the infinitesimal generator of $\tilde{Y}(t)$ is

$$A = (\Pi \cdot \sigma) \cdot (\Pi \cdot \sigma)^\top = \Pi \sigma \sigma^\top \Pi^\top = \Pi \Sigma \Pi^\top . \tag{D.36}$$

Since $\Sigma \succcurlyeq \sigma_{\min}^2 I$, we readily get

$$A \succcurlyeq \sigma_{\min}^2 \Pi \Pi^\top \succcurlyeq \sigma_{\min}^2 \pi_{\min}^2 I \tag{D.37}$$

with $\sigma_{\min} > 0$ (by assumption) and $\pi_{\min} := \lambda_{\min}(\Pi\Pi^\top) > 0$ (because $\Pi$ is surjective, so it has full rank). This shows that the principal symbol $\Pi\Sigma\Pi^\top$ of the generator of $\tilde{Y}$ is uniformly positive-definite, so $\tilde{Y}$ is itself uniformly elliptic.

For the second component of our proof of positive recurrence, recall that $x^* \in \mathrm{ri}\,\mathcal{X}$, so there exists some sufficiently small $r > 0$ such that the (compact convex) set

$$\mathcal{K}_r := \mathbb{B}_r \cap \mathcal{X} = \{x \in \mathcal{X} : \|x - x^*\| \le r\} \tag{D.38}$$

lies in its entirety within $\mathrm{ri}\,\mathcal{X}$. We then claim that the inverse image

$$\tilde{\mathcal{D}}_r := \tilde{Q}^{-1}(\mathcal{K}_r) = \{\tilde{y} \in \tilde{\mathcal{Y}} : \|\tilde{Q}(\tilde{y}) - x^*\| \le r\} \tag{D.39}$$

of $\mathcal{K}_r$ under the restricted mirror map $\tilde{Q}$ is compact. To see this, note first that $\tilde{\mathcal{D}}_r = \partial h(\mathcal{K}_r)$ by Proposition B.1.[8] Thus, given that $\mathcal{K}_r$ is a convex body in $\tilde{\mathcal{V}}$ that is entirely contained in the (relative) interior of the prox-domain $\mathcal{X}_h$ of $h$ (because $\mathrm{ri}\,\mathcal{X} \subseteq \mathrm{dom}\,\partial h \equiv \mathcal{X}_h$), it follows that $\partial h(\mathcal{K}_r)$ is itself compact by the upper hemicontinuity of $\partial h$ [24, Remark 6.2.3].

To conclude, note that

$$\begin{aligned}
\inf\{t > 0 : \tilde{Y}(t) \in \tilde{\mathcal{D}}_r\} &= \inf\{t > 0 : \|\tilde{Q}(\tilde{Y}(t)) - x^*\| \le r\} \\
&= \inf\{t > 0 : \|Q(Y(t)) - x^*\| \le r\} \\
&= \inf\{t > 0 : X(t) \in \mathbb{B}_r(x^*)\} \\
&= \tau_r
\end{aligned} \tag{D.40}$$

so it follows from (17) that $\tilde{Y}(t)$ hits $\tilde{\mathcal{D}}_r$ in finite time on average. Since $\tilde{Y}(t)$ is uniformly elliptic, Theorem C.1 shows that it is positive recurrent, as claimed.

---

[8]Strictly speaking, we are viewing here $\partial h$ as taking values in $\tilde{\mathcal{Y}}$ instead of $\mathcal{Y}$; this is a simple matter of identifying $h\colon \mathcal{V} \to \mathbb{R}$ with its canonical restriction to $\tilde{\mathcal{V}} \subseteq \mathcal{V}$.

**Step 4: Estimating the long-run occupation measure.** Since the restricted process $\tilde{Y}(t)$ is a positive recurrent Itô diffusion, standard results show that it admits an invariant distribution $\tilde{\nu}$ on $\tilde{\mathcal{Y}}$ which satisfies the law of large numbers

$$\lim_{t \to \infty} \frac{1}{t} \int_0^t f(\tilde{Y}(s)) \, ds = \int_{\tilde{\mathcal{Y}}} f \, d\tilde{\nu} \tag{D.41}$$

for every $\tilde{\nu}$-integrable test function $f$ on $\tilde{\mathcal{Y}}$. Thus, letting $\nu = \tilde{Q}_* \tilde{\nu} \equiv \tilde{\nu} \circ \tilde{Q}^{-1}$ denote the corresponding push-forward measure on $\mathcal{X}$, we get

$$
\begin{aligned}
\lim_{t \to \infty} \mu_t(\mathbb{B}_r) &= \lim_{t \to \infty} \frac{1}{t} \int_0^t \mathbb{1}\{X(s) \in \mathbb{B}_r\} \, ds \\
&= \lim_{t \to \infty} \frac{1}{t} \int_0^t \mathbb{1}\{\tilde{Q}(\tilde{Y}(s)) \in \mathbb{B}_r\} \, ds \\
&= \lim_{t \to \infty} \frac{1}{t} \int_0^t \mathbb{1}\{\tilde{Y}(s) \in \tilde{\mathcal{D}}_r\} \, ds \\
&= \int_{\tilde{\mathcal{Y}}} \mathbb{1}\{\tilde{y} \in \tilde{\mathcal{D}}_r\} \, d\tilde{\nu}(\tilde{y}) \\
&= \tilde{\nu}(\tilde{\mathcal{D}}_r) \, .
\end{aligned}
\tag{D.42}
$$

In a similar manner, we also get

$$
\begin{aligned}
1 - \tilde{\nu}(\tilde{\mathcal{D}}_r) &= \lim_{t \to \infty} \frac{1}{t} \, \mathbb{E}\left[ \int_0^t \mathbb{1}\{X(s) \notin \mathbb{B}_r\} \, ds \right] && \text{\# b/c } \lim_{t \to \infty} \mu_t \text{ is deterministic} \\
&\leq \lim_{t \to \infty} \frac{1}{t} \, \mathbb{E}\left[ \int_0^t \frac{\|X(s) - x^*\|^2}{r^2} \, ds \right] && \text{\# b/c } \frac{\|X(s) - x^*\|^2}{r^2} \geq 1 \text{ outside } \mathbb{B}_r \\
&\leq \lim_{t \to \infty} \frac{1}{r^2} \left[ \frac{F_0}{\alpha t} + \frac{\sigma_{\max}^2}{2\alpha K} \right] && \text{\# by (D.25)} \\
&= \frac{r_\sigma^2}{r^2} \, .
\end{aligned}
\tag{D.43}
$$

Our claim then follows by combining Eqs. (D.42) and (D.43). ∎

# E    Analysis and results in discrete time

In this appendix, we proceed to prove the discrete-time results presented in Section 4.

**E.1. The null-monotone case.** We begin with our analysis for for null-monotone games. For convenience, we restate Theorem 3 below:

**Theorem 3** (Null-monotone games). *Suppose that* (FTRL) *is run in a null-monotone game $\mathcal{G}$, and let $x^*$ be an equilibrium of $\mathcal{G}$. Suppose further that $h^*$ is strongly convex, and let $F_t = F(x^*, Y_t)$, where $F$ is the induced Fenchel coupling* (B.22). *Then $\lim_{t \to \infty} \mathbb{E}[F_t] = \infty$.*

*Proof.* By a second-order Taylor expansion with Lagrange remainder, there exists $w_t \in [Y_t, Y_{t+1}]$ such that:

$$F_{t+1} = F_t + \gamma \langle \hat{v}_t, X_t - x^* \rangle + \frac{\gamma^2}{2} \hat{v}_t \nabla^2 h^*(w_t) \hat{v}_t \, . \tag{E.1}$$

Since $\mathcal{G}$ is null-monotone, we have $\langle v(X_t), X_t - x^* \rangle = 0$ by assumption, and thus

$$F_{t+1} = F_t + \gamma \langle U_t, X_t - x^* \rangle + \frac{\gamma^2}{2} \hat{v}_t \nabla^2 h^*(w_t) \hat{v}_t \tag{E.2}$$

$$\geq F_t + \gamma \langle U_t, X_t - x^* \rangle + \frac{m}{2} \gamma^2 \|\hat{v}_t\|_*^2 \tag{E.3}$$

where $m$ denotes here the strong convexity modulus of $h^*$. Moving forward, note that (i) $\mathbb{E}[\langle U_t, X_t - x^* \rangle] = \mathbb{E}[\langle \mathbb{E}[U_t \mid \mathcal{F}_t], X_t - x^* \rangle] = 0$; and (ii) $\inf_t \mathbb{E}[\|\hat{v}_t\|_*^2] \geq \inf \mathbb{E}[\|V(x;\omega)\|_*^2] > 0$, so there exists some $V_* > 0$ such that $\mathbb{E}[\|\hat{v}_t\|_*^2] \geq V_*^2$ for all $t$. We thus get

$$
\begin{aligned}
\mathbb{E}[F_{t+1}] &\geq \mathbb{E}[F_t] + \frac{m}{2}\gamma^2\,\mathbb{E}[\|\hat{v}_t\|_*^2] \\
&\geq \mathbb{E}[F_t] + m\gamma^2 V_*^2/2 \\
&\geq F_0 + m\gamma^2 V_*^2 t/2
\end{aligned}
\tag{E.4}
$$

Our result then follows by taking the limit $t \to \infty$. ∎

**E.2. The strongly monotone case.** We now turn to our main result for strongly monotone games, which we restate below for convenience.

**Theorem 4** (Strongly monotone games). *Suppose that* (FTRL) *is run in an $\alpha$-strongly monotone game $\mathcal{G}$, and consider the hitting time*

$$
\tau_r := \inf\{t > 0 : X(t) \in \mathbb{B}_r(x^*)\}
\tag{22}
$$

*where $\mathbb{B}_r(x^*) = \{x : \|x - x^*\| \leq r\}$ is a ball of radius $r$ centered on the (necessarily unique) equilibrium $x^*$ of $\mathcal{G}$. Then, for all $r > r_\sigma := \sqrt{\gamma(\sigma^2 + \beta^2)/(\alpha K)}$, we have*

$$
\mathbb{E}[\tau_r] \leq \frac{1}{\alpha\gamma(r^2 - r_\sigma^2)} \times \begin{cases} F_0 & \text{if } X_0 \notin \mathbb{B}_r(x^*), \\ F_0 + \alpha\gamma r^2 & \text{if } X_0 \in \mathbb{B}_r(x^*), \end{cases}
\tag{23}
$$

*where $F_0 = F(x^*, Y_0)$. If, in addition, $x^*$ is interior, $X_t$ admits a unique invariant distribution to which it converges in total variation, and we have*

$$
\lim_{t \to \infty} \frac{1}{t} \mathbb{E}\left[ \sum_{s=0}^{t} \mathbb{1}\{X_t \in \mathbb{B}_r(x^*)\} \right] \geq 1 - r_\sigma^2/r^2
\tag{24}
$$

*for all $r > r_\sigma$ such that $\mathbb{B}_r(x^*) \subseteq \mathrm{ri}\,\mathcal{X}$.*

*Proof.* The main theme of our proof shadows the continuous-time analysis, but it requires distinct tools and techniques to address the specific challenges that arise in the discrete-time Markov chain setting (where, among others, the main tools of stochastic calculus cannot be applied). In a nutshell, this proceeds along the following sequence of steps. First, we derive an upper bound on the expected hitting time of the process to a neighborhood of the equilibrium. Subsequently, we reduce the dynamics to a "reduced space" (formally an affine quotient of the dual space), removing redundant directions and ensuring the process evolves within a minimal and non-degenerate domain. Within this reduced space, we show that the induced Markov process satisfies several crucial probabilistic properties. Specifically, we prove:

- Irreducibility: any open set in the state space can be reached with positive probability.

- Minorization: after entering certain regions of the space, the process mixes sufficiently to allow for probabilistic regeneration.

- Uniform control of return times: the expected time to revisit a neighborhood of equilibrium remains bounded regardless of the starting point within that neighborhood.

These properties collectively enable the construction of a regeneration structure, a probabilistic framework that ensures the process repeatedly returns to a well-behaved region of the state space with sufficient mixing. In turn, this enables us to establish positive Harris recurrence of the learning dynamics, a key property which ensures the existence and uniqueness of a stationary invariant distribution.

To streamline our presentation, we follow a step-by-step approach, as outlined below.

**Step 1: Deriving a hitting estimate.** Due to measurability issues, we cannot apply Dynkin's lemma directly in the discrete-time setting, which makes the proof more involved. Moreover, unlike in the continuous-time regime, we need to distinguish between different initializations. Specifically, we consider two cases depending on whether the initial state $X_0$ lies within the ball $\mathbb{B}_r(x^*)$ or not.

- *Case 1: $X_0 \notin \mathbb{B}_r(x^*)$.*

  Letting $F_t := F(x^*, Y_t)$ and unfolding (B.29b), we readily obtain:

  $$F_t \le F_0 + \gamma \sum_{s=0}^{t-1} \langle \hat{v}_s, X_s - x^* \rangle + \frac{\gamma^2}{2K} \sum_{s=0}^{t-1} \|\hat{v}_s\|_*^2 \qquad (\text{E.5})$$

  and, setting $t \leftarrow \tau_r \wedge t$, we get:

  $$F_{\tau_r \wedge t} \le F_0 + \gamma \sum_{s=0}^{(\tau_r \wedge t)-1} \langle \hat{v}_s, X_s - x^* \rangle + \frac{\gamma^2}{2K} \sum_{s=0}^{(\tau_r \wedge t)-1} \|\hat{v}_s\|_*^2 \qquad (\text{E.6})$$

  Thus, taking expectation conditional on the initial state $Y_0 = y$, we have

  $$\mathbb{E}\big[F_{\tau_r \wedge t}\big] \le F_0 + \gamma \, \mathbb{E}\left[\sum_{s=0}^{(\tau_r \wedge t)-1} \langle \hat{v}_s, X_s - x^* \rangle\right] + \frac{\gamma^2}{2K} \, \mathbb{E}\left[\sum_{s=0}^{(\tau_r \wedge t)-1} \|\hat{v}_s\|_*^2\right]$$

  $$\le F_0 + \sum_{s=0}^{t-1} \mathbb{E}\left[\left(\gamma \langle \hat{v}_s, X_s - x^* \rangle + \frac{\gamma^2}{2K} \|\hat{v}_s\|_*^2\right) \mathbb{1}(\tau_r \ge s+1)\right] \qquad (\text{E.7})$$

  For notational convenience, we denote each summand above per

  $$D_s := \mathbb{E}\left[\left(\gamma \langle \hat{v}_s, X_s - x^* \rangle + \frac{\gamma^2}{2K} \|\hat{v}_s\|_*^2\right) \mathbb{1}(\tau_r \ge s+1)\right]$$

  and noting that the random variable $\mathbb{1}(\tau_r \ge s+1)$ is $\mathcal{F}_s$-measurable, we get

  $$D_s = \mathbb{E}\left[\mathbb{E}\left[\left(\gamma \langle \hat{v}_s, X_s - x^* \rangle + \frac{\gamma^2}{2K} \|\hat{v}_s\|_*^2\right) \mathbb{1}(\tau_r \ge s+1) \,\Big|\, \mathcal{F}_s\right]\right]$$

  $$= \mathbb{E}\left[\mathbb{1}(\tau_r \ge s+1) \, \mathbb{E}\left[\gamma \langle \hat{v}_s, X_s - x^* \rangle + \frac{\gamma^2}{2K} \|\hat{v}_s\|_*^2 \,\Big|\, \mathcal{F}_s\right]\right]$$

  $$= \mathbb{E}\left[\mathbb{1}(\tau_r \ge s+1)\left(\gamma \langle v(X_s), X_s - x^* \rangle + \frac{\gamma^2}{2K} \, \mathbb{E}\big[\|\hat{v}_s\|_*^2 \,\big|\, \mathcal{F}_s\big]\right)\right]$$

  $$= \mathbb{E}\left[\mathbb{1}(\tau_r \ge s+1)\left(-\gamma \alpha \|X_s - x^*\|^2 + \frac{\gamma^2}{2K} \, \mathbb{E}\big[\|\hat{v}_s\|_*^2 \,\big|\, \mathcal{F}_s\big]\right)\right] \qquad (\text{E.8})$$

  where we used that $\mathbb{E}[U(X_s, \omega_s) \,|\, \mathcal{F}_s] = 0$. At this point, we note that $\mathbb{E}\big[\|\hat{v}_s\|_*^2 \,|\, \mathcal{F}_s\big] \le 2\,\mathbb{E}\big[\|v(X_s)\|_*^2 + \|U(X_s, \omega_s)\|_*^2 \,|\, \mathcal{F}_s\big] \le 2(\beta^2 + \sigma^2)$, and since $r_\sigma^2 \equiv \gamma(\beta^2 + \sigma^2)/(\alpha K)$, we get

  $$D_s \le \mathbb{E}\left[\left(-\gamma \alpha \|X_s - x^*\|^2 + \alpha \gamma r_\sigma^2\right) \mathbb{1}(\tau_r \ge s+1)\right]$$

  $$\le \mathbb{E}\left[\left(-\gamma \alpha r^2 + \alpha \gamma r_\sigma^2\right) \mathbb{1}(\tau_r \ge s+1)\right] \qquad (\text{E.9})$$

  where in the last step we used that $\|X_s - x^*\| \ge r^2$ on $\{\tau_r \ge s+1\}$. Thus, plugging the above bound into (E.7), we obtain

  $$\mathbb{E}\big[F_{\tau_r \wedge t}\big] \le F_0 + \sum_{s=0}^{t-1} D_s \le F_0 + \sum_{s=0}^{t-1} \mathbb{E}\left[\left(-\gamma \alpha r^2 + \alpha \gamma r_\sigma^2\right) \mathbb{1}(\tau_r \ge s+1)\right]$$

  $$= F_0 - \alpha \gamma (r^2 - r_\sigma^2) \, \mathbb{E}\left[\sum_{s=0}^{(\tau_r \wedge t)-1} 1\right]$$

  $$= F_0 - \alpha \gamma (r^2 - r_\sigma^2) \, \mathbb{E}[(\tau_r \wedge t)] \qquad (\text{E.10})$$

  As $F$ is nonnegative, we readily obtain that

  $$\alpha \gamma (r^2 - r_\sigma^2) \, \mathbb{E}[(\tau_r \wedge t)] \le F_0 \qquad (\text{E.11})$$

  and, since $r_\sigma < r$, we get:

  $$\mathbb{E}[(\tau_r \wedge t)] \le \frac{1}{\alpha \gamma (r^2 - r_\sigma^2)} F_0 \qquad (\text{E.12})$$

  Finally, taking $t \to \infty$, and invoking the monotone convergence theorem [19], we get

  $$\mathbb{E}[\tau_r] \le \frac{1}{\alpha \gamma (r^2 - r_\sigma^2)} F_0 \qquad (\text{E.13})$$

- *Case 2:* $X_0 \in \mathbb{B}_r(x^*)$. In this case, we have:

$$
\begin{aligned}
\mathbb{E}[\tau_r] &= \mathbb{E}[\mathbb{1}(Q(Y_1) \in \mathbb{B}_r(x^*)) + \mathbb{1}(Q(Y_1) \notin \mathbb{B}_r(x^*))\tau_r] \\
&= \mathbb{P}(Q(Y_1) \in \mathbb{B}_r(x^*)) + \mathbb{E}\big[\mathbb{1}(Q(Y_1) \notin \mathbb{B}_r(x^*))(1 + \mathbb{E}_{Y_1}[\tau_r])\big] \\
&= \mathbb{P}(Q(Y_1) \in \mathbb{B}_r(x^*)) + \mathbb{P}(Q(Y_1) \notin \mathbb{B}_r(x^*)) + \mathbb{E}\big[\mathbb{1}(Q(Y_1) \notin \mathbb{B}_r(x^*))\,\mathbb{E}_{Y_1}[\tau_r]\big] \\
&= 1 + \mathbb{E}\big[\mathbb{1}(Q(Y_1) \notin \mathbb{B}_r(x^*))\,\mathbb{E}_{Y_1}[\tau_r]\big] \\
&\leq 1 + \mathbb{E}\bigg[\mathbb{1}(Q(Y_1) \notin \mathbb{B}_r(x^*))\big(\alpha\gamma(r^2 - r_\sigma^2)\big)^{-1} F(x^*, Y_1)\bigg] \\
&\leq 1 + \big(\alpha\gamma(r^2 - r_\sigma^2)\big)^{-1} \mathbb{E}[F(x^*, Y_1)] \\
&\leq 1 + \big(\alpha\gamma(r^2 - r_\sigma^2)\big)^{-1} \mathbb{E}\bigg[F_0 + \gamma\langle \hat{v}_0, X_0 - x^*\rangle + \frac{\gamma^2}{2K}\|\hat{v}_0\|_*^2\bigg] \\
&\leq 1 + \big(\alpha\gamma(r^2 - r_\sigma^2)\big)^{-1} \big(F_0 + \gamma\langle v(X_0), X_0 - x^*\rangle + \alpha\gamma r_\sigma^2\big) \\
&\leq 1 + \big(\alpha\gamma(r^2 - r_\sigma^2)\big)^{-1} \big(F_0 - \gamma\alpha\|X_0 - x^*\|^2 + \alpha\gamma r_\sigma^2\big) \\
&\leq 1 + \big(\alpha\gamma(r^2 - r_\sigma^2)\big)^{-1} \big(F_0 + \alpha\gamma r_\sigma^2\big) \\
&= \frac{F_0 + \alpha\gamma r^2}{\alpha\gamma(r^2 - r_\sigma^2)}
\end{aligned}
\tag{E.14}
$$

Thus, collectively, we get:

$$
\mathbb{E}[\tau_r] \leq \frac{1}{\alpha\gamma(r^2 - r_\sigma^2)} \times \begin{cases} F_0 & \text{if } X_0 \notin \mathbb{B}_r(x^*), \\ F_0 + \alpha\gamma r^2 & \text{if } X_0 \in \mathbb{B}_r(x^*), \end{cases}
\tag{E.15}
$$

**Step 2: Descending to the restricted process.** As in the continuous-time case, establishing positive recurrence requires analyzing a "restricted" version of the process. To that end, we follow the same construction as in *Step 2 of Theorem 2*, and we define the canonical surjection $\Pi\colon \mathcal{Y} \twoheadrightarrow \tilde{\mathcal{Y}}$ by restricting the action of $y \in \mathcal{Y}$ to $\tilde{\mathcal{V}}$, that is,

$$
\langle \Pi(y), z\rangle = \langle y, z\rangle \quad \text{for all } z \in \tilde{\mathcal{V}}.
\tag{E.16}
$$

whose kernel of $\Pi$ is precisely the *annihilator* $\mathrm{Ann}(\tilde{\mathcal{V}})$ of $\tilde{\mathcal{V}}$, i.e.,

$$
\ker \Pi = \mathrm{Ann}(\tilde{\mathcal{V}}) = \{y \in \mathcal{Y} : \langle y, z\rangle = 0 \text{ for all } z \in \tilde{\mathcal{V}}\}
\tag{E.17}
$$

In addition, we consider the *restricted mirror map* $\tilde{Q}\colon \tilde{\mathcal{Y}} \to \mathcal{X}$ given by

$$
\tilde{Q}(\tilde{y}) = Q(y) \quad \text{whenever } \Pi(y) = \tilde{y}.
\tag{E.18}
$$

Accordingly, letting

$$
\tilde{Y}_t = \Pi(Y_t)
\tag{E.19}
$$

and applying $\Pi$ to (FTRL) yields the "restricted" process

$$
\tilde{Y}_{t+1} = \Pi \cdot Y_{t+1} = \Pi \cdot Y_t + \gamma(\Pi \cdot v(X_t) + \Pi \cdot U_t) = \tilde{Y}_t + \gamma(\tilde{v}(X_t) + \tilde{U}_t)
\tag{E.20}
$$

where $X_t = Q(Y_t) = \tilde{Q}(\tilde{Y}_t)$ and, in a slight abuse of notation, we are overloading the symbol $\Pi$ to denote both the linear map $\Pi\colon \mathcal{Y} \to \tilde{\mathcal{Y}}$ and its representation as a matrix. Finally, writing $\tilde{Y}$ as

$$
\tilde{Y}_{t+1} = \tilde{Y}_t + \gamma\tilde{v}(\tilde{Q}(\tilde{Y}_t)) + \gamma \mathsf{U}(\tilde{Q}(\tilde{Y}_t), \omega_t)
\tag{E.21}
$$

we conclude that it is a time-homogeneous Markov process, and we denote its kernel by $\tilde{q}$, where for any $\tilde{y} \in \tilde{\mathcal{Y}}$ and Borel set $\mathcal{A} \subseteq \tilde{\mathcal{Y}}$, we have $\tilde{q}(\tilde{y}, \mathcal{A}) = \mathbb{P}(\tilde{Y}_{t+1} \in \mathcal{A} \mid \tilde{Y}_t = \tilde{y})$.

**Step 3: Recurrence of the restricted process.** To establish the recurrence of the restricted process, we first need to understand the effect of $\Pi$ on the distribution of $\mathsf{U}(x)$. As stated in the assumptions in Section 4, the probability distribution $\nu_x$ of $\mathsf{U}(x)$ decomposes as $\nu_x = \nu_x^c + \nu_x^\perp$. Noting the push-forward measure is linear, we readily obtain that $\Pi_* \nu_x = \Pi_* \nu_x^c + \Pi_* \nu_x^\perp$, where $\Pi_* \nu_x$ denotes the push-forward measure $\mathcal{A} \mapsto (\nu_x \circ \Pi^{-1})(\mathcal{A})$. For notational convenience, we denote $\widehat{\mathcal{Y}} \equiv \mathrm{Ann}(\tilde{\mathcal{V}})$ and $p(x,y) \equiv p_x(y)$. Then, each $y \in \mathcal{Y}$ can be decomposed as $y = \tilde{y} + \hat{y}$, and since $\Pi$ has full column-rank, the measure $\Pi_* \nu_x^c$ has density with respect to the Lebesgue measure $\lambda_{\tilde{\mathcal{Y}}}$ on $\tilde{\mathcal{Y}}$, given by

$$\tilde{p}(x,\tilde{y}) = \int_{\widehat{\mathcal{Y}}} p(x,\tilde{y},\hat{y}) d\lambda_{\widehat{\mathcal{Y}}}(\hat{y}) \tag{E.22}$$

where $\lambda_{\widehat{\mathcal{Y}}}$ is the Lebesgue measure on $\widehat{\mathcal{Y}}$. Importantly, the density $\tilde{p}$ satisfies the following properties, which will be crucial for establishing the recurrence of the process. We formalize these in the proposition below, whose proof is deferred until after the theorem.

**Proposition E.1.** *Let the function $\tilde{p}$ as defined in* (E.22). *Then:*

*(i) For any compact set $\mathcal{K} \subseteq \mathcal{X}$ and every $\tilde{y} \in \tilde{\mathcal{Y}}$, it holds $\inf_{x \in \mathcal{K}} \tilde{p}(x,\tilde{y}) > 0$.*

*(ii) The function $\tilde{p}$ is (jointly) lower semi-continuous.*

**Lebesgue irreducibility.** We now show that the restricted process $\tilde{Y}_t$ is Lebesgue irreducible; that is, starting from any point in its domain, the process has a positive probability of reaching any open set with nonzero Lebesgue measure. This property is crucial for establishing recurrence, as it ensures that the process does not avoid regions of the space indefinitely.

For this, let a Borel measurable set $\mathcal{A} \subseteq \tilde{\mathcal{Y}}$ with $\lambda_{\tilde{\mathcal{Y}}}(\mathcal{A}) > 0$. We will show that $\tilde{q}(\tilde{y}, \mathcal{A}) > 0$ for all $\tilde{y} \in \tilde{\mathcal{Y}}$, which implies that $\mathcal{A}$ can be reached from any state $\tilde{y}$ in one step with positive probability.

$$\begin{aligned}
\tilde{q}(\tilde{y}, \mathcal{A}) &= \mathbb{P}\big(\tilde{Y}_{t+1} \in \mathcal{A} \,\big|\, \tilde{Y}_t = \tilde{y}\big) \\
&= \mathbb{P}\big(\tilde{y} + \gamma \tilde{v}(\tilde{Q}(\tilde{y})) + \gamma \tilde{\mathsf{U}}(\tilde{Q}(\tilde{y}), \omega) \in \mathcal{A}\big) \\
&= \mathbb{P}\big(\tilde{\mathsf{U}}(\tilde{Q}(\tilde{y}), \omega) \in \{\gamma^{-1}\mathcal{A} - \gamma^{-1}\tilde{y} - \tilde{v}(\tilde{Q}(\tilde{y}))\}\big) \\
&= \mathbb{P}\big(\tilde{\mathsf{U}}(\tilde{Q}(\tilde{y}), \omega) \in \mathcal{A}_{\tilde{y}}\big) \\
&\geq \int_{\mathcal{A}_{\tilde{y}}} \tilde{p}(\tilde{Q}(\tilde{y}), z) d\lambda_{\tilde{\mathcal{Y}}}(z) \tag{E.23}
\end{aligned}$$

where $\mathcal{A}_{\tilde{y}} \equiv \gamma^{-1}\mathcal{A} - \gamma^{-1}\tilde{y} - \tilde{v}(\tilde{Q}(\tilde{y}))$ with $\lambda_{\tilde{\mathcal{Y}}}(\mathcal{A}_{\tilde{y}}) = \gamma^{-d}\lambda_{\tilde{\mathcal{Y}}}(\mathcal{A}) > 0$. Finally, since $\tilde{p}(\tilde{Q}(\tilde{y}), \cdot)$ strictly positive, we conclude that $\tilde{q}(\tilde{y}, \mathcal{A}) > 0$, which shows that $\tilde{Y}_t$ induced by (E.20) is Lebesgue-irreducible.

**Harris recurrence.** Our next step is to show that $\tilde{Y}_t$ is Harris recurrent. This means that the process returns to every set of positive Lebesgue measure infinitely often with probability one. Establishing Harris recurrence is a key step toward proving ergodicity, as it ensures that the process does not drift away or get trapped. For this, we will show that $\tilde{\mathcal{D}}_r = \{\tilde{y} \in \tilde{\mathcal{Y}} : \|Q(\tilde{y}) - x^*\| \leq r\}$ is a recurrent set from which we can go "everywhere" with positive probability. Importantly, the set $\tilde{\mathcal{D}}_r$ is compact as shown in *Step 3 of Theorem 2*.

The first part to prove Harris recurrence is immediate from *Step 1* of our proof; namely, since $\mathbb{E}_{\tilde{y}}[\tau_r] < \infty$ for any initial condition $\tilde{y} \in \tilde{\mathcal{Y}}$, we readily get that $\mathbb{P}_{\tilde{y}}(\tau_r < \infty) = 1$.

For the second part, we will prove the so-called minorization property; that is, there exists a nontrivial measure $\mu$ and a constant $\alpha > 0$ such that

$$\tilde{q}(\tilde{y}, \mathcal{A}) \geq \alpha \mu(\mathcal{A}) \qquad \text{for all } \tilde{y} \in \tilde{\mathcal{D}}_r \text{ and Borel sets } \mathcal{A} \subseteq \tilde{\mathcal{Y}}. \tag{E.24}$$

This condition implies that, from any point in $\tilde{\mathcal{D}}_r$ the process has a uniformly lower-bounded probability of reaching any set $\mathcal{A}$ in one step according to the reference measure $\mu$.

To establish the minorization condition (E.24), we define for notational convenience the function $f : \tilde{\mathcal{Y}} \times \tilde{\mathcal{Y}} \to \mathcal{X}_h \times \tilde{\mathcal{Y}}$ as

$$f(\tilde{y}, z) = \left( \tilde{Q}(\tilde{y}), \gamma^{-1}(z - \tilde{y}) - \tilde{v}(\tilde{Q}(\tilde{y})) \right) \tag{E.25}$$

which is continuous as a composition of continuous functions. With this definition in hand, we perform the change of variables in (E.23), and we have:

$$\tilde{q}(\tilde{y}, \mathcal{A}) \geq \gamma^{-d} \int_{\mathcal{A}} \tilde{p}(f(\tilde{y}, z)) d\lambda_{\tilde{y}}(z)$$

$$\geq \gamma^{-d} \int_{\mathcal{A}} \inf_{\tilde{y} \in \tilde{\mathcal{D}}_r} \tilde{p}(f(\tilde{y}, z)) d\lambda_{\tilde{y}}(z) \tag{E.26}$$

To finally construct the measure $\mu$, we need to ensure that

$$0 < \int_{\tilde{\mathcal{Y}}} \inf_{\tilde{y} \in \tilde{\mathcal{D}}_r} \tilde{p}(f(\tilde{y}, z)) d\lambda_{\tilde{y}}(z) < \infty \tag{E.27}$$

To this end, we state the following proposition, whose proof is deferred until after the theorem to maintain the flow.

**Proposition E.2.** *The density $\tilde{p}$ satisfies:*

$$0 < \int_{\tilde{\mathcal{Y}}} \inf_{\tilde{y} \in \tilde{\mathcal{D}}_r} \tilde{p}(f(\tilde{y}, z)) d\lambda_{\tilde{y}}(z) < \infty \tag{E.28}$$

With Proposition E.1 in hand, we define the measure $\mu$ as

$$\mu(\mathcal{A}) := \frac{\int_{\mathcal{A}} \inf_{\tilde{y} \in \tilde{\mathcal{D}}_r} \tilde{p}(f(\tilde{y}, z)) d\lambda_{\tilde{y}}(z)}{\int_{\tilde{\mathcal{Y}}} \inf_{\tilde{y} \in \tilde{\mathcal{D}}_r} \tilde{p}(f(\tilde{y}, z)) d\lambda_{\tilde{y}}(z)} \qquad \text{for all Borel } \mathcal{A} \subseteq \tilde{\mathcal{Y}}. \tag{E.29}$$

Therefore, (E.26) becomes:

$$\tilde{q}(\tilde{y}, \mathcal{A}) \geq \gamma^{-d} \int_{\mathcal{A}} \inf_{\tilde{y} \in \tilde{\mathcal{D}}_r} \tilde{p}(f(\tilde{y}, z)) d\lambda_{\tilde{y}} = \gamma^{-d} \int_{\tilde{\mathcal{Y}}} \inf_{\tilde{y} \in \tilde{\mathcal{D}}_r} \tilde{p}(f(\tilde{y}, z)) d\lambda_{\tilde{y}}(z) \cdot \mu(\mathcal{A}) \tag{E.30}$$

Thus, setting $\alpha \equiv \gamma^{-d} \int_{\tilde{\mathcal{Y}}} \inf_{\tilde{y} \in \tilde{\mathcal{D}}_r} \tilde{p}(f(\tilde{y}, z)) d\lambda_{\tilde{y}}(z)$, we conclude the minorization condition (E.24).

Therefore, the set $\tilde{\mathcal{D}}_r$ is recurrent and $\mu(\tilde{\mathcal{D}}_r) > 0$ (since $\lambda_{\tilde{y}}(\tilde{\mathcal{D}}_r) > 0$), and thus by [17, Proposition 11.2.1] the Markov process $\tilde{Y}_t$ admits an invariant measure. In addition, based on the equivalence (D.40) the expected return time $\mathbb{E}[\tau_r]$ to $\tilde{\mathcal{D}}_r$ is uniformly bounded for all initial conditions $\tilde{y}$ on $\tilde{\mathcal{D}}_r$, due to the continuity of the Fenchel coupling $F$. Therefore, invoking [55, Theorem 13.0.1], we conclude that the process $\tilde{Y}_t$ admits a unique invariant probability measure $\tilde{\nu}$, and the law of $\tilde{Y}_t$ converges to $\tilde{\nu}$ *in total variation* for every initial condition $\tilde{y} \in \tilde{\mathcal{Y}}$.

**Step 4: Estimating the long-run occupation measure.** Finally, for the last part, letting $F_t := F(x^*, Y_t)$ and unfolding (B.29b), we obtain:

$$F_t \leq F_0 + \gamma \sum_{s=0}^{t-1} \langle \hat{v}_s, X_s - x^* \rangle + \frac{\gamma^2}{2K} \sum_{s=0}^{t-1} \|\hat{v}_s\|_*^2 \tag{E.31}$$

Taking expectations in both sides, we readily get

$$0 \leq \mathbb{E}[F_t] \leq \mathbb{E}\left[ F_0 + \gamma \sum_{s=0}^{t-1} \langle \hat{v}_s, X_s - x^* \rangle + \frac{\gamma^2}{2K} \sum_{s=0}^{t-1} \|\hat{v}_s\|_*^2 \right]$$

$$\leq F_0 - \alpha\gamma \, \mathbb{E}\left[ \sum_{s=0}^{t-1} \|X_s - x^*\|^2 \right] + t\alpha\gamma r_\sigma^2 \tag{E.32}$$

Therefore, by rearranging terms and dividing both sides by $t$, we have:

$$\frac{1}{t} \mathbb{E}\left[ \sum_{s=0}^{t-1} \|X_s - x^*\|^2 \right] \leq \frac{1}{\alpha\gamma t} F_0 + r_\sigma^2 \tag{E.33}$$

Moreover, we have:

$$\frac{1}{t}\,\mathbb{E}\left[\sum_{s=0}^{t-1}\mathbb{1}\{X_s \notin \mathbb{B}_r(x^*)\}\right] \leq \frac{1}{r^2 t}\,\mathbb{E}\left[\sum_{s=0}^{t-1}\|X_s - x^*\|^2\right] \leq \frac{1}{\alpha\gamma t r^2}F_0 + \frac{r_\sigma^2}{r^2} \tag{E.34}$$

Now, note that $\{X_s \notin \mathbb{B}_r(x^*)\} \equiv \{\tilde{Y}_t \notin \tilde{\mathcal{D}}_r\}$ by construction, and thus

$$\frac{1}{t}\,\mathbb{E}\left[\sum_{s=0}^{t-1}\mathbb{1}\{X_s \notin \mathbb{B}_r(x^*)\}\right] = \frac{1}{t}\,\mathbb{E}\left[\sum_{s=0}^{t-1}\mathbb{1}\{\tilde{Y}_t \notin \tilde{\mathcal{D}}_r\}\right] \tag{E.35}$$

Taking $t \to \infty$, and invoking Birkhoff's individual ergodic theorem [23, Theorem 2.3.4], we readily get that the mean occupation measure $\mathcal{A} \mapsto t^{-1}\,\mathbb{E}\left[\sum_{s=0}^{t-1}\mathbb{1}\{\tilde{Y}_t \in \mathcal{A}\}\right]$ converges strongly to the invariant measure $\tilde{\nu}$, and therefore

$$\lim_{t\to\infty}\frac{1}{t}\,\mathbb{E}\left[\sum_{s=0}^{t-1}\mathbb{1}\{X_s \notin \mathbb{B}_r(x^*)\}\right] = \lim_{t\to\infty}\frac{1}{t}\,\mathbb{E}\left[\sum_{s=0}^{t-1}\mathbb{1}\{\tilde{Y}_t \notin \tilde{\mathcal{D}}_r\}\right] = 1 - \tilde{\nu}(\tilde{\mathcal{D}}_r) \tag{E.36}$$

and, using (E.34), we have:

$$\tilde{\nu}(\tilde{\mathcal{D}}_r) \geq 1 - \frac{r_\sigma^2}{r^2} \tag{E.37}$$

and our proof is complete. ∎

To keep the presentation self-contained, we restate and prove Proposition E.1 and Proposition E.2 below.

**Proposition E.1.** *Let the function $\tilde{p}$ as defined in* (E.22). *Then:*

 *(i) For any compact set $\mathcal{K} \subseteq \mathcal{X}$ and every $\tilde{y} \in \tilde{\mathcal{Y}}$, it holds $\inf_{x\in\mathcal{K}} \tilde{p}(x,\tilde{y}) > 0$.*

 *(ii) The function $\tilde{p}$ is (jointly) lower semi-continuous.*

*Proof.*    (i) For the first part, let $\tilde{y} \in \tilde{\mathcal{Y}}$. Then

$$\inf_{x\in\mathcal{K}}\tilde{p}(x,\tilde{y}) = \inf_{x\in\mathcal{K}}\int_{\widehat{\mathcal{Y}}}p(x,\tilde{y},\hat{y})d\lambda_{\widehat{\mathcal{Y}}}(\hat{y}) \geq \int_{\widehat{\mathcal{Y}}}\inf_{x\in\mathcal{K}}p(x,\tilde{y},\hat{y})d\lambda_{\widehat{\mathcal{Y}}}(\hat{y}) > 0 \tag{E.38}$$

 (ii) For the second part, let $(x,\tilde{y}) \in \mathcal{X}\times\tilde{\mathcal{Y}}$, and let a sequence $\{(x_t,\tilde{y}_t)\}_{t\in\mathbb{N}}$ with $\lim_{t\to\infty}(x_t,\tilde{y}_t) = (x,\tilde{y})$. Since $p$ is jointly continuous, applying Fatou's lemma [19], we get

$$\tilde{p}(x,\tilde{y}) = \int_{\widehat{\mathcal{Y}}}p(x,\tilde{y},\hat{y})d\lambda_{\widehat{\mathcal{Y}}}(\hat{y}) = \int_{\widehat{\mathcal{Y}}}\liminf_{t\to\infty}p(x_t,\tilde{y}_t,\hat{y})d\lambda_{\widehat{\mathcal{Y}}}(\hat{y})$$

$$\leq \liminf_{t\to\infty}\int_{\widehat{\mathcal{Y}}}p(x_t,\tilde{y}_t,\hat{y})d\lambda_{\widehat{\mathcal{Y}}}(\hat{y})$$

$$= \liminf_{t\to\infty}\tilde{p}(x_t,\tilde{y}_t) \tag{E.39}$$

 i.e.,

$$\tilde{p}(x,\tilde{y}) \leq \liminf_{t\to\infty}\tilde{p}(x_t,\tilde{y}_t) \tag{E.40}$$

 and the result follows. ∎

**Proposition E.2.** *The density $\tilde{p}$ satisfies:*

$$0 < \int_{\tilde{\mathcal{Y}}}\inf_{\tilde{y}\in\tilde{\mathcal{D}}_r}\tilde{p}(f(\tilde{y},z))d\lambda_{\tilde{y}}(z) < \infty \tag{E.28}$$

*Proof.* The upper bound is trivial since $\tilde{p}$ is a probability density and

$$\int_{\tilde{\mathcal{Y}}}\inf_{\tilde{y}\in\tilde{\mathcal{D}}_r}\tilde{p}(f(\tilde{y},z))d\lambda_{\tilde{y}}(z) \leq \int_{\tilde{\mathcal{Y}}}\tilde{p}(f(\tilde{y},z))d\lambda_{\tilde{y}}(z) \leq 1 \tag{E.41}$$

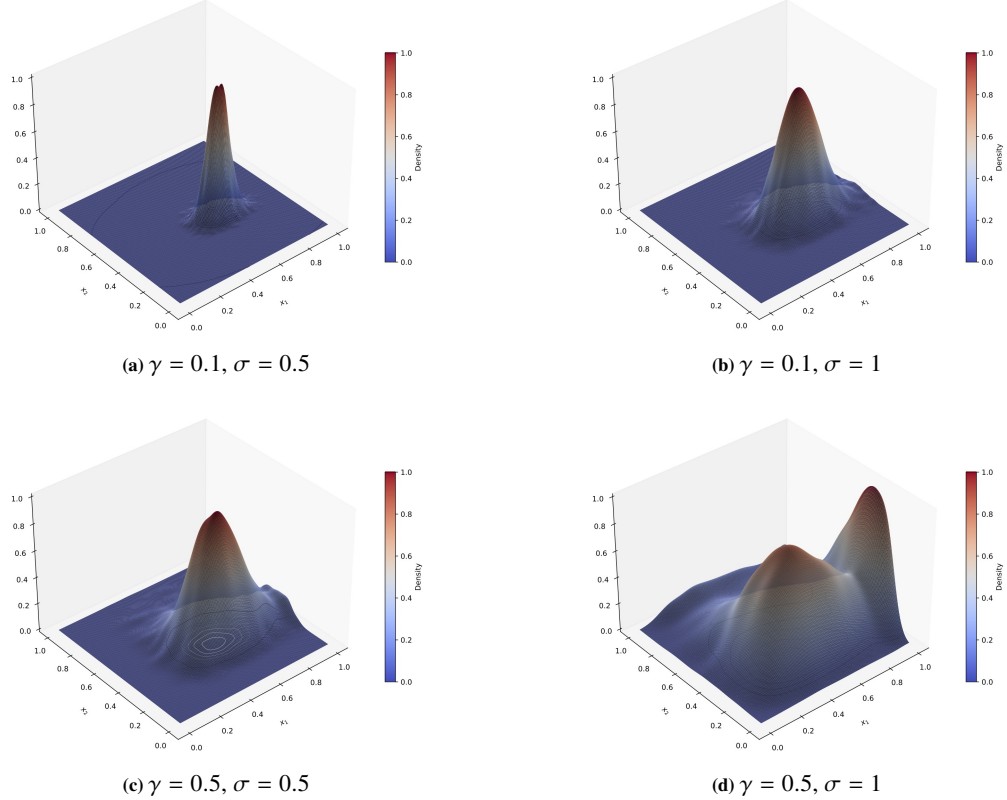

(a) $\gamma = 0.1, \sigma = 0.5$

(b) $\gamma = 0.1, \sigma = 1$

(c) $\gamma = 0.5, \sigma = 0.5$

(d) $\gamma = 0.5, \sigma = 1$

**Figure 2:** Visualization of the long-run occupancy measure for the min-max game with loss-gain function $f(x_1, x_2)$. Each plot shows the empirical density of the final iterates of $10^5$ runs of (FTRL) for $10^2$ steps, starting from uniformly random initial conditions. The surface plot encodes density via both height and color. Each row corresponds to a different step-size $\gamma \in \{0.1, 0.5\}$, while the columns vary the noise level $\sigma \in \{0.5, 1\}$.

For the lower bound, we will show that

$$\inf_{\tilde{y} \in \tilde{\mathcal{D}}_r} \tilde{p}(f(\tilde{y}, z)) > 0 \qquad \text{for all } z \in \tilde{\mathcal{Y}}. \tag{E.42}$$

Suppose not, i.e., there exists $z_0 \in \tilde{\mathcal{Y}}$ such that $\inf_{\tilde{y} \in \tilde{\mathcal{D}}_r} \tilde{p}(f(\tilde{y}, z_0)) = 0$. Since $\tilde{\mathcal{D}}_r$ is compact and $\tilde{p} \circ f$ is lower semi-continuous, the infimum over $\tilde{\mathcal{D}}_r$ is realized, meaning that there exists $\tilde{y}_0 \in \tilde{\mathcal{D}}_r$ such that $\tilde{p}(f(\tilde{y}_0, z_0)) = 0$, or, equivalently,

$$\tilde{p}\Big(\tilde{Q}(\tilde{y}_0), \gamma^{-1}(z_0 - \tilde{y}_0) - \tilde{v}(\tilde{Q}(\tilde{y}_0))\Big) = 0 \tag{E.43}$$

This contradicts Proposition E.1 for $\mathcal{K} \leftarrow \mathcal{K}_r$. Finally, since we integrating over a set with positive measure, our result follows. ∎

# F Further numerical results and details

In this section, we present some additional numerical simulations to illustrate and validate our theoretical findings. To this end, we consider two simple yet representative examples: ($i$) a strongly monotone two-player min-max game on the unit square; and ($ii$) a finite zero-sum game (as an example of a null-monotone game).

**Strongly monotone games.** We consider the strongly monotone two-player min-max game defined by $f : [0, 1] \times [0, 1] \to \mathbb{R}$ with

$$f(x_1, x_2) = -(x_1 - 0.5)^2 + 0.5 x_1 x_2 + 2(x_2 - 0.5)^2 \tag{F.1}$$

and entropic regularization. To be more precise, the payoff functions of the two players are given by $u_1(x_1, x_2) = f(x_1, x_2) = -u_2(x_1, x_2)$, and $x^* = (20/33, 14/33)$ is the unique Nash equilibrium point.

Fig. 2 demonstrates the behavior of (FTRL) under varying step sizes and noise levels for the min-max game defined by the function $f(x_1, x_2)$. Specifically, we consider step-sizes $\gamma \in \{0.1, 0.5\}$, and stochastic feedback of the form $\hat{v}_t = v(X_t) + \sigma\omega_t$, where $\omega \sim N(0, I_2)$ for $\sigma \in \{0.5, 1\}$. For each $(\gamma, \sigma)$ configuration, we perform $10^5$ independent trials, each running for $10^2$ steps. The initial state $Y_0$ for each trial was drawn uniformly at random from $[0, 1]^2$. Each surface represents the empirical density of the final (FTRL) iterates, while the color overlay visualizes their distribution across the $10^5$ independent trials. Warmer (red) regions indicate higher concentration of final iterates, whereas cooler (blue) regions correspond to lower probability of ending in those regions, as indicated by the colorbar on the side. We observe that smaller step sizes and lower noise levels lead to a tighter concentration of the final iterates around the Nash equilibrium. In contrast, increasing either the step size or the noise variance results in a more dispersed distribution. This behavior aligns with both intuition and our theoretical findings: higher noise introduces greater stochastic variability, while larger step sizes amplify this effect by inducing more aggressive updates that are prone to overshooting, ultimately increasing the spread of the iterates.

To further explore the behavior of (FTRL) under different noise levels and step sizes, we conduct an additional set of experiments summarized in Figs. 3 and 4. These figures illustrate the distance from $x^*$ of the final iterate and the hitting time in a neighborhood of $x^*$ with varying radii. Specifically, we consider step sizes $\gamma \in \{0.01, 0.02, 0.05, 0.1, 0.2, 0.5\}$ and stochastic feedback of the form $\hat{v}_t = v(X_t) + \sigma\omega_t$ for noise levels $\sigma \in \{0.01, 0.05, 0.1, 0.5, 1\}$. For each $(\gamma, \sigma)$ configuration, we perform 100 independent runs, each consisting of 10,000 iterations. The initial state $Y_0$ in each run is drawn uniformly at random from $[0, 1]^2$. The first plot reports the *average final distance* of the iterates from the equilibrium, averaged across the 100 runs, while the subsequent plots show the *hitting time* required for the iterates to enter a neighborhood of the equilibrium of radius $r \in \{0.005, 0.01, 0.05, 0.1\}$.

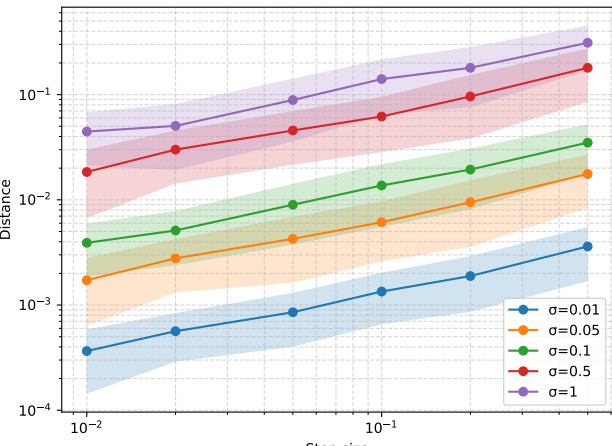

**Figure 3:** Average final distance from equilibrium for different values of the step-size $\gamma$ and the noise level $\sigma$. Each point represents the mean over 100 independent runs of length 10,000, with shaded regions indicating one standard deviation.

**Null-monotone games.** Fig. 5 shows the empirical distribution of the final iterates under the (FTRL) dynamics in the classic matching pennies game with entropic regularization, played over the probability simplex with payoff matrix

$$P = \begin{bmatrix} (+1, -1) & (-1, +1) \\ (-1, +1) & (+1, -1) \end{bmatrix}.$$

The unique Nash equilibrium of the game is the mixed strategy $(0.5, 0.5)$ for both players. As before, we consider stochastic feedback of the form $\hat{v}_t = v(X_t) + \sigma\omega_t$, where $\omega \sim N(0, I_2)$ for $\gamma \in \{0.1, 0.2\}$ and $\sigma \in \{1, 2\}$. For each $(\gamma, \sigma)$ configuration, we perform $10^5$ independent trials, each running for $10^2$ steps. Each surface plot corresponds to a different combination of step-size and noise variance,

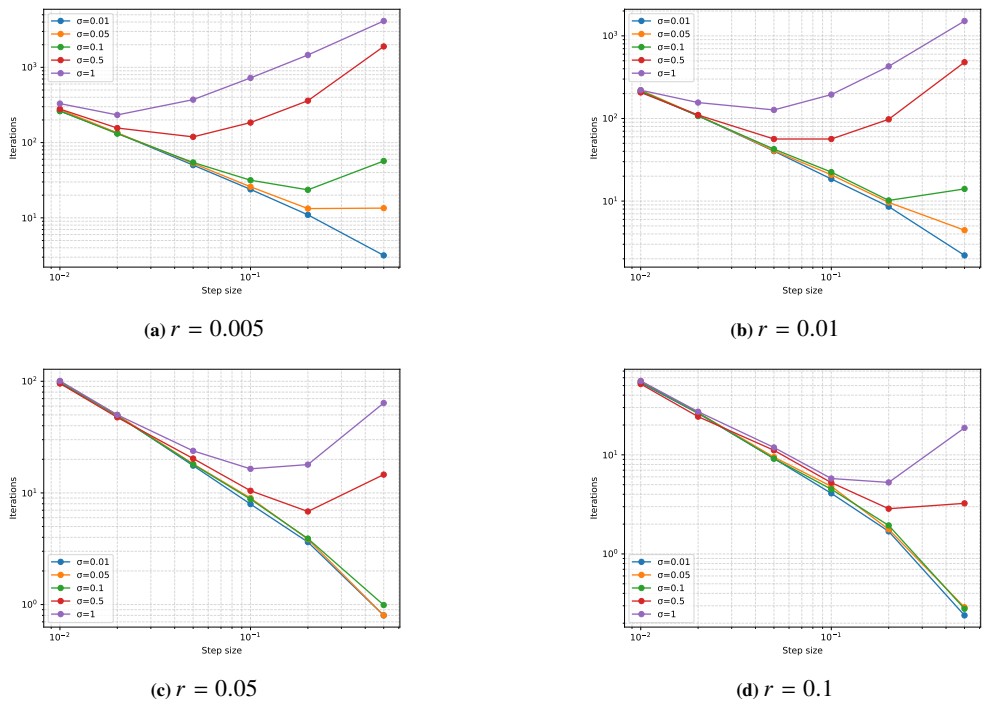

**(a)** $r = 0.005$

**(b)** $r = 0.01$

**(c)** $r = 0.05$

**(d)** $r = 0.1$

**Figure 4:** Average hitting time (in iterations) to a neighborhood of the equilibrium $x^*$ with radius $r \in \{0.005, 0.01, 0.05, 0.1\}$, computed over 100 runs for each $(\gamma, \sigma)$ pair.

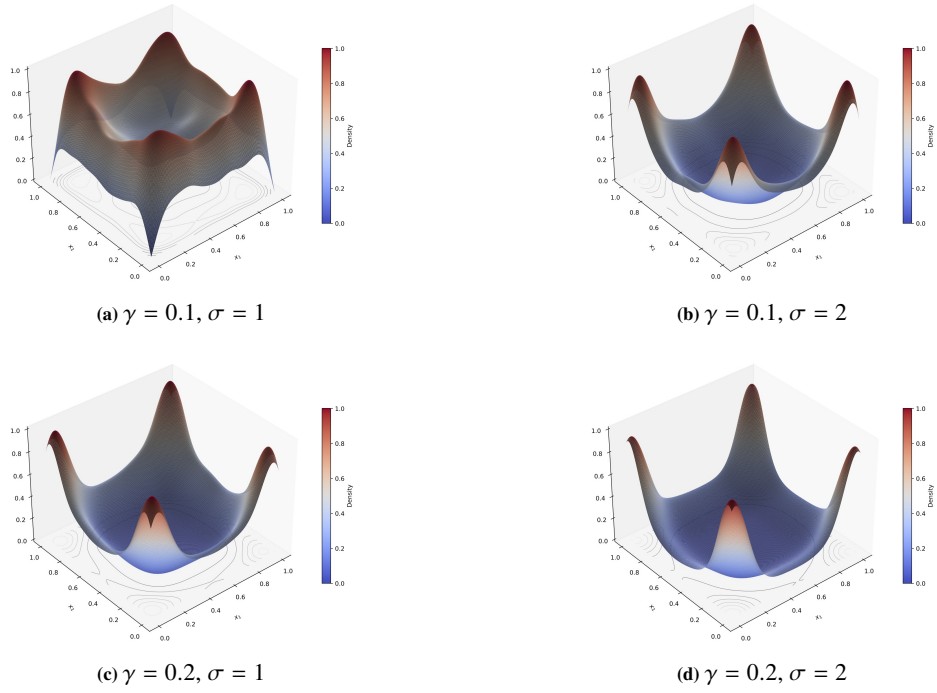

**(a)** $\gamma = 0.1, \sigma = 1$

**(b)** $\gamma = 0.1, \sigma = 2$

**(c)** $\gamma = 0.2, \sigma = 1$

**(d)** $\gamma = 0.2, \sigma = 2$

**Figure 5:** Visualization of the long-run occupancy measure for the bilinear game with entropic regularization. Each plot shows the empirical density of the final iterates of $10^5$ runs of (FTRL) for $10^2$ steps, starting from uniformly random initial conditions. The surface plot encodes density via both height and color. Each row corresponds to a different step-size $\gamma \in \{0.1, 0.2\}$, while the columns vary the noise level $\sigma \in \{1, 2\}$.

with the empirical density of the final iterates represented through both height and color over the simplex domain. We see that across all configurations, the iterates tend to concentrate near the corners of the simplex, reflecting the instability of the interior equilibrium in the presence of noise. This consistent shift toward extreme points highlights the system's inherent tendency to escape the central equilibrium under stochastic perturbations.

