# OpenReview forum: "Multi-Agent Learning under Uncertainty: Recurrence vs. Concentration"
_NeurIPS.cc/2025/Conference — NeurIPS 2025 spotlight_

### Official Review · Reviewer_aVWf · 2025-06-19

**Clarity:** 3
**Significance:** 3
**Originality:** 3
**Rating:** 5
**Confidence:** 3

**Summary:**

This paper investigates the long-run behavior of regularized multi-agent learning algorithms (notably FTRL) in continuous games under persistent uncertainty. It establishes a sharp dichotomy between null-monotone and strongly monotone games: in the former, stochastic dynamics drift away from equilibrium with no concentration, while in the latter, they recurrently return and concentrate around equilibrium within a quantifiable radius tied to noise. The analysis covers both continuous-time (via stochastic differential equations) and discrete-time (with constant step-size) settings, providing explicit estimates on hitting times and invariant distributions. The work offers the first rigorous characterization of when regularized learning dynamics stabilize versus diverge under noise.

**Questions:**

1. The analysis uses general mirror maps defined via strongly convex regularizers. While Euclidean and entropic regularization are covered in examples, it remains unclear whether the qualitative behavior (e.g., recurrence bounds, invariant measure support) is strongly affected by the choice of regularizer. Could the authors comment on how robust their concentration results are to this choice, and whether some regularizers might yield tighter bounds?
2. The discrete-time recurrence argument relies heavily on a minorization condition and the construction of a regeneration structure. Could the authors comment on how realistic or restrictive these assumptions are in practical applications of FTRL (e.g., reinforcement learning or decentralized optimization)? Are there classes of noise where these conditions might fail?
3. If an application is known to induce a null-monotone game, what should a practitioner do—should they regularize differently, or adapt step sizes?
4. The divergence results are shown under constant learning rates. Would using a slightly decaying learning rate (e.g., $\gamma_t = \gamma / \sqrt{t}$) in null-monotone games avoid divergence while preserving good empirical performance?
5. The bounds are clean and intuitive (e.g., scaling with $\sigma^2$ and $1/\alpha$), but it is unclear whether they are conservative or sharp in practice. Do the empirical examples in the paper or supplementary suggest that these are close to optimal?

**Ethical Concerns:**

["NO or VERY MINOR ethics concerns only"]

**Final Justification:**

Thanks to the author for the detailed response. The explanations and justifications clearly addressed my questions and concerns. I'd keep the "accept" score and recommend the paper to be accepted.

**Limitations:**

Yes

**Paper Formatting Concerns:**

No major formatting issues

**Quality:**

3

**Strengths And Weaknesses:**

Strengths:
- Quality: The paper is technically strong, providing rigorous theoretical analysis of stochastic learning dynamics in both continuous and discrete time. The use of tools like Fenchel coupling, Dynkin’s formula, and martingale analysis is precise and appropriate. The theorems are carefully stated and proved, with all assumptions clearly laid out, and the supplementary material offers thorough proofs and additional context.

- Significance: The work addresses a foundational problem in multi-agent learning—understanding the behavior of learning algorithms under uncertainty—through a lens that is both practically relevant (constant step-size FTRL dynamics) and theoretically underexplored. The results clarify which game structures lead to stability versus divergence, providing actionable insight for the design and analysis of learning algorithms in games.

- Originality: The central contribution—a precise recurrence vs. concentration dichotomy for stochastic regularized learning—is novel and fills a clear gap in the literature. While some special cases have been studied (e.g., stochastic approximation with vanishing step sizes), this paper breaks new ground by characterizing the long-run distributional behavior under persistent noise and constant learning rates.

- Clarity: The writing is mostly clear and well-organized, especially considering the technical depth. The structure follows a logical progression from intuition-building examples to general results, with motivating commentary throughout. Figures (e.g., Fig. 1) and appendices are well-integrated and help elucidate the theoretical claims.

Weaknesses:
- Clarity (minor): While the main exposition is clear, the technical density in some sections—especially the discrete-time analysis—may limit accessibility to readers unfamiliar with stochastic processes and advanced convex analysis. A high-level summary of the proof strategies in Section 4 (discrete-time) would help orient the reader before diving into the details.

- Scope of Experiments: The numerical illustrations are minimal and largely illustrative. While this is primarily a theoretical paper, slightly more extensive empirical validation (e.g., tracking empirical hitting times and occupation measures over varied games/noise levels) would help bridge the theory–practice gap and reinforce the significance of the results.

- Assumptions: Some of the regularity conditions (e.g., smoothness of mirror maps, Lipschitz noise models, interior equilibrium) are reasonable but could be discussed more explicitly in terms of how they limit applicability to real-world learning settings.

---

> ### Author Rebuttal · Authors · 2025-07-31
>
> Dear reviewer,
>
> Thank you for your time, your constructive input, and strong positive evaluation! We reply to your remarks and questions below:
>
> > While the main exposition is clear, the technical density in some sections—especially the discrete-time analysis—may limit accessibility to readers unfamiliar with stochastic processes and advanced convex analysis. A high-level summary of the proof strategies in Section 4 (discrete-time) would help orient the reader before diving into the details.
>
> Of course—we would be happy to take advantage of the extra page available at the first revision opportunity to provide a high-level overview of our proof strategy. In a nutshell, this proceeds along the following sequence of steps:
> - First, we reduce the dynamics to a "reduced space" (formally an affine quotient of the dual space), removing redundant directions and ensuring the process evolves within a minimal and non-degenerate domain.
> - Within this reduced space, we show that the induced Markov process satisfies several crucial probabilistic properties. Specifically, we prove:
>     1. **Irreducibility:** any open set in the state space can be reached with positive probability.
>     2. **Minorization:** after entering certain regions of the space, the process mixes sufficiently to allow for probabilistic regeneration.
>     3. **Uniform control of return times:** the expected time to revisit a neighborhood of equilibrium remains bounded regardless of the starting point within that neighborhood.
> - These properties collectively enable the construction of a regeneration structure—a probabilistic framework that ensures the process repeatedly returns to a well-behaved region of the state space with sufficient mixing. In turn, this enables us to establish positive Harris recurrence of the learning dynamics—a key property which ensures the existence and uniqueness of a stationary invariant distribution.
>
> We will be happy to provide a step-by-step walkthrough of this proof structure—thanks for bringing it up.
>
> > The numerical illustrations are minimal and largely illustrative. While this is primarily a theoretical paper, slightly more extensive empirical validation (e.g., tracking empirical hitting times and occupation measures over varied games/noise levels) would help bridge the theory–practice gap and reinforce the significance of the results.
>
> Thank you for the suggestion—we will be happy to provide statistics on hitting times and the occupancy measure of the process to provide an additional layer of empirical validation to our results. The way that OpenReview has been set up this year, it is not possible to include any simulations during the rebuttal stage (or even the result thereof through an anonymized picture or github link), but we commit to do so at the first revision opportunity.
>
> ---
> > Some of the regularity conditions (e.g., smoothness of mirror maps, Lipschitz noise models, interior equilibrium) are reasonable but could be discussed more explicitly in terms of how they limit applicability to real-world learning settings.
>
> Of course—we will be happy to provide a more explicit and detailed discussion of these assumptions. We did not do so in the submitted version because most of them are quite standard and we were constrained for space, but we will take advantage of the extra page available in the revision reound to provide more details on this.
>
> ---
> > The analysis uses general mirror maps defined via strongly convex regularizers. While Euclidean and entropic regularization are covered in examples, it remains unclear whether the qualitative behavior (e.g., recurrence bounds, invariant measure support) is strongly affected by the choice of regularizer. Could the authors comment on how robust their concentration results are to this choice, and whether some regularizers might yield tighter bounds?
>
> The geometry of the regularizer is reflected in its strong convexity modulus $K$, which can in turn be seen as a measure of its curvature relative to the ambient norm of the space (and, dually to the above, the Lipschitz smoothness modulus of the players' choice map $Q$). Our bounds depend implicitly on $K$ and they indicate a trade-off between the degree of concentration of the process around the "event horizon" beyond which the noise dominates the drift, and the time required to hit this region. These bounds tighten when the induced Bregman divergence $D(p,x) = h(p) - h(x) - \langle \nabla h(x),p-x\rangle$ of the regularizer is well-approximated by (a multiple of) the ambient norm squared, but making a precise comparison between different regularizers is otherwise very difficult. We will make sure to point this out in the final version of our paper.
>
> ---
> > The discrete-time recurrence argument relies heavily on a minorization condition and the construction of a regeneration structure. Could the authors comment on how realistic or restrictive these assumptions are in practical applications of FTRL (e.g., reinforcement learning or decentralized optimization)? Are there classes of noise where these conditions might fail?
>
> The noise covariance minorization condition is not restrictive, and it can be trivially achieved by injecting even a small amount of Gaussian noise in the process (e.g., as is done in many incarnations of SGD to ensure that the process does not get stuck in spurious local minima). We can think of somewhat degenerate stochastic first-oracles that do not satisfy this condition—e.g., when the size of a minibatch is particularly large—but, otherwise, this assumption is standard in the literature.
>
> ---
> > If an application is known to induce a null-monotone game, what should a practitioner do—should they regularize differently, or adapt step sizes?
>
> That's an excellent question. In a deterministic context, the standard remedy is to consider an extrapolation-based variant of FTRL such as optimistic or extra-gradient methods. In the presence of uncertainty however, Hsieh et al. ("Explore Aggressively, Update Conservatively: Stochastic Extragradient Methods with Variable Stepsize Scaling", NeurIPS 2020) showed that even extrapolation-based methods fail to converge under uncertainty.
>
> To ensure convergence, a practitioner's best bet in this case would be to take an extrapolation-based variant of FTRL (that is, modify FTRL with an inclusion of an interim gradient-like step), with different step-size policies in the extrapolation and update phases, in the spirit of the paper of Hsieh et al. above. Of course, this would come with additional difficulties from a theoretical standpoint (Hsieh et al. only provide convergence guarantees for unconstrained problems) but this would at least mitigate *some* of the difficulties identified in this context. We will make sure to mention this in the final version of our paper.
>
> ---
> > The divergence results are shown under constant learning rates. Would using a slightly decaying learning rate in null-monotone games avoid divergence while preserving good empirical performance?
>
> Thank you for the insightful question—unfortunately, the short answer is "no".
>
> In more detail, assuming a variable step-size $\gamma_t$, the expectation of the Fenchel-coupling in equation (E.4) of Appendix E can be seen to scale as
> $$
> \mathbb{E}[F_{t+1}]
> \geq \Theta\left(\sum_{k=1}^t \gamma_t^2 \right).
> $$
> As a result, we would still observe non-convergence (in expectation) for any step-size sequence, and divergence (again, on expectation) for any step-size with $\sum_t \gamma_t^2 = \infty$.
>
> This type of behavior is what makes learning under uncertainty particularly difficult in the absence of a consistent drift toward equilibrium. We thank you for giving us the opportunity to discuss this phenomenon, we will be sure to include a version of the above discussion in our revision.
>
> ---
> > The bounds are clean and intuitive, but it is unclear whether they are conservative or sharp in practice. Do the empirical examples in the paper or supplementary suggest that these are close to optimal?
>
> Yes, the bounds are tight—showing this was one of the aims of the "gentle start" in Section 3.1, but we realize that this might not have been as clear as intended.
>
> To make this precise, consider the stochastic gradient descent/ascent (SGDA) dynamics of L155 applied to the simple quadratic min-max problem (6b). These dynamics represent an Ornstein-Uhlenbeck process, so they can be solved explicitly to yield the closed-form expression (9), that is,
> $$
> X(t)
>     = X(0) e^{-t}
>     + \sigma \int_0^t e^{-(t-s)} \: dW(s)
> $$
> where $W$ is an ordinary Brownian motion. Thanks to this *exact* expression, we are able to derive in Appendix D a likewise *exact* expression for the probability transition kernel of the process, and compute the variance of $X(t)$ in closed form as
> $$
> \mathbb{E}[X(t)]
>     = \frac{\sigma^2}{2} (1 - e^{-2t})
> $$
> cf. Eq. (D.7) in Appendix D. [In fact, $X(t)$ is zero-mean Gaussian with variance given by the above expression]
>
> This shows that $X(t)$ cannot achieve a radius of concentration sharper than $\sigma$, an observation which, mutatis mutandis, coincides with the statement and result of Theorem 2 (importantly, all assertions therein can be put in 1-1 correspondence with the respective claims of Proposition 2).
>
> Similar considerations apply to the bilinear / null-monotone framework; however, because the process here diverges to infinity on average, the comparison is, by necessity, more qualitative in nature.
>
> We understand that this was not sufficiently clear in the submitted version, so we will make sure to include a version of the above discussion at the first revision opportunity—many thanks for bringing this up.
>
> ---
> Thank you again for your time and positive evaluation—we remain at your disposal if you have any further questions or remarks, and we are looking forward to further constructive exchanges during the discussion phase!
>
> Kind regards,
>
> The authors

---

### Official Review · Reviewer_a9vV · 2025-06-21

**Clarity:** 2
**Significance:** 1
**Originality:** 2
**Rating:** 5
**Confidence:** 2

**Summary:**

This paper examines how persistent uncertainty affects multi-agent "follow-the-regularized-leader" (FTRL) learning algorithms. The primary contribution is identifying a stark difference in behavior based on the game's structure. In null-monotone games, learning dynamics drift away from equilibrium, showing no long-term concentration. Conversely, in strongly monotone games, the dynamics are pulled toward equilibrium but continue to fluctuate within a concentrated region whose size is determined by the amount of uncertainty. The authors provide formal analysis for this dichotomy in both continuous and discrete-time models.

**Questions:**

* The notion of ‘uncertainty’ has many interpretations. Here, the authors interpret it as noise on the applied gradient updates, which could represent random disturbances in the game. Another interpretation of ‘uncertainty’ is epistemic uncertainty about the policy and learning dynamics of other agents, influencing the best response computation of the agents. Crucially, this epistemic uncertainty can be reduced by observing past actions from the other agents and doing Bayesian posterior updates. Are any insights from this paper relevant to the notion of uncertainty as epistemic uncertainty?
* Line 317: the authors claim that constant step size is much more common in practice. However, in modern machine learning, learning rate schedulers are crucial to obtain competitive results. Hence, the authors should clarify and maybe make other arguments to justify their focus on a constant learning rate. It would also be great to comment on how the core results and insights would change when using learning rate schedulers.
* It would be great to use some example games that are more relevant to real-world settings to illustrate the importance of the insights gained by this paper to a broader audience.

**Ethical Concerns:**

["NO or VERY MINOR ethics concerns only"]

**Final Justification:**

The authors provided further motivation and examples on why their work is relevant for machine learning and multi-agent RL in general, thereby addressing my main concern. Therefore I raised my score to a 5.

**Limitations:**

yes

**Paper Formatting Concerns:**

/

**Quality:**

3

**Strengths And Weaknesses:**

Strengths:
* Theoretical quality: the paper contains a thorough mathematical analysis of the learning dynamics in null-monotone and strongly monotone games, augmented with examples providing further intuitions, and a detailed supplementary material with proofs and required background information.

Weaknesses
* While the authors motivate their contributions by applications to machine learning and data science, it is hard to see how it can be of practical relevance in non-trivial machine learning settings. The assumptions of individually concave and monotone games are rarely satisfied in multi-agent machine learning setups, and it is unclear how the insights of the paper transfer to insights into machine learning. While the authors provide some game-theoretic examples of strongly monotone and null-monotone games, it would be beneficial to provide a discussion on how the core results of these paper can provide insights to more practical machine learning settings.

---

> ### Author Rebuttal · Authors · 2025-07-31
>
> Dear reviewer,
>
> Thank you for your time, your constructive input, and positive evaluation! We reply to your remarks and questions below:
>
> > The assumptions of individually concave and monotone games are rarely satisfied in multi-agent machine learning setups, and it is unclear how the insights of the paper transfer to insights into machine learning.
> > [...]
> > It would be great to use some example games that are more relevant to real-world settings to illustrate the importance of the insights gained by this paper to a broader audience.
>
>
> We would like to point out that our primary motivation lies in applications to online / multi-agent learning, and we provide a series of concrete applications of this type in Appendix A. In addition, the covariance matrix optimization problem described in Example A.3 is also an important application at the intersection of digital signal processing, machine learning, and information theory, and it can be modeled precisely as a monotone game.
>
> In a similar direction, we provide below yet another example with its roots in content-based image retrieval and metric learning for image similarity search is as follows. To that end, consider a database $\mathcal{D}$ of images represented as vectors $v_i\in\mathbb{R}^d$, $i\in\mathcal{D}$, and consider the Mahalanobis distance
> $$
> d_X(v_1,v_2)
> = (v_1-v_2)^\top X (v_1-v_2)
> $$
> with respect to some positive-definite precision matrix $X\in\mathbb{R}^{d\times d}$. If each image $v_i$ has a set of similar and dissimilar images, $S_i$ and $U_i$ respectively, the goal is to find a precision matrix so that $d(v,v') < d(v,v'')$ whenever $v$ is similar to $v'$ but dissimilar to $v''$. By standard arguments, this boils down to minimizing the distributed optimization objective
> $$
> f(X)
> = \sum_{i\in\mathcal{D}} \sum_{j\in S_i} \sum_{k\in U_i}
> L(d_X(v_i,v_j) - d_X(v_i,v_k)) + \varepsilon\|X - I\|_F^2
> $$
> where $L$ is a convex empirical loss function (typically quadratic), $\varepsilon$ is a regularization parameter, and $\|\cdot\|_F$ denotes the Frobenius norm, cf. Bellet et al., "*A survey on metric learning for feature vectors and structured data*".
> To reduce model complexity and to better exploit correlations between features that reduce over-fitting effects, it is also common to proxy low-rank precision matrices by including a trace constraint of the form $\mathrm{tr} X \leq c$ for some $c \ll d$. Finally, when $\mathcal{D}$ is large, it is common to distribute the (convex) objective above over several computing nodes, thus giving rise to a convex potential game played over the spectraplex $\mathcal{X} = \{X\succeq 0 : \mathrm{tr}(X) ≤ c \}$. Since the game admits a (strongly) convex potential, it is strongly monotone, cf. Lim et al., "*Robust structural metric learning*", ICML 2013.
>
> Other ML-relevant examples include the allocation of computing resources in GPU clusters, Kelly auctions for online ad placement, etc. We will be happy to describe these applications in more detail in the first revision opportunity (though, even with the extra page available, it will be challenging to transfer them all to the main part of the text).
>
> ---
> > The notion of ‘uncertainty’ has many interpretations. Here, the authors interpret it as noise on the applied gradient updates, which could represent random disturbances in the game. Another interpretation of ‘uncertainty’ is epistemic uncertainty about the policy and learning dynamics of other agents, influencing the best response computation of the agents. Crucially, this epistemic uncertainty can be reduced by observing past actions from the other agents and doing Bayesian posterior updates. Are any insights from this paper relevant to the notion of uncertainty as epistemic uncertainty?
>
> It is true that uncertainty can be interpreted in multiple ways. In our work, we focus on aleatoric uncertainty, modeled as persistent noise in the agents’ gradient feedback and reflecting random disturbances in the environment, fluctuations in the agents' payoffs, etc. This type of uncertainty is inherent and irreducible, and it directly affects the agents' learning dynamics.
>
> By contrast, epistemic uncertainty refers to an agent’s lack of knowledge about the game—such as the policies, strategies, or update rules of the other players. This type of uncertainty can indeed be reduced over time through observation and inference, for example via Bayesian updating. However, such inference presumes that an agent knows what they are trying to model: the number of other agents, their action spaces, and a hypothesis class for their behavior. Our framework explicitly assumes none of this information is available—in fact, the players may not even be aware that they are involved in a game. Specifically, each player is assumed to operate in an online, decentralized setting, receiving only noisy feedback about their own payoffs. They have no visibility regarding the number or behavior of the other players, let alone the structure of their opponents' strategy spaces, so epistemic issues do not come about in this context.
>
> In this sense, drawing a direct connection between our analysis and epistemic uncertainty is difficult and lies beyond the scope of this work. While both forms of uncertainty affect the agents' learning behavior, they do so in fundamentally different ways, and they have arise from drastically different origins and modeling assumptions. Extending our framework to incorporate belief formation or adaptive modeling of other agents would be an interesting direction for future work, but it would be a completely different paper.
>
> ---
> > Line 317: the authors claim that constant step size is much more common in practice. However, in modern machine learning, learning rate schedulers are crucial to obtain competitive results. Hence, the authors should clarify and maybe make other arguments to justify their focus on a constant learning rate. It would also be great to comment on how the core results and insights would change when using learning rate schedulers.
>
> We would like to point out here that our remark in L317 that constant step-sizes are "more common in practice" was made in reference to *vanishing* step-size schedules, not the more sophisticated cosine / restart schedulers that are used in many state-of-the-art ML models.
>
> With regard to this, the use of a vanishing step-size—typically satisfying a Robbins-Monro condition of the form $\sum_n \gamma_n = \infty$ and $\sum_n \gamma_n^2 < \infty$ or a variant thereof—is quite common in stochastic approximation, but it comes with significant caveats. Chief among these is their inherent inverse recency bias: as the step-size shrinks, newer information receives less weight in the update process, which runs contrary to the guiding principle that iterative algorithms should prioritize recent data, as emphasized by Nesterov (2009, p. 224).
>
> These limitations manifest both in theory and in practice:
>
> - **From a theory standpoint,** vanishing step-sizes can degrade performance. Notably, FTRL under such schedules can exhibit superlinear regret even in simple settings like the exponential weights algorithm, as shown by Orabona and Pál (2018). This is one of the main reasons that FTRL has been primarily developed and analyzed in the online learning literature with constant step-size policies, cf. Shalev-Shwartz (2011), Lattimore & Szepesvári (2020), and references therein.
> - **From a practical standpoint,** algorithmic variants with a vanishing step-size are often sluggish in the beginning, requiring many iterations before reaching a neighborhood of an equilibrium. By contrast, constant step-size methods tend to reach very rapidly the vicinity of a solution (often within  0.1% accuracy), cf. Dieuleveut et al. (2020).
>
> Now, in terms of non-vanishing training schedules, many state-of-the-art transformers and LLMs utilize step-size schedules that are often constant over billions (or even trillions) of samples. As an example, we cite below an excerpt from the training schedule used for DeepSeek (DeepSeek v3 technical report, Section 4.2):
>
> ```
> We keep a constant learning rate of 2.2 × 10−4 until the model consumes 10T training tokens.
> [...]
> During the training of the final 500B tokens, we keep a constant learning rate of 2.2 × 10−5 in the first 333B tokens, and switch to another constant learning rate of 7.3 × 10−6 in the remaining 167B tokens.
> ```
>
> Of course, there are no "one-size-fits-all" solutions and, ultimately, both vanishing, and non-vanishing/ constant step-size methods have their respective strengths and limitations. Choosing the appropriate approach depends critically on the specific context and goals of the problem at hand, and while our analysis does not cover the more sophisticated cosine / restarting step-size schedulers used in some DL applications, it opens the door to a rigorous analysis of their properties and long-run behavior.
>
> ---
> Thank you again for your time and positive evaluation—we remain at your disposal if you have any further questions or remarks, and we are looking forward to further constructive exchanges during the discussion phase!
>
> Kind regards,
>
> The authors

---

> > ### Comment · Reviewer_a9vV · 2025-08-01
> >
> > Thank you for the detailed answers, they addressed my main concerns and I raised my score correspondingly.

---

> > > ### Author Response · Authors · 2025-08-03
> > >
> > > We are sincerely grateful for your thoughful input, questions and engagement—the exchange was a pleasure, thank you again!
> > >
> > > Warm regards,
> > >
> > > The authors

---

### Official Review · Reviewer_kZLR · 2025-07-01

**Clarity:** 3
**Significance:** 3
**Originality:** 3
**Rating:** 5
**Confidence:** 3

**Summary:**

This paper analyses the convergence of follow-the-regularized-leader (FTRL) algorithms in games with uncertainty, in both the continuous and discrete case.
The authors have both negative and positive results. On the negative side, they show that - differently from  full-information models of learning - games under uncertainty do not converge in general. On the positive side, they show that in strongly monotone games FTRL always return to a neighbourhood of the equilibrium in finite time (which they estimate).

**Questions:**

1) The authors make some assumptions about the class of games they analyse (actions selected from a closed convex subset, payoff individually concave, \beta-Lipschitz smoothness). It would be helpful to understand the strength of those assumptions, as well as which results break down if such assumption are removed.

2) Def. 1: I don't understand the difference between merely monotone and null-monotone, as in both cases \alpha = 0.

3) The authors mention different types of uncertainty in the introduction, but then in Sec. 3 all these are modelled as Brownian motion. Is this a suitable way of modelling all the different types of uncertainty mentioned in the introduction?

4) Results are provided for null monotone games and strict monotone games. Any intuition about the situation for merely monotone games.

**Ethical Concerns:**

["NO or VERY MINOR ethics concerns only"]

**Final Justification:**

Happy with the paper and rebuttal.

**Limitations:**

1) The related literature is discussed and compared, although there is no dedicated related work section.
For clarity, I'd suggest the authors add a section specifically devoted to comparison with the SOTA.

2) The authors make some assumptions about the class of games they analyse (actions selected from a closed convex subset, payoff individually concave, \beta-Lipschitz smoothness, \sigma is bounded and Lipschitz
continuous). Please refer to the question above.

**Quality:**

4

**Strengths And Weaknesses:**

This paper makes two main contributions:

+ The authors show that in null-monotone games with an unbounded action space, the sequence of play under FTRL drifts away to infinity on average.

+ In strongly monotone games, the authors show that the mean time required to reach a given distance
from the game's equilibrium is finite, and they provide an explicit estimate thereof.

+ The FTRL model of learning encompasses several relevant multi-agent learning algorithms of practical interest.

+ The results appear to be correct, although I did not check all the details.

- The paper is extremely technical and definitely not an easy reading. However, the authors make an effort to make their paper accessible to a wider audience, e.g., consider Sec. 3.1.

---

> ### Author Rebuttal · Authors · 2025-07-30
>
> Dear reviewer,
>
> Thank you for your time, positive evaluation and encouraging words! We reply to your remarks and questions below:
>
> ---
> > The authors make some assumptions about the class of games they analyse (actions selected from a closed convex subset, payoff individually concave, \beta-Lipschitz smoothness). It would be helpful to understand the strength of those assumptions, as well as which results break down if such assumption are removed.
>
> These assumptions are all quite standard in the study of online learning and games with continuous action spaces, and most of the positive results in the literature hinge on a variant of these assumptions.
>
> With regard to the specific assumptions that you mention, some concern the structure of the problem at hand, and some its regularity. The role of each type of assumptions is different, so we present some details based on this grouping below, before discussing their eventual relaxation.
>
> 1. First, regarding our structural assumptions—that is, convexity of the players' action sets and monotonicity of the game being played. These are both required to have an amenable "convex structure", which is in turn required to establish the existence of an equilibrium and/or be able to characterize it via first-order variational conditions. Without a structural characteriztion of this type, it is not even clear what type of solution concept could be relevant or meaningful, and any hope of global convergence is ruled out. In a sense, these assumptions comprise the game-theoretic equivalent of the convex / non-convex separation in minimization problems—but with the added difficulty that, in non-convex games, equilibria may not even exist, whereas minimizers (either local or global) are generically guaranteed to exist in non-convex minimization problems.
>
> 2. Second, regarding our regularity assumptions (closedness of the players' action sets, Lipschitz smoothness, etc): these assumptions are mostly technical in nature and, for the most part, they can be relaxed at the cost of a (significantly) more involved presentation. They are very common in practice and, as such, they are almost universal in the literature.
>
> Now, regarding the relaxation of these assumptions: relaxing the latter (regularity assumptions) is possible, e.g., by assuming some variant of Lipschitz smoothness (such as local Lipschitz smoothness, relative smoothness, Hölder smoothness or some other relaxation), and the end results should be comparable (but at the cost of a significantly more difficult trajectory to achieve them). Relaxing the former (the problem's structural assumptions) is considerably trickier: some of our results would go through as long as the game's equilibrium admits a global variational characterization (e.g., via a Minty variational inequality or a similar structure). If, however, the game admits distinct components of equilibria, all bets are off: in that case, FTRL (and most other learning schemes that we are aware of) could transit in perpetuity between the game's different equilibrium components, and characterizing the mean sojourn and transition times of the process would only be possible in very specific settings.
>
> All in all, the set of assumptions that we consider represents a certain "sweet spot" between theoretical tractability and practical relevance, and this is why they are so widely used in the literature.
>
> ---
> > Def. 1: I don't understand the difference between merely monotone and null-monotone, as in both cases α = 0.
>
> "*Null-monotone*" means that the inequality holds with equality everywhere, i.e.,
> $$
> \langle v(x')-v(x), x-x' \rangle = 0
> \quad
> \text{for all $x,x' \in \mathcal{X}$}.
> $$
> By contrast, "*merely monotone*" (or, more simply, "*monotone*") means that
> $$
> \langle v(x')-v(x), x-x' \rangle \leq 0
> \quad
> \text{for all $x,x' \in \mathcal{X}$}
> $$
> i.e., that the defining inequality holds with $\alpha=0$ (but not necessarily as an equality throughout).
>
> ---
> > The authors mention different types of uncertainty in the introduction, but then in Sec. 3 all these are modelled as Brownian motion. Is this a suitable way of modelling all the different types of uncertainty mentioned in the introduction?
>
> We should point out here that we only use Brownian motion as a model in Section 3.1, which is intended as a "gentle start" for our more general model. In Section 3.2, payoff shocks and disturbances are instead modeled by an *arbitrary* martingale process $M_i(t)$, which acts as a "catch-all" disturbance term (accounting for "colored noise" effects, correlations, stochastic dependencies, etc.). The representation (12) of $M(t)$ via a Brownian integrator is not an assumption per se, but a consequence of the martingale representation theorem, which allows us to express any homogeneous square-integrable martingale in this form.
>
> We mentioned this briefly in the footnote at the bottom of p. 6, but we realize that the current presentation could potentially lead to confusion on this point. We will be happy to make this clear and include a version of the above discussion in the first revision opportunity.
>
> ---
> > Results are provided for null monotone games and strict monotone games. Any intuition about the situation for merely monotone games?
>
> In the context of our paper, merely monotone games could be viewed as a "hybrid" between null and strictly monotone games. Specifically, at any given action profile of a merely monotone game, there would be directions of motion where the (symmetrized) Jacobian of the players' gradient field has a zero eigenvalue, and directions of motion with positive eigenvalues; either set (but not both) could be empty, the former corresponding to the "null-monotone" directions, the latter corresponding to the "strongly monotone" ones. Our analysis suggests that, in the presence of uncertainty, FTRL would tend to "wander around" the null-monotone directions, and be carried along the strongly monotone directions toward the game's set of equilibria. However, making this statement precise would require a highly technical treatment that lies well beyond the scope of our paper, so we leave this undertaking as a promising direction for future research (which we will be sure to mention in the revised version of our paper).
>
> > The related literature is discussed and compared, although there is no dedicated related work section. For clarity, I'd suggest the authors add a section specifically devoted to comparison with the SOTA.
>
> Thank you very much for the suggestion—absolutely, we will be happy to organize the related work under a dedicated section to make the comparison with the state of the art clearer and easier to follow. We will implement this minor restructuring in the first revision opportunity.
>
> ---
> Thank you again for your input and strongly positive evaluation—we remain at your disposal if you have any further questions or remarks, and we are looking forward to further constructive exchanges during the discussion phase!
>
> Kind regards,
>
> The authors

---

### Official Review · Reviewer_M4ZL · 2025-07-03

**Clarity:** 4
**Significance:** 4
**Originality:** 4
**Rating:** 5
**Confidence:** 2

**Summary:**

The paper investigates the problem of convergence in multi-agent learning with noisy feedback. The authors start from analyzing the hitting times and long-run behavior of stochastic gradient descent/ascent (S-GDA) dynamics the bilinear min-max and quadratic min-max games in continuous time, which are stylized examples of null-monotone and strongly monotone games. They proceed with the analysis in the general case of follow-the-regularized-leader (FTRL) dynamics, which contains S-GDA as a special case, in null-monotone and strongly monotone games. They show that, in null-monotone games, the dynamics drift away from equilibrium and requires infinite time to return, whereas in strongly monotone games, the dynamics drift towards the unique equilibrium and return arbitrarily close to its origin infinitely often with probability 1. Next, the authors explore the FTRL dynamics under standard assumptions in discrete-time using their insights from the continuous-time analysis. Similar to the continuous-time case, they prove that the dynamics drift away to infinity in null-monotone games and drift towards the equilibrium in strongly-monotone games.

**Questions:**

1. It is noted that the standard tools of stochastic analysis are often inapplicable in the discrete time analysis of FTRL's long-run behavior. Could you shortly clarify the novelties and technical difficulties setting the current analysis apart from the standard techniques in the literature, particularly in the (vanishing step size) FTRL analysis in discrete time?

**Ethical Concerns:**

["NO or VERY MINOR ethics concerns only"]

**Final Justification:**

The authors analyze the FTRL algorithm in the theoretically understudied case of constant step-sizes. This non-trivial extension has a theoretical and empirical justification as also noted by other reviewers. The paper is clearly-written, and the authors will include further discussion that will enhance the readability. The paper is also mostly self-consistent with Appendices B and C.

**Limitations:**

yes

**Quality:**

4

**Strengths And Weaknesses:**

**Quality and Clarity:** The paper is clearly written and appears technically sound. The organization of the paper makes it easy-to-follow. The illustrations in toy examples nicely accompanies the results and discussion. It would also interesting to see the results in play in larger-scale experiments.

**Significance and Originality:** The work differs from existing works as they explore either (1) a more general class of algorithms (e.g. FTRL rather than S-GDA) or (2) a different choice of step sizes (constant rather than vanishing). The constant step size in FTRL is noted by the authors as the dominant choice in practice due to its simplicity, robustness and superior empirical performance.

---

> ### Author Rebuttal · Authors · 2025-07-30
>
> Dear reviewer,
>
> Thank you for your time, positive evaluation and encouraging words! We reply to your remarks and questions below:
>
> > It is noted that the standard tools of stochastic analysis are often inapplicable in the discrete time analysis of FTRL's long-run behavior. Could you shortly clarify the novelties and technical difficulties setting the current analysis apart from the standard techniques in the literature, particularly in the (vanishing step size) FTRL analysis in discrete time?
>
> Your question has two parts: stochastic analysis techniques (for the continuous-time setting), and stochastic approximation techniques (for the discrete-time analysis in the vanishing step-size regime).
>
> 1. Regarding the former (comparison to the continuous-time regime), the core technical difficulty is the lack of an analogue of Itô's forumla and, a fortiori, a suitable Dynkin lemma for getting concrete hitting time estimates based on the infinitesimal generator of the process. Because of this, estimating the invariant measure of the process (and/or establishing recurrence) is a much more difficult affair which requires a series of intricate Lyapunov estimates from the theory of discrete-time Markov chains, used in turn to establish Lebesgue irreducibility and Harris recurrence (see Appendix E for the details). Specifically, in continuous time, positive recurrence and the existence of an invariant distribution are almost immediate once we establish uniform ellipticity of the diffusion process (Steps 2 and 3 of the proof of Theorem 2 in Appendix D).  By contrast, in the discrete-time regime, the analysis requires a regeneration argument to ensure that the process repeatedly returns to a well-behaved region of the state space with sufficient mixing. This regeneration structure is crucial for establishing positive Harris recurrence of the learning dynamics—a key stochastic stability property that guarantees the existence and uniqueness of a stationary invariant distribution. These subtleties are unique to the homogeneous, discrete-time setting, and they have no bona fide analogue in continuous time.
>
> 2. Regarding the latter (comparison to the stochastic approximation regime), it should be noted that the use of a vanishing step-size ultimately means that the noise "washes out" in the long run, and the trajectories of the process can be well-approximated by the flow of an ordinary differential equation (the precise notion is that of an asympotic pseudotrajectory, cf. Benaïm, 1999, and references therein). Because of this, the long-run behavior of FTRL with a vanishing step-size is fundamentally different, and is captured by the so-called internally chain transitive (ICT) sets of a *deterministic* dynamical system. By contrast, the constant-step version of FTRL exhibits ergodicity, so one is naturally led to study questions of concentration instead of convergence—which, is significantly more demanding than studying a deterministic ODE, even in the presence of a Lyapunov function.
>
> ---
> Thank you again for your time and positive evaluation—we remain at your disposal if you have any further questions or remarks, and we are looking forward to further constructive exchanges during the discussion phase!
>
> Kind regards,
>
> The authors

---

> > ### Comment · Reviewer_M4ZL · 2025-08-04
> >
> > I thank the authors for the response. I find the paper interesting. Having read the discussions, I will increase my score. I must, however, note that I have limited exposure to the field and hope that other reviewers will offer a clearer perspective on the significance of the gap this paper addresses in the literature. I also would like to add a few comments:
> > - As also noted in the discussion with reviewer aVWf, it would be helpful to highlight the position of Section 3.1 in the paper and that it shows the tightness of the analysis.
> > - The example provided as a response to reviewer a9vV would be a worthy addition to the Examples section.

---

> > > ### Author Response · Authors · 2025-08-06
> > >
> > > Our sincere thanks for your thoughtful input, your time, and your engagement. We were happy to address your questions, and we will make sure to include the relevant parts of our replies—thank you again!
> > >
> > > Warm regards,
> > >
> > > The authors

---

### Decision · Program_Chairs · 2025-09-17

**Decision:**

Accept (spotlight)

**Comment:**

The paper examines the convergence behavior of regularized learning algorithms in continuous games under the randomness and uncertainty in learning vs when there is no uncertainty. They study two versions of games: null-monotone and strongly monotone. In each of these games, they study both continuous-time and discrete-time variants. They find that, in null-monotone games, uncertainty drifts the dynamics from equilibrium, In strongly monotone games, the uncertainty still drifts away but not as much as in the null-monotone games. In particular, the dynamics will end up in a near-equilibrium region whose size scales with the level of uncertainty.

**Strength**. A well-written paper, novel insights into the dynamic of regularized learning in games under uncertainty -- this is a significant problem that is relevant to both ML theory and practice. Solid and rigorous analysis with good examples and detailed supplemental material.

**Weaknesses**. No major weaknesses from the reviews, mostly concerned about clarifications. I myself find that the consideration of two special classes of games (null-monotone and strongly monotone) is quite limited and would expect/be curious if the same phenomenon about "recurrence vs concentration" would persist in more general classes of games. However, the authors have done a good job of justifying the applications of the games they consider.

The paper initially received positive reviews consistently from all the reviewers. After the rebuttal, one reviewer has raised their score to 5, leaving that the paper receives the score of 5 from all reviewers. I enjoy reading the author-reviewer discussion of this paper. The reviewers have provided insightful questions and the authors have explained well with additional contexts that definitely make the paper more accessible to broader audiences. For example, I particularly like the author answers about the structural and regularity assumptions in response to Reviewer kZRL and the response to Reviewer a9Vv about the justification of the assumptions in the considered games. I would encourage the authors to integrate such discussion as much as you can into the revision, even in the appendix, to make the paper more accessible.